# Rethinking Genomic Modeling Through Optical Character Recognition

Hongxin Xiang [1 2 *]   Pengsen Ma [1 2 *]   Yunkang Cao [2 3]   Di Yu [1 2]   Haowen Chen [1 2]   Xinyu Yang [1 2 ✉]
Xiangxiang Zeng [1 2 ✉]

⌂ Main Page      �services Code      ✆ OpenReview.net

## Abstract

Recent genomic foundation models largely adopt large language model architectures that treat DNA as a one-dimensional token sequence. However, exhaustive sequential reading is structurally misaligned with sparse and discontinuous genomic semantics, leading to wasted computation on low-information background and preventing understanding-driven compression for long contexts. Here, we present OpticalDNA, a vision-based framework that reframes genomic modeling as Optical Character Recognition (OCR)-style document understanding. OpticalDNA renders DNA into structured visual layouts and trains an OCR-capable vision–language model with a *visual DNA encoder* and a *document decoder*, where the encoder produces compact, reconstructible visual tokens for high-fidelity compression. Building on this representation, OpticalDNA defines prompt-conditioned objectives over core genomic primitives—reading, region grounding, subsequence retrieval, and masked span completion—thereby learning layout-aware DNA representations that retain fine-grained genomic information under a reduced effective token budget. Across diverse genomic benchmarks, OpticalDNA consistently outperforms recent baselines; on sequences up to 450k bases, it achieves the best overall performance with nearly $20\times$ fewer effective tokens, and surpasses models with up to $985\times$ more activated parameters while tuning only 256k *trainable* parameters.

---

*Equal contribution [1]College of Computer Science and Electronic Engineering, Hunan University [2]Yuelushan Laboratory [3]School of Artificial Intelligence and Robotics, Hunan University. Correspondence to: Xinyu Yang <yangxinyu621@foxmail.com>, Xiangxiang Zeng <xzeng@hnu.edu.cn>.

*Proceedings of the $43^{rd}$ International Conference on Machine Learning*, Seoul, South Korea. PMLR 306, 2026. Copyright 2026 by the author(s).

## 1. Introduction

DNA sequence modeling is fundamental to understanding gene regulation and genome function (Oliver, 1996), with broad impact on disease diagnosis (Zhang et al., 2020; Dalla-Torre et al., 2025), precision medicine (Ashley, 2016), and drug discovery (Barbadilla-Martínez et al., 2025). Recent advances in large language models (LLMs) have made sequence-based representation learning the dominant paradigm for genomic foundation models (Dalla-Torre et al., 2025; Nguyen et al., 2024; He et al., 2025), framing DNA as a one-dimensional sequence over a four-letter alphabet (A/T/C/G). Despite their success, sequential reading faces fundamental limitations in genomic modeling:

*Limitation 1: lack of structure-aware reading.* Existing genomic foundation models inherit inductive biases from natural language modeling, assuming dense semantics under exhaustive token-by-token reading (Vaswani et al., 2017; Devlin et al., 2019). However, genomic function is sparse and discontinuous, embedded in long stretches of low-information background (Nurk et al., 2022; Rao et al., 2014), leading to jump-like dependencies across distant loci (Fig. 1a). As a result, sequential reading is structurally wasteful: most computation is spent scanning background rather than modeling sparse functional regions (Zaheer et al., 2020). In controlled comparisons, 1D genomic modeling baselines also exhibit inferior accuracy (Fig. 1c; see Appendix A for details) indicating that purely sequential representations are suboptimal even in long-context settings.

*Limitation 2: lack of understanding-driven compression.* More fundamentally, this inefficiency stems from the low-information-density nature of genomic sequences (Reich et al., 2001; Consortium et al., 2012), making compression a first-class operation for scalable modeling. However, high-fidelity compression requires semantic understanding—identifying sparse, task-relevant structures while suppressing extensive background—rather than uniformly processing all tokens (Yu et al., 2023). Existing token-level LLM paradigms largely lack such understanding-driven compression: they still operate on base/$k$-mer tokens (Zhou et al., 2023; Dalla-Torre et al., 2025) and allocate comparable computation to background

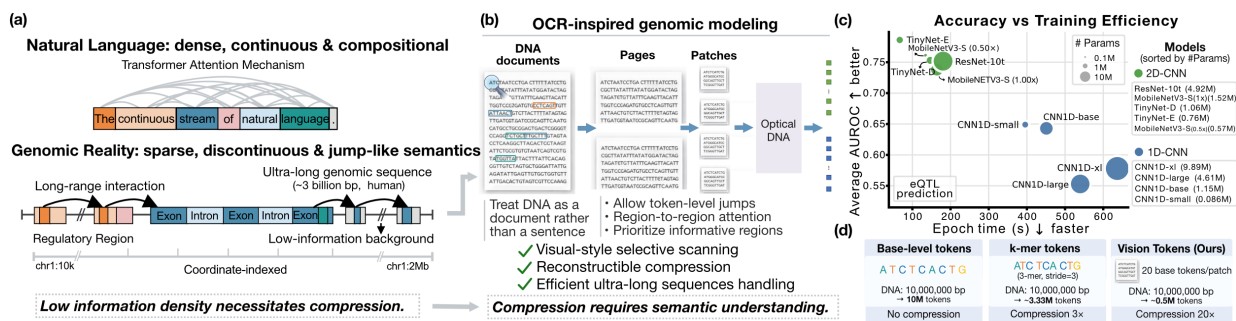

**Figure 1.** From sequential reading to selective genomic scanning. (a) Sparse, discontinuous genomic signals make sequential modeling inefficient. (b) OCR-inspired genomic modeling enables efficient, reconstructible visual compression. (c) 2D CNNs outperform 1D CNNs in accuracy–efficiency trade-offs on eQTL prediction. (d) Vision tokens substantially reduce the effective token count.

and functional regions. Consequently, runtime and memory scale poorly with sequence length, leading to unfavorable computational complexity (Nguyen et al., 2023b). This limitation is also reflected empirically by the poor training efficiency of sequential baselines under the same setting (Fig. 1c; Appendix A), motivating compression mechanisms explicitly coupled with genomic understanding.

To address these challenges, we adopt a vision-based reformulation of genomic modeling and present OpticalDNA, the first framework that casts genomic understanding as an OCR-style document comprehension problem (Blecher et al., 2023; Kim et al., 2022) (see Appendix B for the motivation behind this visual reformulation). Specifically, to overcome Limitation 1, we render 1D sequences into structured 2D layouts, converting discontinuous long-range dependencies into spatial structure amenable to region-aware inductive biases; in controlled comparisons, 2D CNN backbones exhibit a clearly better accuracy–efficiency trade-off than 1D CNN baselines (Fig. 1c). To address Limitation 2, we treat DNA as a document rather than a sentence (Fig. 1b), enabling selective, region-aware processing via token-level jumps. Concretely, base-level content is mapped into compact, reconstructible *vision tokens* (Fig. 1d), substantially reducing the effective token count compared with base or $k$-mer tokenization while preserving fine-grained information. Building on this formulation, OpticalDNA renders genomic sequences into multi-page DNA documents and trains an OCR-capable vision–language model with a DNA visual encoder and a document decoder (Liu et al., 2024). We observe a close alignment between document understanding and genomic reasoning: OCR-style primitives such as grounding, retrieval, and span completion naturally correspond to variant localization (Poplin et al., 2018), subsequence retrieval (Madden, 2013), and missing-interval inference (Browning & Browning, 2016). Guided by this alignment, OpticalDNA introduces genome-oriented, prompt-conditioned objectives over reading, grounding, retrieval, and completion, leveraging the layout of rendered DNA pages to enable

base-level understanding with region-level interpretability (Fig. 2). Overall, our main contributions are as follows:

- To the best of our knowledge, we are the first to reformulate genomic sequence modeling as an OCR-style document understanding problem, introducing a new paradigm for genome-scale representation learning.

- We introduce OpticalDNA, the first vision-based DNA foundation model that learns genomic representations through structured visual layouts and region-centric reasoning, instead of sequential token modeling.

- We design six OCR-inspired tasks aligned with practical genomic analysis workflows, covering core genomic operations (reading, grounding, retrieval, and completion).

- Extensive experiments show that OpticalDNA consistently outperforms state-of-the-art baselines, including models up to $985\times$ larger, while using nearly $20\times$ fewer effective tokens on long-range tasks.

## 2. Related Work

### 2.1. Sequence-based Genomic Foundation Models

Genomic foundation models have predominantly adopted sequence-based formulations inspired by LLMs, representing DNA as a one-dimensional token stream and learning contextual representations via Transformer architectures. Masked language modeling (Zhou et al., 2023; Sanabria et al., 2024; Dalla-Torre et al., 2025), autoregressive modeling (Nguyen et al., 2023b; 2024), and long-context extensions (Avsec et al., 2021) have been widely explored, yielding strong performance on tasks such as promoter detection (Zhou et al., 2023), regulatory element prediction (Marin et al., 2023), and functional annotation (Quang & Xie, 2016; Feng et al., 2025b). To address the extreme length of genomic sequences, prior work has introduced architectural modifications including convolutional front-ends

(Quang & Xie, 2016), hierarchical pooling (Avsec et al., 2021), sparse or linear attention (Yang et al., 2022), and memory-based mechanisms (Nguyen et al., 2023b). These techniques reduce computational cost by compressing local neighborhoods or restricting attention patterns, while preserving the sequence modeling paradigm. However, since sequence-based models fundamentally treat genomes as flat token sequences, genomic coordinates and interval-level operations—central to real-world genomic analysis (Quinlan & Hall, 2010)—are only implicitly encoded through positional embeddings or attention weights (Avsec et al., 2021; Zhou et al., 2023), making region-specific inspection, subsequence retrieval, and explicit evidence localization indirect and difficult to supervise (Novakovsky et al., 2023). In contrast, departing from token-centric sequence modeling, we represent genomes as coordinate-indexed objects and elevate interval localization, retrieval, and region-based reasoning to first-class primitives.

## 2.2. Optical Character Recognition Models

Recent years have seen rapid progress in large-scale OCR foundation models (Team et al., 2025; Feng et al., 2025a; Wei et al., 2025), advancing from character-level transcription to holistic document understanding under unified vision–language training (Xu et al., 2020; Kim et al., 2022). Modern OCR systems increasingly incorporate layout- and structure-aware objectives, enabling long-document processing with explicit spatial organization and flexible access patterns (Huang et al., 2022; Lee et al., 2023; Blecher et al., 2023). In particular, vision–language models trained with OCR-style supervision leverage coordinate-anchored text representations to jointly model *what* content is present and *where* it appears, supporting region-level inspection and grounded interactions (Luo et al., 2024; Wei et al., 2025; Cheng et al., 2025a) that are difficult to express in purely sequential formulations. Although OCR has been extensively studied for natural documents, its application to genomic sequences remains largely unexplored; therefore, we first bridge OCR and genomic modeling by reformulating genomic analysis as an OCR-style learning problem.

# 3. OpticalDNA

## 3.1. Problem Formulation and Method Overview

Fig. 2 summarizes our pipeline. We recast DNA sequence modeling as a document understanding problem. Given a genomic sequence $S = (s_1, \ldots, s_N)$ of length $N$, where each $s_i \in \{A, C, G, T, N\}$, we render it into a structured multi-page DNA document $\mathcal{D}(S)$ consisting of $P$ pages:

$$\mathcal{D}(S) = \left( \{\text{page}_p\}_{p=0}^{P-1}, \{\mathcal{B}^{(p)}\}_{p=0}^{P-1} \right), \quad (1)$$

where $p$ indexes the pages, $\text{page}_p \in \mathbb{R}^{H \times W \times 3}$, and $\mathcal{B}^{(p)}$ denotes bounding boxes of DNA intervals on page $p$ (see Section 3.2). To support coordinate-indexed reasoning, we define an interval-to-region mapping $\Phi : (i, j) \mapsto b_{ij}$ and inverse region-to-interval mapping $\Phi^{-1}$. Each region is represented as a bounding box $b_{ij} = [img\_id, x_1, y_1, x_2, y_2]$, with $img\_id \in \{0, \ldots, P-1\}$ indicating the page index and $(x_1, y_1, x_2, y_2)$ specified in the original page pixel coordinates. $0 \le i < j < N$ defines a contiguous interval in $\mathcal{S}$; $b_{ij}$ is extracted from the dense annotations $\{\mathcal{B}^{(p)}\}$ in DNA document $D$. Based on $\Phi$ and $\{\mathcal{B}^{(p)}\}$, we instantiate prompts $q \in \mathcal{Q}$ for six OCR-style tasks spanning four genomic primitives (recognition, grounding, retrieval, completion; see Section 3.3). To understand $D$, we pretrain a visual encoder $E_\theta$ and a document decoder $G_\psi$ with prompt-conditioned document supervision: the encoder takes only the page images and produces visual tokens $Z = E_\theta(\{\text{page}_p\})$, and the decoder generates task outputs $\hat{Y} = G_\psi(Z, q)$ (see Section 3.4). This shifts learning from global sequence-token prediction to prompt-driven region supervision over a spatially indexed genome. After pretraining, we reuse the pretrained encoder as a general-purpose representation extractor and attach a lightweight MLP head $g_\phi$ for downstream prediction, i.e., $\hat{y} = g_\phi(E_\theta(\{\text{page}_p\}))$, with pretraining and downstream training details described in Section 3.5.

## 3.2. DNA Document Generation

**DNA documents with bounding boxes.** Given a DNA sequence $S$ of length $N$ loaded from FASTA, we convert it into a *DNA document* $\mathcal{D}(S)$. This representation links 1D sequence positions to 2D pixel regions, enabling base-level localization and region-of-interest (ROI)-based operations.

**Rendering DNA sequences into pages.** We rasterize $S$ as monospace-like text onto a fixed-resolution canvas, writing nucleotides row-by-row (left-to-right within each row, then top-to-bottom across rows). The rendered content contains only A/C/G/T/N; any lowercase letters in the source are converted to uppercase, and we do not insert auxiliary markers such as indices, separators, or coordinate tokens. If the full sequence cannot fit into a single page, rendering continues onto subsequent pages until completion. Unless otherwise specified, we use font_size=14 and line_spacing=1.6, which typically yields approximately 1,800 nucleotides per page under our configuration.

**Nucleotide-level bounding-box annotations.** For each page $p$, we store an ordered list of nucleotide annotations:

$$\mathcal{B}^{(p)} = \left[ b_0^{(p)}, b_1^{(p)}, \ldots, b_{N_p-1}^{(p)} \right], \quad (2)$$

where $N_p$ is the number of nucleotides rendered on page $p$. Each $b_k^{(p)}$, $k \in \{0, \ldots, N_p - 1\}$ comprises a character $c_k^{(p)}$,

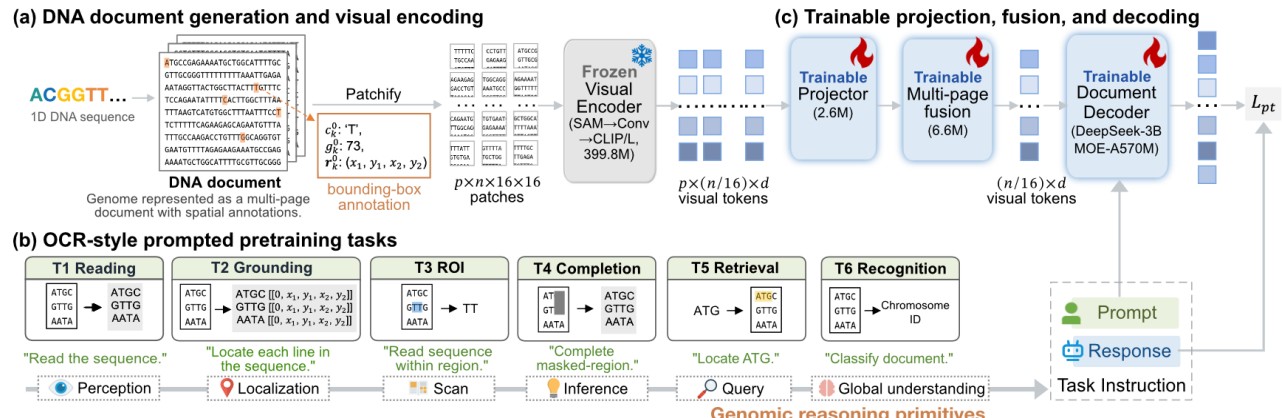

*Figure 2.* Overview of OpticalDNA. (a) Render a 1D genomic sequence into a multi-page DNA document with bounding-box annotations. (b) Construct six OCR-style prompted genomic tasks. (c) Pretrain a visual encoder–document decoder under prompt supervision.

a global index $g_k^{(p)}$, and a pixel-level bounding box $\mathbf{r}_k^{(p)}$:

$$b_k^{(p)} = \left(c_k^{(p)},\ g_k^{(p)},\ \mathbf{r}_k^{(p)}\right), \qquad \mathbf{r}_k^{(p)} = (x_1, y_1, x_2, y_2). \tag{3}$$

Here $c_k^{(p)} \in \{\mathtt{A}, \mathtt{C}, \mathtt{G}, \mathtt{T}, \mathtt{N}\}$ is the rendered nucleotide, $g_k^{(p)} \in [0, N)$ is the global nucleotide index in $S$, and $\mathbf{r}_k^{(p)}$ denotes the glyph bounding box in page pixel coordinates.

### 3.3. OCR-Style Prompted Pretraining Tasks

To pretrain OpticalDNA, we formulate genomic understanding as an OCR-style prompted learning paradigm over a unified conversation interface. We construct six prompt families (T1–T6) spanning four genomic primitives: recognition, grounding, retrieval, and completion (Table 1; see Appendix C for our design principles). Specifically, T1 performs free-form DNA transcription; T2 couples transcription with spatial grounding; T3 transcribes DNA within given ROIs; T4 completes masked regions within given ROIs; T5 localizes all occurrences of a query subsequence; and T6 predicts a chromosome-level class label.

Each training instance is represented as a structured multimodal conversation with exactly two turns:

$$\mathcal{C} = [m^{(u)}, m^{(a)}], \qquad m = (\mathtt{role}, \mathtt{content}, \mathtt{images}), \tag{4}$$

where $m^{(u)}$ specifies the task prompt and $m^{(a)}$ provides the supervision. Given a DNA document $D$ and a prompt $q \in \mathcal{Q}$, we construct the user message as $m^{(u)} = (\mathtt{<|User|>}, q, \mathrm{pages}(D))$ and the assistant message as $m^{(a)} = (\mathtt{<|Assistant|>}, y, \varnothing)$, where $\mathrm{pages}(D) = \{\mathrm{page}_p\}_{p=0}^{P-1}$ denotes the ordered page images carried in the $\mathtt{images}$ field, $y$ is the task-specific target and $\varnothing$ denotes an empty field. For clarity and reproducibility, we detail the conversation construction in Appendix D and provide task-specific prompt definitions and examples in Appendix E.

**Task supervision spaces.** Across tasks, we define a unified supervision space $y \in \mathcal{Y}_t$, where $t \in \{1, \ldots, 6\}$ indexes the tasks. Let $\Sigma = \{\mathtt{A}, \mathtt{C}, \mathtt{G}, \mathtt{T}, \mathtt{N}\}$ denote the DNA alphabet and $b = [img\_id, x_1, y_1, x_2, y_2]$ a pixel-space region; task-specific targets $y \in \mathcal{Y}_t$ are defined as:

$$\mathcal{Y}_1 = \Sigma^*, \quad \mathcal{Y}_2 = (\Sigma^* \times \mathbb{B})^*, \quad \mathcal{Y}_3 = (\Sigma^* \times \mathbb{B})^K,$$
$$\mathcal{Y}_4 = (\Sigma^* \times \mathbb{B})^K, \quad \mathcal{Y}_5 = \Sigma^* \times \mathbb{B}^*, \quad \mathcal{Y}_6 = \mathcal{Y}, \tag{5}$$

where $\mathbb{B}$ is the space of bounding boxes $b$, and $(\cdot)^*$ denotes a variable-length list. Concretely, T2 outputs a list of grounded pairs $(s_i, b_i)$ where $s_i \in \Sigma^*$ is the transcribed DNA string and $b_i \in \mathbb{B}$ is its grounded region; T3 outputs $K$ ROI pairs $(s_k, b_k)$ for user-provided boxes; T4 outputs $K$ masked-completion pairs $(\tilde{s}_k, b_k)$; T5 outputs a query $q \in \Sigma^*$ with a list of matched boxes; and T6 outputs a class label in $\mathcal{Y}$. Ordering (reading order vs. ROI order) and exact formatting contracts are specified in Appendix E.

### 3.4. Visual Encoder and Document Decoder

**Visual encoder and multi-page token fusion.** Given a DNA document $D(S) = \left(\{\mathrm{page}_p\}_{p=0}^{P-1}, \{\mathcal{B}^{(p)}\}_{p=0}^{P-1}\right)$ rendered from a genomic sequence $S$, we transform its multipage images into a compact sequence of visual tokens for prompted decoding. We denote the entire vision-side pipeline as a single visual encoder $E_\theta$. Following DeepSeek-OCR (Wei et al., 2025), $E_\theta$ adopts the SAM–Conv–CLIP-L visual front-end, and additionally includes a learned projector $\Pi_\theta$ and a multi-page fusion module $\mathcal{F}_\theta$ for document-level aggregation.

Each $\mathrm{page}_p \in \mathbb{R}^{H \times W \times 3}$ is an RGB image with $H = W = 640$. We partition each page into non-overlapping $16 \times 16$ patches and encode them with the SAM–Conv–CLIP-L visual front-end. Let $V(\cdot)$ denote this visual front-end. The

*Table 1.* 6 prompt families (T1–T6) spanning OCR-style genomic primitives. See Appendix E for instruction variants and output formats.

| Task | Primitive | Prompt Family (Query) | Supervision / Output |
|------|-----------|----------------------|----------------------|
| T1 | Recognition (Reading) | Free-form DNA transcription (OCR) | Text tokens $\hat{Y}$ (sequence transcription) |
| T2 | Grounding + Recognition | Read DNA text and locate each line/block | Seq–box pairs $\langle \hat{Y}, \hat{b}_{ij} \rangle$ |
| T3 | Recognition (ROI) | ROI-based DNA transcription (boxes given) | Text tokens per ROI $\hat{Y}_{\mathrm{ROI}}$ |
| T4 | Completion (Masked) | Masked-region completion (boxes given) | Reconstructed text tokens $\hat{Y}_{\mathrm{mask}}$ |
| T5 | Retrieval + Grounding | Query-driven DNA subsequence localization | Matched boxes $\{\hat{b}\}$ for all occurrences |
| T6 | Recognition (Global) | Chromosome-level document classification | Class label $\hat{y} \in \mathcal{Y}$ (chromosome ID) |

Conv stage performs a fixed downsampling with ratio 16 along the token axis. We then apply a learned projector $\Pi_\theta$ to align the resulting features to the decoder width $d$:

$$\tilde{U} = \Pi_\theta \big( V(\{\mathrm{page}_p\}_{p=0}^{P-1}) \big) \in \mathbb{R}^{P \times T \times d}, \qquad T = \frac{T_0}{16}, \tag{6}$$

where $T_0$ is the token length before Conv downsampling. Thus, $\tilde{U}[p] \in \mathbb{R}^{T \times d}$ denotes the projected *per-page* token sequence for page $p$. Since a DNA sequence can span multiple pages, we aggregate $\{\tilde{U}[p]\}_{p=0}^{P-1}$ into a fixed-length document representation:

$$Z = E_\theta(\{\mathrm{page}_p\}_{p=0}^{P-1}) = \mathcal{F}_\theta(\tilde{U}) \in \mathbb{R}^{L \times d}, \tag{7}$$

where $L$ is the number of fused document tokens (we use $L = 100$). We instantiate $\mathcal{F}_\theta$ as a multi-head self-attention module with one attention layer and 20 heads, followed by a mean reduction over the page dimension, yielding a compact token sequence invariant to the number of pages.

**Document decoder and `<image>` injection.** We employ an autoregressive document decoder $G_\psi$ based on DeepSeek-3B (Mixture-of-Experts, 570M activated parameters) from DeepSeek-OCR for prompt-conditioned genomic reasoning. Each instance is represented as a two-turn multimodal conversation $\mathcal{C} = [m^{(u)}, m^{(a)}]$. The user message contains a single reserved `<image>` placeholder, whose `images` field carries an ordered list of $P$ page images $\{\mathrm{page}_p\}_{p=0}^{P-1}$. To explicitly specify the number of images bound to the placeholder, we append a meta line `\nNUM_IMAGES=P.` immediately after `<image>` in the prompt. Conditioned on the fused visual tokens $Z$ and a task prompt $q \in \mathcal{Q}$, the decoder autoregressively generates the task-dependent output sequence $\hat{Y} = (\hat{y}_1, \ldots, \hat{y}_T)$:

$$P_\psi(\hat{Y} \mid Z, q) = \prod_{t=1}^{T} P_\psi(\hat{y}_t \mid \hat{y}_{<t}, Z, q), \qquad \hat{y}_t \in \mathcal{V}, \tag{8}$$

where $\mathcal{V}$ denotes the decoder vocabulary.

### 3.5. Training Objectives and Optimization

**Prompt-conditioned generative pretraining.** We pretrain OpticalDNA with the unified OCR-style prompting protocol (Section 3.3). Given a prompt $q \in \mathcal{Q}$ and fused visual tokens $Z = E_\theta(\{\mathrm{page}_p\}_{p=0}^{P-1})$, we optimize the standard teacher-forced autoregressive objective on the assistant response tokens $y = (y_1, \ldots, y_T)$:

$$\mathcal{L}_{\mathrm{pt}} = -\sum_{t=1}^{T} \log P_\psi(y_t \mid y_{<t}, Z, q), \tag{9}$$

where the loss is computed only over the assistant span.

**Task balancing and instance randomization.** To balance the six prompt families and improve robustness, we sample the task index $t \sim \mathrm{Cat}(\boldsymbol{\pi})$ with $\boldsymbol{\pi} = (\pi_1, \ldots, \pi_6)$ for T1–T6. To improve robustness for ROI-centric supervision, we optionally apply tail truncation to tasks that rely on page-level region items (T2/T3/T5), by deleting a suffix of line-/box-level supervision items with a base rate $\rho_{\mathrm{base}}$ and an additional randomized rate capped by $\rho_{\mathrm{max}}$. For line-/block-based supervision, we randomize the span by sampling $n_{\mathrm{samp}} \sim \mathrm{Uniform}\{n_{\mathrm{samp}}^{\min}, \ldots, n_{\mathrm{samp}}^{\max}\}$, optionally sampling source lines without replacement (Unique lines). For subsequence localization (T5), we sample the query length $\ell \sim \mathrm{Uniform}\{\ell_{\min}, \ldots, \ell_{\max}\}$ and control whether overlapping matches are allowed.

**Optimization and trainable components.** We freeze the DeepSeek-OCR visual front-end (SAM–Conv–CLIP-L). The document decoder $G_\psi$ is fine-tuned with LoRA, the multi-page fusion module $\mathcal{F}_\theta$ is trained with full-parameter updating, and the projector $\Pi_\theta$ is tuned either with LoRA or full-parameter updating depending on the setting (See Section 4.1). This design enables efficient adaptation while preserving the stability of large pretrained backbones.

## 4. Experiments

### 4.1. Experimental Settings

**Pre-training.** All pretraining experiments are conducted on $8\times$H100 GPUs. For fair comparison with prior work (Nguyen et al., 2023a; Duan et al., 2025), we pretrain OpticalDNA on HG38 (Schneider et al., 2017) using the same sequence sampling protocol as JanusDNA (Duan et al., 2025).

*Table 2.* AUROC comparison on eQTL tasks from DNALONGBench. The best and second-best results are highlighted in **bold** and underlined, respectively. The expert model refers to Enformer. * denotes results reproduced with frozen backbones and a linear probe.

| Models #Trainable param | Expert Model (252M) | HyenaDNA (1.6M) | Caduceus-Ph (7.7M) | NT-v2-500M* (1.03K) | GENERator-1.2B* (10.24K) | JanusDNA w/o mid-Attn (7.662M) | JanusDNA mlp w/o mid-Attn (7.745M) | OpticalDNA Linear Probing (256K) | OpticalDNA MLP (1.3M–2.3M) |
|---|---|---|---|---|---|---|---|---|---|
| AT | 0.741 | 0.479 | 0.690 | 0.764 | 0.795 | 0.803 | **0.852** | **0.852** | **0.852** |
| AS | 0.736 | 0.513 | 0.759 | 0.730 | 0.730 | 0.741 | 0.769 | **0.813** | 0.788 |
| CCF | 0.639 | 0.584 | 0.690 | 0.744 | 0.703 | 0.771 | 0.802 | 0.812 | **0.819** |
| MS | 0.621 | 0.487 | 0.789 | 0.828 | 0.819 | 0.803 | 0.864 | **0.880** | 0.877 |
| NT | 0.683 | 0.511 | 0.842 | 0.824 | 0.828 | 0.877 | **0.914** | 0.884 | 0.900 |
| SNSES | 0.710 | 0.471 | 0.812 | 0.833 | 0.854 | 0.875 | 0.903 | 0.904 | **0.931** |
| SSELL | 0.700 | 0.544 | 0.692 | 0.749 | 0.726 | 0.706 | **0.846** | 0.835 | 0.832 |
| Thyroid | 0.612 | 0.529 | 0.703 | 0.706 | 0.749 | 0.752 | 0.793 | 0.791 | **0.876** |
| WB | 0.689 | 0.512 | 0.769 | 0.773 | 0.834 | 0.794 | 0.821 | 0.895 | **0.927** |
| Average | 0.681 | 0.514 | 0.750 | 0.772 | 0.782 | 0.791 | 0.840 | 0.852 | **0.867** |

We extract 2048-bp windows and render each window into a DNA document. We use a two-stage adaptation schedule: Stage 1 trains $\mathcal{F}_\theta$ with full updates and tunes $G_\psi$ with LoRA for 227,600 steps (about 8 days), and Stage 2 further tunes the projector $\Pi_\theta$ for 190,000 steps (about 3 days) while keeping $\mathcal{F}_\theta$ fixed (Appendix F.1). We also pretrain on rice using the Nipponbare (*O. sativa japonica*) T2T-NIP genome from RiceSuperPIRdb (Shang et al., 2022). We scan the genome with a 2048-bp window and 1920-bp overlap, and train for 150,000 steps (about 5 days; Appendix F.2).

**Downstream Tasks and implementation details.** To comprehensively evaluate genotype-to-phenotype modeling under long genomic contexts, we evaluate OpticalDNA on three long-sequence DNA benchmarks: DNALONG-BENCH (Appendix G.1), Rice Subspecies Benchmark (RiceSubBench; Appendix G.2), and Rice Whole-Genome Phenotype Benchmark (RiceWGPB; Appendix G.3), together forming a progressive evaluation from canonical regulatory modeling, to robustness under population-level genetic variation, and finally to genome-wide phenotype prediction (see Appendix G.4). We use the HG38-pretrained OpticalDNA to evaluate DNALONGBENCH, and the rice-pretrained OpticalDNA to evaluate RiceSubBench and RiceWGPB. For DNALONGBENCH, we follow the same experimental setup as JanusDNA (Duan et al., 2025), using identical data splits and reporting AUROC; for RiceSub-Bench we adopt a random split and report accuracy and AUROC, while for RiceWGPB we report AUROC. For efficiency, we extract per-page visual features, mean-pool across pages, and flatten token features to form a fixed-dimensional embedding. We evaluate two lightweight probing heads: (i) linear probing with a single fully-connected layer, and (ii) a shallow MLP with a linear bottleneck layer (hidden size 8 or 16), an activation (GELU or identity), a dropout layer (rate 0.1), and a final linear classifier. We report test performance corresponding to the best validation score and the hyperparameter search space is provided in Appendix H.1. Furthermore, strict contamination analyses show negligible train–test leakage, with full-sequence over-lap rates of 0.279% for eQTL, 0.489% for RiceSubBench, and similarly low rates for RiceWGPB; see Appendix I.

## 4.2. Main Results

### 4.2.1. DNA LONG RANGE BENCHMARK

We evaluate OpticalDNA on the eQTL prediction tasks from DNALONGBENCH and follow JanusDNA with identical data splits, reporting AUROC (Table 2). With a 256K-parameter linear probe, OpticalDNA achieves state-of-the-art performance (0.852 average AUROC across nine GTEx tissues), improving over HyenaDNA (0.514, +65.8% relative) (Nguyen et al., 2023a), Caduceus-Ph (0.750, +13.6%) (Schiff et al., 2024), JanusDNA (w/o mid-Attn, 0.791, +7.7%) (Duan et al., 2025), NT-v2-500M (0.772, +10.4%) (Dalla-Torre et al., 2025), and GENERator-1.2B (0.782, +9.0%) (Wu et al., 2025). For NT-v2-500M and GENERator-1.2B, which do not natively support 450 kb inputs, we evaluate them using chunked sequence representations with frozen backbones and linear probes. OpticalDNA outperforms JanusDNA on all tissues while using ∼30× fewer trainable parameters for adaptation (256K vs. ∼7.7M), and exceeds the expert model (Avsec et al., 2021) that activates 252M parameters at inference (0.681 average AUROC) with up to 985× fewer *activated* parameters. Replacing the linear probe with a lightweight MLP head (1.3M–2.3M parameters) further improves the average AUROC from 0.852 to 0.867 (+1.8% relative) and achieves the best performance on 5/9 tissues; in particular, relative to the previous state-of-the-art JanusDNA MLP (w/o mid-Attn), OpticalDNA yields clear gains on WB (0.927 vs. 0.821) and Thyroid (0.876 vs. 0.793). Overall, these results show that OpticalDNA preserves long-range regulatory signals under ultra-long genomic contexts with an accuracy–efficiency trade-off via parameter- and token-efficient adaptation.

### 4.2.2. RICE SUBSPECIES BENCHMARK

We evaluate OpticalDNA on RiceSubBench, which measures cross-subspecies generalization from in-domain *japon-*

*Table 3.* Generalization performance on RiceSubBench from in-domain to far-OOD evaluation (Accuracy / AUROC).

| Model | In-Domain japonica | Near-OOD aus | Mid-OOD rufipogon | Far-OOD barthii | glaberrima |
|---|---|---|---|---|---|
| Evo-2 (7B) | 0.486 / 0.700 | 0.509 / 0.714 | 0.500 / 0.714 | 0.532 / 0.725 | 0.489 / 0.705 |
| LucaOne (1.8B) | 0.510 / 0.703 | 0.551 / 0.723 | 0.589 / 0.760 | 0.556 / 0.745 | 0.526 / **0.736** |
| OpticalDNA (409M) | **0.590 / 0.739** | **0.556 / 0.725** | **0.639 / 0.762** | **0.608 / 0.747** | **0.599** / 0.731 |

*ica* to Near-/Mid-/Far out-of-distribution (OOD) subspecies, reporting accuracy and AUROC (Table 3). We compare against two highly competitive foundation-model baselines: Evo-2 (7B) (Brixi et al., 2025), trained on OpenGenome2 with 8.8 trillion DNA base pairs spanning all domains of life, and LucaOne (1.8B) (He et al., 2025), pretrained on nucleic acid and protein sequences from 169,861 species; reproduction and hyperparameter details are provided in Appendix H.2. Despite using substantially fewer parameters for representation extraction (409M vs. 7B/1.8B), OpticalDNA achieves the best accuracy on all splits, including in-domain *japonica* (0.590) and all OOD settings. The advantage becomes more pronounced under stronger distribution shift, where OpticalDNA improves over LucaOne by +8.49% on *rufipogon* (0.639 vs. 0.589), +9.35% on *barthii* (0.608 vs. 0.556), and +13.88% on *glaberrima* (0.599 vs. 0.526), and consistently outperform Evo-2 across all OOD groups. In terms of AUROC, OpticalDNA achieves the best results on 4/5 subspecies splits, indicating robustness under cross-subspecies distribution shift. Overall, OpticalDNA achieves strong OOD transfer across rice subspecies, while being far smaller than multi-species foundation baselines.

### 4.2.3. RICE WHOLE-GENOME PHENOTYPE

We further evaluate OpticalDNA on *rice whole-genome phenotype benchmark* (RiceWGPB), using a ∼400M-base rice genome as input to predict organism-level traits (thousand-grain weight, TGW; leaf rolling index, LRI). As shown in Table 4, OpticalDNA achieves the lowest RMSE on both traits, outperforming Evo-2 and LucaOne with substantially fewer parameters (409M vs. 7B/1.8B). Crucially, OpticalDNA is more practical at the genome scale: on a representative 389.8M-base genome (NH158), it reduces inference time from hours (Evo-2) and tens of minutes (LucaOne) to 12.3 minutes. These results highlight OpticalDNA as an effective and efficient solution for genome-wide trait prediction under ultra-long inputs (see Appendix G.3 for evaluation details).

*Table 4.* RMSE and inference time on RiceWGPB under ∼400M-base inputs.

| Model | TGW (g) ↓ | LRI-15SZ (%) ↓ | Time ↓ |
|---|---|---|---|
| Evo-2 (7B) | 3.056 | 9.617 | 5h40m |
| LucaOne (1.8B) | 8.817 | 9.740 | 32.5m |
| OpticalDNA (409M) | **2.952** | **9.531** | **12.3m** |

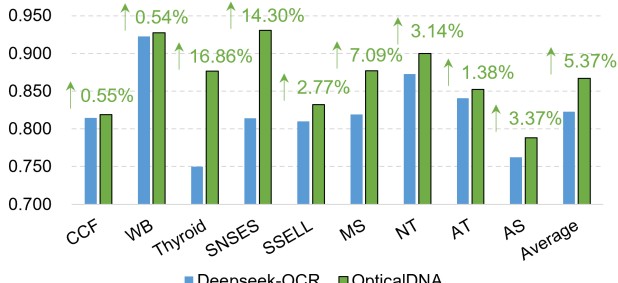

*Figure 3.* Ablation on DNALONGBENCH (AUROC). Green arrows and numbers denote the relative gain over DeepSeek-OCR.

### 4.3. Ablation Study

Built on DeepSeek-OCR (Wei et al., 2025), OpticalDNA is compared with its backbone in ablation studies under the same evaluation protocol. We examine two questions: **Q1** whether OpticalDNA improves downstream DNA prediction, and **Q2** whether it enhances DNA image transcription.

**Q1: Downstream DNA prediction.** Fig. 3 reports eQTL AUROC on nine GTEx tissues. OpticalDNA improves DeepSeek-OCR on *all* tissues, yielding a +5.37% relative gain on the overall average. The gains are especially pronounced on more challenging tissues, such as Thyroid (+16.86%) and SNSES (+14.30%), and remain consistent on MS (+7.09%), AS (+3.37%), NT (+3.14%), and SSELL (+2.77%). These results indicate that our DNA-specific document formulation and pretraining yield transferable representations for long-range functional genomics prediction. Detailed per-tissue results are provided in Appendix J.1.

**Q2: DNA transcription.** We randomly sample 1,000 validation examples from both HG38 and rice pretraining corpora to evaluate single-page image-to-text DNA transcription. To assess robustness, we randomly truncate each DNA document by removing up to 90% of its tail and transcribe the remaining single-page input into nucleotide sequences. Since the target length is known, we compare predictions against different prefixes of the ground-truth (GT) sequence and report Exact Match (EM) and character similarity (CS) (Table 5). Across all prefix ratios and both corpora, OpticalDNA substantially outperforms DeepSeek-OCR, achieving near-perfect EM under short prefixes (e.g., 97.3/98.5 at 10%) and remaining strong at full-length transcription

(79.6/74.9 at 100%), while DeepSeek-OCR stays near zero EM throughout. OpticalDNA also maintains consistently high CS (90.6–100.0), indicating stable nucleotide-level fidelity under severe truncation and varying decoding lengths. Furthermore, length-wise OCR transcription results show that OpticalDNA-Rice remains reliable under moderate sequence density and degrades mainly near the single-page capacity (Appendix K.1).

*Table 5.* EM and CS performance of DeepSeek-OCR (DS) and OpticalDNA (Opt) over GT prefix ratios on HG38 and rice (1,000 each) under random tail truncation (up to 90%).

| GT Prefix (%) | HG38 (1,000) | | Rice (1,000) | |
|---|---|---|---|---|
| | EM (%) ↑ DS/Opt | CS (%) ↑ DS/Opt | EM (%) ↑ DS/Opt | CS (%) ↑ DS/Opt |
| 10 | 8.6/97.3 | 84.6/100.0 | 7.4/98.5 | 83.1/100.0 |
| 20 | 4.5/95.8 | 46.9/99.3 | 3.7/96.7 | 46.0/99.5 |
| 30 | 2.8/93.4 | 31.4/97.9 | 2.5/94.3 | 30.4/98.0 |
| 40 | 2.4/91.3 | 23.9/96.9 | 2.2/92.0 | 23.2/97.0 |
| 50 | 2.1/89.8 | 18.6/95.7 | 2.0/89.7 | 17.9/96.0 |
| 60 | 1.7/88.6 | 15.6/94.6 | 1.7/87.4 | 14.9/95.0 |
| 70 | 1.6/87.0 | 13.3/93.9 | 1.5/85.0 | 12.5/94.1 |
| 80 | 1.6/85.0 | 11.2/92.7 | 1.2/82.4 | 10.3/93.0 |
| 90 | 1.3/82.5 | 10.0/91.8 | 1.2/78.1 | 9.1/91.6 |
| 100 | 1.2/79.6 | 8.8/90.7 | 1.2/74.9 | 7.9/90.6 |

### 4.4. ROI Identification Capability Evaluation

We evaluate ROI identification on T2 by sampling 1,000 validation examples from HG38 and rice pretraining corpora. Table 6 shows that OpticalDNA achieves strong line-level grounding under the strict IoU= 0.99 criterion: OpticalDNA-HG38 attains higher joint accuracy (0.9820) and page-level strict success (0.8763), while OpticalDNA-Rice exhibits near-perfect line-count consistency (0.9991). Both variants achieve near pixel-level localization error ($\ell_\infty \leq 0.3941$ px). Further metric and evaluation details are provided in Appendix K.2, with additional evaluation details for other tasks in Appendix K.3 (T3), Appendix K.4 (T5), Appendix K.5 (T4), and Appendix K.6 (T6).

### 4.5. Interpretability Analysis

We probe model interpretability with Grad-CAM on the *multi-page fusion* representations (see Appendix K.7 for implementation details). Fig. 4 visualizes two pages from a donor-labeled Japonica test document. Unlike sequence-based models that distribute attention diffusely at the base level, OpticalDNA produces localized attributions over short contiguous regions, reflecting region-level processing induced by visual tokenization and local receptive fields. This phrase-like reading behavior mirrors how humans process long text and is particularly well suited for long DNA documents, as it avoids exhaustive base-wise scanning while preserving informative local patterns under a fixed token

*Table 6.* ROI identification on T2 (IoU= 0.99).

| Model | LCM ↑ | Joint ↑ | Strict ↑ | $\ell_\infty \downarrow$ |
|---|---|---|---|---|
| OpticalDNA-HG38 | 0.9871 | 0.9820 | 0.8763 | 0.3941 |
| OpticalDNA-Rice | 0.9991 | 0.9692 | 0.8084 | 0.3243 |

**Abbrev.** LCM: line-count match; Joint: line-level joint accuracy (text+ROI); Strict: page-level strict success.

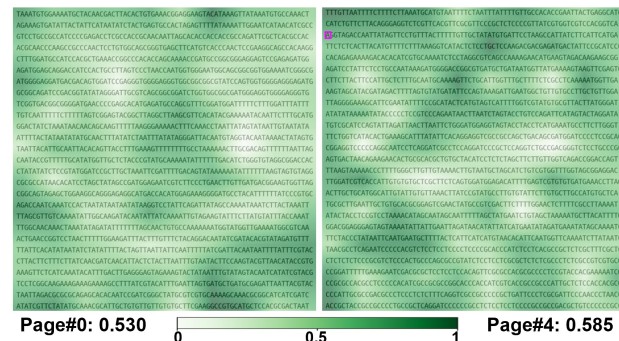

*Figure 4.* Grad-CAM visualization on *multi-page fusion* for a donor case (two pages). Purple boxes indicate donor splice sites; numbers denote page-level mean attribution.

budget. Importantly, Grad-CAM activations concentrate around the annotated donor splice site (purple box), and the corresponding page receives higher overall attribution, indicating that OpticalDNA can both localize biologically meaningful splice signals and prioritize the most relevant pages during inference. Overall, these results indicate that OpticalDNA learns structured, region-aware representations aligned with genomic reasoning, while also offering practical interpretability, highlighting its potential utility for large-scale genomic and transcriptomic analyses.

### 4.6. Visual Compression Analysis

We evaluate how the visual compression of OpticalDNA affects downstream performance by varying the rendering resolution. Increasing the resolution allows each page to cover a longer DNA span; since the patch size is fixed, the number of visual tokens per page also increases, leading to a higher overall compression ratio. Table 7 shows that as the resolution increases from 512 to 1280, the number of visual tokens per page rises from 64 to 400 and the compression ratio increases from 19.0 to 21.2, while the average AUROC remains nearly unchanged (0.849–0.852). This suggests that OpticalDNA supports stronger visual compression with minimal performance variation under long-context settings. Token bottleneck results further support this robustness under direct visual-token reduction (Appendix L), and an end-to-end computational cost comparison with representative genomic foundation models is provided in Appendix M.

*Table 7.* Average AUROC under different rendering resolutions on nine eQTL tasks. #Page and #Token denote the number of rendered pages and visual tokens per page, respectively.

| Resolution | #Page | #Token | Compression ratio | AUROC |
|---|---|---|---|---|
| 512 | 370 | 64 | 19.0 | 0.849 |
| 640 | 226 | 100 | 19.9 | **0.852** |
| 1024 | 84 | 256 | 20.9 | 0.851 |
| 1280 | 53 | 400 | 21.2 | 0.849 |

### 4.7. Rendering Robustness

We evaluate rendering sensitivity on the Adipose Subcutaneous eQTL task by varying text density, aspect ratio, layout mode, starting position, and font (Appendix N.1 for detailed configurations). As shown in Fig. 5, OpticalDNA remains robust under common formatting changes, with several variants matching or outperforming the original layout (0.813 AUROC), such as dense rendering, wide+dense rendering, and font variations. The main degradation appears in rigid fixed-window layouts, suggesting that preserving local sequence continuity is more favorable than imposing artificial fragment boundaries. Detailed rendering ablations and formatting-robustness analyses are provided in Appendix N.

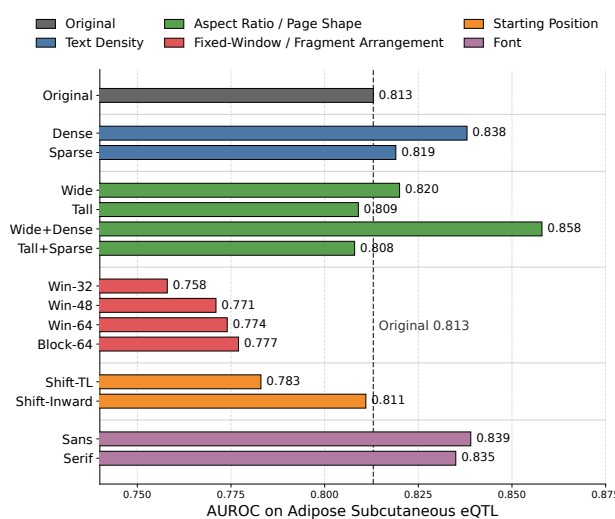

*Figure 5.* Rendering robustness on the Adipose Subcutaneous eQTL task. Bars are grouped by perturbation type (Appendix N for details) and the dashed line marks the original layout performance.

We further evaluate low-resolution robustness by downsampling the rendered $640 \times 640$ DNA images while keeping the DNA content and layout fixed. Fig. 6 shows that OpticalDNA maintains stable AUROC under moderate downsampling, with noticeable degradation mainly at extreme resolutions. Detailed results are provided in Appendix N.2.

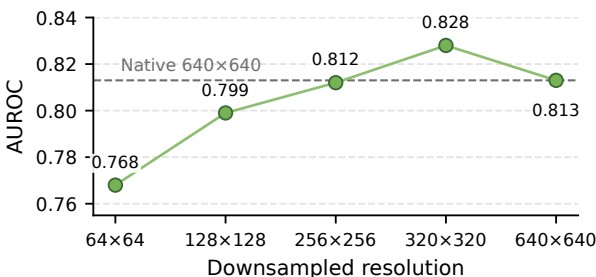

*Figure 6.* Low-resolution robustness on the Adipose Subcutaneous eQTL task. AUROC remains stable under moderate downsampling and degrades mainly at extreme resolutions.

### 4.8. Cross-Species Transfer

We evaluate whether OpticalDNA can transfer from human to rice by applying the HG38-pretrained model to RiceSub-Bench. As shown in Fig. 7, OpticalDNA-HG38 clearly outperforms HG38-pretrained Caduceus and JanusDNA, and remains close to rice-pretrained OpticalDNA. This cross-species transfer result suggests that OpticalDNA's gain is not simply due to domain-matched rice pretraining, but also reflects the transferability of the OCR-style visual formulation. Detailed per-dataset results and additional RiceWGPB controls are provided in Appendix O.

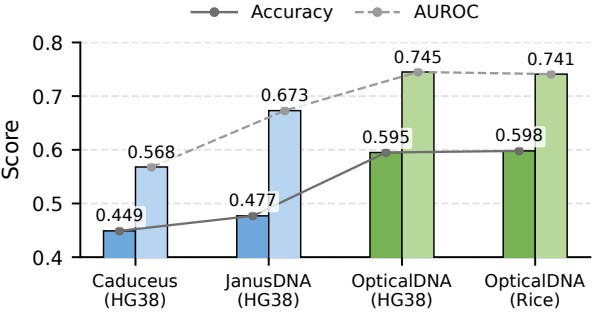

*Figure 7.* AUROC of AS eQTL under different image resolutions.

## 5. Conclusion

We propose OpticalDNA, a vision-based framework that reformulates genomic modeling as OCR-style document understanding, offering an alternative to sequence-token-centric learning. By rendering DNA into structured visual layouts, learning compact and reconstructible visual tokens, and training with prompt-conditioned objectives over core genomic primitives, OpticalDNA enables layout-aware representations that better capture sparse and discontinuous genomic semantics. Experiments show that this document-style formulation achieves strong performance with improved computational efficiency, providing a scalable direction for long-context genomic representation learning.

## Acknowledgements

This work was supported by the Yuelushan Laboratory Breeding Program (grant nos. YLS-2026-ZY01001, YLS-2026-ZY01002, YLS-2026-ZY01003), the Fundamental and Interdisciplinary Disciplines Breakthrough Plan of the Ministry of Education of China (JYB2025XDXM602), the National Natural Science Foundation of China (grant nos. U22A037, 62425204, 62122025, 62450002, 62432011, 62472165), the China Postdoctoral Science Foundation (grant no. 2025M771569), the YueLuShan Center Industrial Innovation (grant no. 2025YCII0125).

## Impact Statement

This paper presents OpticalDNA, a vision-based reformulation of genomic foundation modeling for long-context biological sequences. By rendering DNA as structured documents and learning prompt-conditioned objectives for reading, region grounding, retrieval, and span completion, OpticalDNA improves the accuracy–efficiency trade-off for genome-scale modeling and enables efficient reasoning over ultra-long genomic contexts. Its potential benefits include lowering computational barriers for whole-genome analysis and supporting applications such as functional element discovery, variant localization, and genome-wide phenotype prediction, while also offering a document-understanding perspective for interpretable and structure-aware biological sequence modeling. Potential risks arise from the sensitive nature of genomic data: stronger genome-scale models could be misused for privacy-invasive inference or could amplify biases when trained or evaluated on unevenly distributed data across species, ancestries, or genomic contexts. Although this work is methodological and does not introduce new human-identifiable data, responsible deployment should follow established genomic data governance practices, including access control, de-identification, evaluation under distribution shift, and careful consideration of privacy, consent, and fairness.

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

# A. Comparison Details Between 1D and 2D Genomic Modeling

## A.1. Comparison Details Between 1D CNN and 2D CNN

This section presents a controlled accuracy–efficiency comparison between (i) 1D CNNs that operate on DNA as a one-dimensional token sequence and (ii) 2D CNNs that operate on our *DNA document* representation, where each DNA sequence is rendered into $n$ pages of $640 \times 640$ images (with $n$ determined by sequence length). The goal is to quantify whether treating DNA as a document (vision-style processing) yields a better practical trade-off for long-range functional genomics prediction.

**Compared models.** We compare 1D CNN baselines and 2D CNN backbones on the eQTL prediction benchmark across nine tissues. For 1D CNNs, we start from the SimpleCNN baseline adopted in DNALONGBENCH (Cheng et al., 2025b). Empirically, this baseline provides limited performance on our eQTL setup (Table S2), motivating us to build upon its overall design and optimize the architecture to obtain a stronger and scalable family of 1D CNNs. Concretely, we build a family of 1D CNN variants (CNN1D-small/base/large/xl) by increasing depth and width under a consistent macro design; the preset specifications are summarized in Table S1. All CNN1D variants use unit dilations and no max pooling within convolutional blocks, with dropout mildly increasing with model size. For 2D CNNs, we select lightweight TIMM backbones with comparable parameter budgets, including MobileNetV3-S ($0.50\times/1.00\times$), TinyNet (D/E), and ResNet-10t.

**Experimental setup.** To ensure a controlled comparison, all models are trained with the same optimization protocol (EPOCHS$= 10$, LR$= 5 \times 10^{-3}$, batch size$= 4$). We report the average AUROC across nine tissues, model size, and two practical efficiency metrics: wall-clock time per epoch (*Epoch Time*) and throughput (*Throughput*). Here, *Throughput* is measured in *samples/s* and is therefore independent of whether a sample is represented as a token sequence or as rendered images. For stable estimation, *Epoch Time* and *Throughput* are measured at the 5th training epoch (per tissue) and then averaged over all nine tissues. All experiments are conducted on a single NVIDIA RTX 4090 GPU with an AMD EPYC 7402 24-Core CPU and 503 GB RAM.

**Results and discussion.** Table S2 shows that 2D CNN backbones consistently achieve a better accuracy–efficiency trade-off than 1D CNN baselines on eQTL prediction. On accuracy, lightweight 2D models such as TinyNet-E achieve the best average AUROC (0.786), exceeding all 1D CNN variants in our comparison (CNN1D-small/base/large/xl). Meanwhile, the original SimpleCNN baseline is markedly weaker (0.528), and even our optimized 1D CNN family, while substantially improving over SimpleCNN, still falls short of the best 2D backbones. These gains do not come at the cost of practicality: 2D CNNs are also faster in wall-clock training. For example, TinyNet-E reduces the per-epoch training time to 66.9 s (measured at epoch 5), compared to 386.6–636.8 s for the 1D CNN variants. Its throughput is also competitive (33.7 samples/s), indicating that the improved wall-clock time is not an artifact of redefining the unit of work. Overall, these results suggest that, under similar or smaller parameter budgets, 2D document-style processing yields both stronger predictive performance and superior practical efficiency, supporting the necessity of our 2D DNA document representation for long-range functional genomics prediction.

*Table S1.* Preset specifications of 1D CNN baselines used in Table S2. Each model is defined by stage-wise channels $C$, kernel sizes $K$, and strides $S$. "Block" indicates standard convolutional (Conv) or residual (Res) 1D blocks.

| Variant | $C$ (channels) | $K$ (kernels) | $S$ (strides) | Block / Dropout |
|---|---|---|---|---|
| SimpleCNN | (128, 64, 32) | (3, 3, 3) | (1, 1, 1) | Conv / 0.00 |
| small | (32, 64, 128, 128) | (7, 5, 3, 3) | (2, 2, 2, 1) | Conv / 0.05 |
| base | (64, 128, 256, 256) | (11, 7, 5, 3) | (2, 2, 2, 1) | Res / 0.10 |
| large | (96, 192, 384, 384, 384) | (11, 9, 7, 5, 3) | (2, 2, 2, 1, 1) | Res / 0.10 |
| xl | (112, 224, 448, 448, 448, 448) | (15, 11, 9, 7, 5, 3) | (2, 2, 2, 1, 1, 1) | Res / 0.15 |

## A.2. Comparison with Stronger 1D Sequence Modeling Baselines

Beyond 1D CNNs, we further evaluate several stronger 1D sequence modeling architectures that are commonly used to capture long-range dependencies, including recurrent models, linear attention, and sparse attention. Specifically, we include BiGRU as an RNN-based baseline, Performer as a linear-attention baseline, and Longformer as a sparse-attention baseline. Following Appendix A.1, all models are trained and evaluated under the same 9-tissue GTEx eQTL setting.

*Table S2.* Accuracy–efficiency comparison on eQTL prediction. We report the average AUROC across nine tissues, model size, and training efficiency. For stability, *Epoch Time* and *Throughput* are measured at the 5th training epoch and averaged over the nine tissues. The SimpleCNN AUROC is taken from DNALONGBENCH (Cheng et al., 2025b).

| Model | Average AUROC | #Params (M) ↓ | Epoch Time (s) ↓ | Throughput (samples/s) ↑ |
|---|---|---|---|---|
| SimpleCNN | 0.528 | 0.033 | - | - |
| CNN1D-small | 0.649 | 0.086 | 395.6 | 95.1 |
| CNN1D-base | 0.643 | 1.150 | 451.0 | 29.0 |
| CNN1D-large | 0.553 | 4.611 | 539.7 | 13.3 |
| CNN1D-xl | 0.578 | 9.890 | 636.8 | 8.5 |
| MobileNetV3-S (0.50×) | 0.760 | 0.570 | 138.3 | 16.1 |
| TinyNet-E | 0.786 | 0.764 | 66.9 | 33.7 |
| TinyNet-D | 0.755 | 1.060 | 143.5 | 15.7 |
| MobileNetV3-S (1.00×) | 0.749 | 1.520 | 170.8 | 13.2 |
| ResNet-10t | 0.755 | 4.924 | 175.2 | 12.8 |

The results are shown in Table S3. Among these 1D baselines, the RNN model performs the best, achieving an average AUROC of 0.6575. The linear-attention (LinearAttn) and sparse-attention (SparseAttn) baselines achieve 0.6279 and 0.6163 AUROC, respectively. However, all of them remain substantially below the 2D baselines, including TinyNet-E and OpticalDNA. In addition, sparse attention incurs a significantly higher computational cost without improving predictive performance.

These results suggest that the advantage of 2D genomic modeling is not merely due to the weakness of localized 1D CNNs. Even when stronger 1D sequence models are used, the document-style 2D representation provides a more favorable accuracy–efficiency trade-off for sparse genomic signals.

*Table S3.* Comparison with stronger 1D sequence modeling baselines on the 9-tissue GTEx eQTL benchmark.

| Model | Average AUROC | #Params (M) ↓ | Epoch Time (s) ↓ | Throughput (samples/s) ↑ |
|---|---|---|---|---|
| RNN | 0.6575 | 0.73 | 27.27 | 82.92 |
| LinearAttn | 0.6279 | 3.29 | 25.42 | 88.84 |
| SparseAttn | 0.6163 | 3.29 | 95.48 | 23.69 |

### A.3. Comparison with 1D Transformer Baselines

We further compare against 1D Transformer baselines, which natively model global interactions and therefore address the concern that the improvement of 2D models may simply result from the limited receptive field of 1D CNNs. Following Appendix A.1, we evaluate Transformer-tiny and Transformer-small under the same 9-tissue eQTL setting, and compare them with representative 2D baselines.

As shown in Table S4, Transformer-tiny achieves 0.475 AUROC with 0.406M parameters, while Transformer-small achieves 0.575 AUROC with 1.793M parameters. Both are substantially below MobileNetV3-S and TinyNet-E, which achieve 0.760 and 0.786 AUROC, respectively. Moreover, the 1D Transformer baselines are significantly less efficient in terms of epoch time and throughput.

These results further rule out CNN locality as the primary explanation for the performance gap. Even with global 1D modeling, the 1D Transformer baselines remain substantially weaker than the 2D models. This supports our hypothesis that the performance gain is closely related to the document-style 2D representation and its computation pattern, which better align with sparse and discontinuous genomic signals.

### A.4. Controlled Comparison with Matched Token Budget and Shared Backbone

The above comparisons evaluate a broad range of 1D and 2D architectures. To further disentangle representation effects from backbone and compression advantages, we conduct a controlled experiment in which the token budget and downstream backbone are matched. The experimental settings follow Appendix A.1.

Specifically, the same genomic input is represented either as a 1D sequence or as a 2D image document. For the 2D setting,

*Table S4.* Comparison with 1D Transformer baselines on the 9-tissue GTEx eQTL benchmark.

| Model | Average AUROC | #Params (M) ↓ | Epoch Time (s) ↓ | Throughput (samples/s) ↑ |
|---|---|---|---|---|
| Transformer-tiny | 0.475 | 0.406 | 1150.2 | 1.363 |
| Transformer-small | 0.575 | 1.793 | 2980.8 | 0.47 |
| MobileNetV3-S (0.50×) | 0.760 | 0.570 | 138.3 | 16.1 |
| TinyNet-E | 0.786 | 0.764 | 66.9 | 33.7 |

the input is rendered as a $336 \times 336$ genomic image document. Both representations are tokenized and reduced to the same number of tokens, i.e., 10,000 tokens, and then processed by the same Transformer backbone under the same training and evaluation pipeline. Under this setup, the downstream model architecture and token count are fixed, making the input representation the primary remaining difference.

Table S5 shows the average AUROC on the 9-tissue eQTL benchmark. The 2D image-document representation achieves 0.626 AUROC, compared with 0.558 AUROC for the 1D sequence representation. This corresponds to a 6.8% absolute improvement, or a 12.20% relative improvement.

This controlled comparison isolates the representation effect and shows that the gain does not come from token compression or backbone differences. Under the same token budget and Transformer backbone, the document-style 2D formulation better represents sparse and discontinuous genomic signals, leading to stronger long-context prediction performance.

*Table S5.* Average AUROC on the 9-tissue eQTL benchmark under matched token count and shared Transformer backbone.

| Representation | # Tokens | Average AUROC |
|---|---|---|
| 1D Sequence | 10,000 | 0.558 |
| 2D Image Document ($336 \times 336$) | 10,000 | 0.626 |
| $\Delta$ | – | ↑ 12.20% |

## B. Motivation: Why Reformulate DNA as Visual Documents?

DNA is inherently a one-dimensional sequence, and sequence models are a natural and powerful starting point for genomic modeling. OpticalDNA does not aim to reject this view or claim that DNA is biologically a two-dimensional object. Instead, our motivation is to address a specific limitation of one-dimensional sequence processing in long-context genomic modeling: the accuracy–efficiency bottleneck caused by exhaustive sequential computation.

This bottleneck becomes especially pronounced when genomic inputs extend from hundreds of kilobases to whole-genome scale. In such regimes, the central challenge is not merely how to represent nucleotides, but how to allocate computation efficiently over extremely long sequences. Genomic signals are often sparse, discontinuous, and region-centric: functional elements, regulatory variants, splice sites, and phenotype-associated loci occupy relatively small regions embedded in long stretches of low-information background. A sequential model, however, typically allocates computation across the entire token stream, regardless of whether a local region is informative for the downstream task. As sequence length increases, this uniform allocation leads to unfavorable scaling in both memory and runtime.

OpticalDNA addresses this issue by reformulating the one-dimensional sequence as a structured visual document. Importantly, this rendering is intentionally simple and does not introduce explicit biological priors. It does not encode genomic landmarks, strand-aware structure, chromatin contacts, regulatory annotations, or biologically grounded spatial layouts. Rather, the rendering defines a lossless, bijective mapping between the original nucleotide sequence and its visual document representation. Each nucleotide remains recoverable through its corresponding position in the rendered document, and the original sequence order is preserved through the deterministic rasterization process.

Therefore, the purpose of the two-dimensional layout is not to claim a better biological geometry for DNA. Instead, it serves as a computational interface. By converting a long sequence into a coordinate-indexed document, OpticalDNA enables region-level access patterns similar to document understanding and OCR: the model can process local visual regions, aggregate page-level information, and perform prompt-conditioned operations such as reading, grounding, retrieval, and completion. In this sense, "structure" refers to computational structure—how information is accessed, compressed, and selectively processed—rather than biological structure.

This distinction is important. The key hypothesis of OpticalDNA is not that two-dimensional rasterization preserves stronger biological inductive biases than sequence models. Rather, the hypothesis is that a document-style representation can reorganize computation in a way that better matches the sparse and discontinuous nature of long genomic signals. Under this view, performance gains arise from the combination of lossless sequence preservation, region-aware visual tokenization, and selective computation over long contexts. The model can allocate capacity to informative regions while suppressing extensive background, producing an accuracy–efficiency trade-off that is difficult to achieve with exhaustive sequential reading.

Empirically, this motivation is supported by controlled comparisons between one-dimensional and two-dimensional processing under long-context settings (see Appendix A). On 450 kb eQTL prediction tasks, 2D backbones achieve a better accuracy–efficiency trade-off than 1D variants under comparable settings. Additional controlled comparisons with 1D Transformer baselines further indicate that the improvement is not solely attributable to model scale or downstream head design, but is consistent with the benefit of reorganizing computation under sparse long-range genomic signals. At whole-genome scale, this distinction becomes even more consequential: sequence models that support very long contexts incur high inference cost, while models that split the genome into shorter chunks may lose long-range information. OpticalDNA provides an alternative by preserving the full sequence through a bijective visual representation while enabling compact visual tokens and region-level aggregation.

In summary, OpticalDNA should be understood as a computational reformulation of genomic sequence modeling. DNA remains a one-dimensional biological sequence, but long-context genomic prediction requires efficient mechanisms for accessing sparse and discontinuous information. The proposed visual document representation preserves the original sequence while reorganizing computation into a layout-aware, region-centric form. Thus, the contribution is not a biologically grounded 2D geometry of DNA, but a lossless document-style reformulation that enables more efficient and effective long-context genomic modeling.

## C. Design Principles of the DNA Task Suite

**Overall design goal.**   The DNA task suite is designed to systematically evaluate how vision–language models understand, reason about, and manipulate DNA sequences when they are presented in visual form. Rather than focusing on a single capability, the tasks are constructed to cover a range of practical DNA analysis operations, including reading sequences, locating regions, completing missing content, searching for subsequences, and performing sequence-level classification.

All tasks share a unified visual input representation, in which DNA sequences are rendered as images. This ensures that differences in performance across tasks can be attributed to differences in reasoning requirements, rather than differences in input modality.

**Separating transcription, localization, and reasoning.**   A key principle in the task design is the explicit separation of three fundamental abilities: (i) nucleotide transcription, (ii) spatial localization, and (iii) higher-level reasoning.

Task T1 evaluates pure transcription ability by requiring the model to read DNA sequences without any spatial grounding. Task T2 adds spatial localization, requiring the model to associate transcribed sequences with their positions in the image. Tasks T3 and T4 further remove region detection from the problem by providing explicit bounding boxes, allowing transcription and inference to be evaluated under controlled spatial conditions. This separation enables precise analysis of which component of the pipeline contributes to model errors or performance gains.

**Controlling information availability.**   Another core design principle is controlling how much nucleotide information is directly observable to the model. Tasks T1–T3 operate on fully visible DNA sequences, evaluating recognition and transcription under complete information. In contrast, Task T4 introduces masked regions at the image level, where the nucleotide content within specified regions is visually removed. The model must infer the original sequence from surrounding context.

This contrast allows the benchmark to probe whether a model relies purely on local visual cues or can leverage longer-range sequence context and structural regularities in DNA.

**From passive reading to active search.**   The task suite distinguishes between passive reading of DNA sequences and active, query-driven reasoning. Tasks T1–T4 require the model to process DNA content based on instructions that implicitly

or explicitly specify spatial regions. In contrast, Task T5 introduces an explicit nucleotide query and requires the model to actively search for all matching occurrences within the visualized DNA sequence.

This task mirrors practical sequence analysis workflows, such as motif search or subsequence retrieval, and evaluates the model's ability to combine symbolic matching with spatial reasoning in a visual context.

**Grounding versus non-grounding tasks.**    Task T6 is intentionally designed as a non-grounding control task. Unlike Tasks T1–T5, it does not require spatial localization, region reasoning, or bounding box prediction. Instead, the model must aggregate information across the entire sequence and produce a single chromosome-level label.

Including Task T6 allows direct comparison between grounding-based tasks and a purely global classification setting, helping to disentangle the benefits of spatial reasoning from those of global sequence aggregation.

**Decoupling instruction phrasing from task definition.**    Across all tasks, multiple instruction variants are provided that differ only in verbosity and explicitness. These variants do not change the underlying task definition or the required output format. Instead, they serve to evaluate robustness to instruction phrasing and to prevent performance from being overly dependent on prompt engineering.

By enforcing strict and unified response contracts for each task, the benchmark ensures that evaluation focuses on task competence rather than on sensitivity to instruction wording.

**Summary.**    Together, the six tasks form a coherent and interpretable benchmark for studying DNA understanding in vision–language models. By systematically varying what is visible, what is specified, and what is queried, the task suite enables fine-grained analysis of transcription accuracy, spatial reasoning, contextual inference, and sequence-level understanding within a unified visual framework.

# D. DNA Conversation Construction

In our framework, each task instance is represented as a structured multimodal conversation. This design provides a unified interface for heterogeneous tasks (e.g., free-form OCR, grounding-based localization, and semantic classification), while keeping the data format consistent across tasks and prompt lengths.

### D.1. DNA Conversation Metadata

**Overview.**    We represent each task instance as a structured multimodal conversation designed specifically for DNA sequence understanding. Each conversation corresponds to a question–answer interaction grounded in one or more visual representations of DNA sequences (e.g., rendered genomic text or layouted sequence images). Concretely, a conversation consists of exactly two turns: a user message specifying a DNA-related task and an assistant message providing the corresponding DNA-centric response.

**Formal definition.**    Formally, a conversation $\mathcal{C}$ is defined as an ordered list of messages:

$$\mathcal{C} = [m^{(u)}, m^{(a)}], \tag{S1}$$

where $m^{(u)}$ denotes the user message and $m^{(a)}$ denotes the assistant message. Each message is represented as a tuple:

$$m = (\texttt{role}, \texttt{content}, \texttt{images}), \tag{S2}$$

where `images` is optional and may be omitted depending on the speaker role.

In our implementation, this structure corresponds to the following metadata format:

```
conversation = [
  {"role": "<|User|>", "content": user_content, "images": images},
  {"role": "<|Assistant|>", "content": sample[self.text_key]},
]
```

**Schema.**    Table S6 summarizes the schema of each message. A conversation instance encodes a DNA-related task by jointly specifying (i) the visual DNA context via `images` and (ii) the task semantics via the textual `content` field.

| Field | Type | Description |
|-------|------|-------------|
| role | String token | Speaker identifier for the message, either `<\|User\|>` or `<\|Assistant\|>`. |
| content | Text | Textual payload of the message. For the user message, this field specifies a DNA-centric task (e.g., sequence transcription, localization, or classification). For the assistant message, it contains the expected DNA sequence output or DNA-related label. |
| images | Image list (optional) | Visual representations of DNA sequences associated with the user message. These images provide the genomic context for reading, locating, or reasoning over DNA content. |

*Table S6.* Conversation metadata schema for DNA sequence understanding tasks.

**role.** The `role` field specifies the speaker identity of a message and is mandatory for all messages. We use two reserved role tokens: `<|User|>` and `<|Assistant|>`. In the context of DNA tasks, these roles distinguish between (i) a user query that defines how a DNA sequence should be interpreted or manipulated, and (ii) an assistant response that outputs DNA sequences, subsequences, or sequence-level semantic annotations.

**content.** The `content` field carries the symbolic representation of DNA-related information. Its semantics depend on the speaker role:

- For `<|User|>` messages, `content` encodes a task prompt operating over DNA sequences, such as requesting transcription of nucleotide characters, localization of a specific subsequence, completion of masked genomic regions, or classification of sequence origin.

- For `<|Assistant|>` messages, `content` represents the expected output, which is typically a DNA sequence over a restricted alphabet (A/C/G/T/N), a grounded sequence–location pair, or a sequence-level biological label (e.g., chromosome identity).

Thus, `content` serves as the primary carrier of DNA symbols and DNA-centric task semantics within the conversation.

**images.** The `images` field provides the visual context in which DNA sequences are embedded. Each image corresponds to a rendered view of DNA sequence content, potentially including layout structure, line breaks, or spatial grouping. These images enable tasks that require reading DNA text from visual form or grounding DNA symbols to spatial regions. The assistant message does not include an `images` field, as it outputs textual DNA information conditioned on the user-provided genomic context. A detailed description of the image construction process is provided in Appendix D.2.

**Consistency constraints.** To ensure that each conversation instance is well-formed and semantically meaningful for DNA sequence understanding, we enforce the following constraints:

- A conversation must contain exactly two messages, ordered as user first and assistant second.

- The user message must have `role=<|User|>` and include an `images` field representing DNA sequence content.

- The assistant message must have `role=<|Assistant|>` and must not include an `images` field.

- Both messages must include a `content` field, which must be interpretable in terms of DNA symbols or DNA-centric task semantics.

**Discussion.** By explicitly grounding the conversation metadata in DNA sequences and their visual representations, this design ensures that the dialogue structure is tightly coupled to genomic reasoning rather than serving as a generic conversational interface. All task-specific behaviors are expressed through DNA-centric prompts and output contracts, while the surrounding interaction structure remains fixed, enabling systematic comparison across tasks and prompt lengths.

## D.2. User Images

This subsection specifies the `images` field in the user message and describes how a DNA sequence is rendered into one or multiple image pages. The `images` field provides the visual context for all DNA-centric tasks in our conversation interface (Appendix D.1), and is the only channel through which the nucleotide sequence is presented in visual form.

**Rendering DNA sequences into pages.** Given a DNA sequence of length $n$ (loaded from FASTA), we render it as monospace-like text into a fixed-resolution RGB image canvas. Each page is a $640 \times 640$ 3-channel image, where nucleotides are written row by row (left-to-right within each row, then top-to-bottom across rows). The rendered content contains only the DNA alphabet `A/C/G/T/N`. Any lowercase letters in the source sequence are converted to uppercase prior to rendering. We preserve explicit line breaks induced by the layout (i.e., wrapping at the fixed line width) and include no auxiliary symbols such as coordinates, indices, or separators.

When the entire sequence cannot fit within a single page, we continue rendering on a second page, then a third page, and so on until completion. With the fixed page geometry and typography settings, each page typically accommodates more than $\sim$1800 nucleotides. Unless otherwise noted, we use a fixed font size of `font_size=14` and line spacing `line_spacing=1.6` for all rendered pages.

**Multi-page images field.** For a given sample, the user-side `images` field contains the ordered list of rendered pages corresponding to that sequence. Semantically, we denote this as:

$$I = [\text{page}_1, \text{page}_2, \dots, \text{page}_P], \tag{S3}$$

where $\text{page}_p$ is the $p$-th $640 \times 640$ RGB image and $P$ is the number of pages required to render the full sequence. In our data pipeline, we store `images` in a nested list form `[[page_1, page_2, ..., page_P]]`. The outer list is an implementation-level wrapper (e.g., sample/batch container), while the inner list preserves the page order and constitutes the actual multi-page representation of a single DNA sequence.

**Nucleotide-level bounding-box metadata.** For each rendered DNA page, we store nucleotide-level bounding-box annotations that establish a precise correspondence between nucleotide symbols, their positions in the original DNA sequence, and their pixel regions on the rendered page. Let the $p$-th page be an RGB image $\text{page}_p \in \mathbb{R}^{H \times W \times 3}$ with fixed resolution $H = W = 640$. We define the nucleotide annotation list for page $p$ as:

$$\mathcal{B}^{(p)} = \left[ b_0^{(p)}, b_1^{(p)}, \dots, b_{N_p-1}^{(p)} \right], \tag{S4}$$

where $N_p$ is the number of nucleotides rendered on page $p$. Each element $b_k^{(p)}$ is defined as:

$$b_k^{(p)} = \left( c_k^{(p)}, \ g_k^{(p)}, \ \mathbf{r}_k^{(p)} \right). \tag{S5}$$

The components are defined as follows:

- $c_k^{(p)} \in \{\text{A}, \text{C}, \text{G}, \text{T}, \text{N}\}$ denotes the nucleotide rendered at index $k$ on page $p$.

- $g_k^{(p)}$ is the *global nucleotide index* (`char_index`), indicating the position of this nucleotide in the original full-length DNA sequence, indexed contiguously across pages.

- $p$ is the page index (`page_index`) identifying on which rendered page the nucleotide appears.

- $k$ is the *page-local nucleotide index* (`page_char_index`), defined as a contiguous 0-based index accumulated across all lines on the same page.

- $\mathbf{r}_k^{(p)} = (x_1, y_1, x_2, y_2)$ is the nucleotide bounding box (`page_bbox`) in page pixel coordinates, corresponding to the rectangle returned by the text rendering engine.

**Example.** A concrete annotation instance is:

```
{'char': 'A',
 'char_index': 0,
 'page_index': 1,
 'page_char_index': 0,
 'page_bbox': (20, 23, 29, 33)}
```

This record indicates that the nucleotide A is the first nucleotide of the original DNA sequence (char_index=0), appears on page 1 as the first nucleotide on that page (page_char_index=0), and occupies the pixel region $(x_1, y_1, x_2, y_2) = (20, 23, 29, 33)$ on the rendered image.

**Consistency with the rendering procedure.** Nucleotide-level bounding boxes and indices are generated during rendering using the text engine at the current drawing cursor. Within each line, nucleotides are rendered left-to-right starting from an initial position $(x_0, y_0)$. For the $i$-th nucleotide in a line, we compute its bounding box $\mathbf{r} = (x_1, y_1, x_2, y_2)$ using textbbox((cursor_x, y_0), ch, font) and then advance the horizontal cursor by setting cursor_x $\leftarrow x_2$. The page-local nucleotide index is assigned as page_char_index = page_char_cursor + i, where page_char_cursor is the number of nucleotides already rendered on the current page before processing the current line. The global nucleotide index is assigned as char_index = global_char_cursor + i, where global_char_cursor denotes the total number of nucleotides rendered before the current page, ensuring a contiguous indexing of nucleotides across pages and lines. This construction guarantees a one-to-one alignment between global nucleotide positions, page-local indices, and pixel-level bounding boxes.

**Relation to downstream tasks.** The above image construction enables a unified set of DNA visual understanding tasks. For example, free-form OCR reads the nucleotide text directly from rendered pages, multi-page OCR aggregates outputs across pages, and grounding-based tasks can map predicted subsequences or masked-region completions back to exact spatial locations using page-level coordinates. We defer the specification of task prompts and output contracts to Appendix D.3.

## D.3. User and Assistant Content

This subsection specifies the semantics of the content field for both user and assistant messages in the DNA-centric conversation interface. The user-side content defines task instructions operating over nucleotide sequences rendered in visual form, while the assistant-side content encodes nucleotide-level outputs or sequence-level predictions derived from the same DNA context. Together, these two roles define the textual interface through which visualized DNA sequences are queried and interpreted.

**User Content.** The user-side content field contains a natural-language instruction specifying what operation should be performed on the DNA sequence shown in the input images. The instruction operates over the visual representation of the sequence and does not explicitly include the nucleotide sequence itself.

**Assistant Content.** The assistant-side content field contains the textual response corresponding to the user instruction. This response is derived from the DNA sequence rendered in the images and may take the form of nucleotide strings, grounded nucleotide regions, or discrete labels.

**From fields to task protocols.** While the content field is defined at the message level, its precise semantics are determined only when a user instruction is paired with a corresponding assistant response. We therefore describe the task-level instruction–response definitions in a dedicated section, where each task is specified as a paired protocol over user and assistant contents (Section E).

## E. Paired Instruction–Response Protocol

This section defines the task-level instruction–response protocols used throughout this work. Each task is specified by a paired design consisting of a user-side instruction and a corresponding assistant-side output contract, both operating over the same visualized DNA sequence. By formulating tasks as paired protocols rather than independent prompts or outputs, we ensure that task semantics, output formats, and evaluation criteria are explicitly defined and reproducible.

### E.1. Task T1: Free-Form DNA Transcription

**Task definition.** Task T1 requires the model to transcribe the entire DNA sequence rendered in the input images. The task is formulated as free-form OCR without spatial grounding or region constraints. Although the user instruction may vary in verbosity, the assistant response is governed by a single, fixed construction rule.

**Response construction.** Let the input consist of $P$ rendered pages, where each page contains multiple lines of nucleotide characters. The assistant response is constructed by reading the DNA sequence in a deterministic order: from left to right within each line, from top to bottom across lines, and from the first page to the last page. Formally, the response is defined as a single text string

$$R = L_{1,1}L_{1,2}\cdots L_{1,m_1}L_{2,1}\cdots L_{P,m_P}, \tag{S6}$$

where $L_{p,i}$ denotes the $i$-th line of nucleotide characters on page $p$, and $m_p$ is the number of lines on that page. The output contains only nucleotide characters from the alphabet A/C/G/T/N and preserves the line breaks induced by the page layout. No additional text or symbols are permitted.

**Instruction variants.** Table S7 summarizes the three instruction variants (SHORT, MEDIUM, and LONG) used for Task T1, together with the unified response template. All instruction variants correspond to the same transcription task and differ only in the explicitness of the instruction. Regardless of instruction length, the assistant output must conform to the response construction defined above.

| Length | Instruction | Response |
|---|---|---|
| SHORT | Free OCR. | page 1, line 1 |
| MEDIUM | Free OCR.
Return only DNA sequence. | page 1, line 2
⋮ |
| LONG | Free OCR.
Return only the DNA sequence (A/C/G/T/N).
Keep line breaks if present.  No extra words. | page $P$, last line |

*Table S7.* Instruction variants and unified response template for Task T1 (Free-Form DNA Transcription).

### E.2. Task T2: DNA Transcription with Spatial Grounding

**Task definition.** Task T2 extends free-form DNA transcription by additionally requiring spatial grounding. Given one or multiple rendered DNA pages, the model must transcribe all nucleotide sequences and associate each transcribed line or text block with its corresponding bounding box on the input images. The task therefore jointly evaluates nucleotide recognition and spatial localization.

**Response construction.** Let the input consist of $P$ rendered pages. For each text region detected on the pages, the assistant outputs a pair consisting of (i) the transcribed nucleotide sequence and (ii) the spatial extent of that sequence in image coordinates. Regions are ordered in reading order: from top to bottom and left to right within each page, and from the first page to the last page. Each region produces exactly one response line.

Formally, the response is a sequence of lines of the form

$$\langle r_i, b_i \rangle,$$

where $r_i$ is the nucleotide sequence transcribed from region $i$, and $b_i = [img\_id, x_1, y_1, x_2, y_2]$ denotes the bounding box of that region.

**Instruction variants and response template.** The instruction variants for Task T2 are summarized in Table S8. All variants correspond to the same grounding-based transcription task and differ only in the explicitness of the instruction. Regardless of instruction length, the assistant output must conform to the unified response template shown in Table S9 and the response construction rules defined above.

| Length | Instruction |
|---|---|
| SHORT | `<|grounding|>Read all DNA text and locate each line.` |
| MEDIUM | `<|grounding|>Read all DNA text and locate each line.`
`Output per line:  <|ref|>SEQ<|/ref|><|det|>[[img_id,x1,y1,x2,y2]]<|/det|>.` |
| LONG | `<|grounding|>Read all DNA text and locate each text line or block.`
`Output one line per region in reading order (top-to-bottom, left-to-right).`
`For each region, output EXACTLY:`
`<|ref|>SEQUENCE<|/ref|><|det|>[[img_id,x1,y1,x2,y2]]<|/det|>`
`Rules:`
`- det MUST be a list-of-boxes.  Even one box must be written as [[...]].`
`- img_id is 0-based index into the images list.  If NUM_IMAGES==1, img_id MUST`
`be 0.`
`Only output these lines.  No extra text.` |

*Table S8.* Instruction variants for Task T2 (DNA Transcription with Spatial Grounding).

| Response Template |
|---|
| `<|ref|>SEQUENCE<|/ref|><|det|>[[img_id,x1,y1,x2,y2]]<|/det|>`
`<|ref|>SEQUENCE<|/ref|><|det|>[[img_id,x1,y1,x2,y2]]<|/det|>`
$\vdots$ |

*Table S9.* Unified response template for Task T2 (DNA Transcription with Spatial Grounding).

## E.3. Task T3: ROI-Based DNA Transcription

**Task definition.** Task T3 focuses on region-of-interest (ROI) based DNA transcription. Unlike Task T2, which requires the model to detect and transcribe all DNA text regions present on the page, Task T3 assumes that the regions of interest are provided explicitly as a set of predefined bounding boxes. The model is required to transcribe the nucleotide sequence contained within each specified region, without performing any form of region proposal or detection. This task isolates nucleotide recognition under known spatial constraints and evaluates transcription accuracy when spatial localization is given.

**Response construction.** Let the input consist of $P$ rendered DNA pages together with a list of $K$ bounding boxes. Each bounding box specifies a rectangular region on a particular page and corresponds to a single DNA text region. The assistant outputs exactly one response line for each bounding box.

Bounding boxes are processed strictly in the order provided by the user. For the $i$-th bounding box, the assistant outputs (i) the nucleotide sequence transcribed from that region and (ii) the spatial coordinates of the region in image space. No reordering, merging, or splitting of regions is permitted.

Formally, the response is a sequence of $K$ lines of the form

$$\langle r_i, b_i \rangle,$$

where $r_i$ denotes the nucleotide sequence contained in the $i$-th region, and $b_i = [img\_id, x_1, y_1, x_2, y_2]$ denotes the bounding box associated with that region.

**Bounding box list representation.** For Task T3, the user instruction includes an explicit list of bounding boxes specifying the regions of interest. Each bounding box is represented as a 5-tuple of integers

$$[img\_id, x_1, y_1, x_2, y_2],$$

where $img\_id$ denotes the zero-based index of the image page, and $(x_1, y_1)$ and $(x_2, y_2)$ specify the top-left and bottom-right coordinates of the rectangular region in image pixel space. The list of bounding boxes is provided as a Python-style nested list, for example:

$$[[0, 30, 915, 960, 945], [0, 32, 960, 880, 990]].$$

Bounding boxes are ordered, and the assistant must generate responses in exactly the same order as the input list.

| Length | Instruction |
|---|---|
| SHORT | `<|grounding|>OCR DNA for boxes (in order):  {boxes}` |
| MEDIUM | `<|grounding|>OCR DNA text for each box in the SAME order.`
`Boxes:`
`{boxes}`
`Output one line per box:  <|ref|>SEQ<|/ref|><|det|>[[img_id,x1,y1,x2,y2]]<|/det|>` |
| LONG | `<|grounding|>OCR DNA text for each bounding box below, in the SAME order.`
`Boxes:`
`{boxes}`
`Output one line per box using EXACTLY:`
`<|ref|>SEQUENCE<|/ref|><|det|>[[img_id,x1,y1,x2,y2]]<|/det|>`
`Rules:`
`- det MUST be list-of-boxes:  [[...]]  for a single box.`
`- img_id is 0-based index into the images list.  If NUM_IMAGES==1, img_id MUST`
`be 0.`
`No extra text.` |

*Table S10.* Instruction variants for Task T3 (ROI-Based DNA Transcription). The placeholder {boxes} is instantiated with the ordered list of bounding boxes specifying the regions of interest.

| Response Template |
|---|
| `<|ref|>SEQUENCE<|/ref|><|det|>[[img_id,x1,y1,x2,y2]]<|/det|>`
`<|ref|>SEQUENCE<|/ref|><|det|>[[img_id,x1,y1,x2,y2]]<|/det|>`
⋮ |

*Table S11.* Unified response template for Task T3 (ROI-Based DNA Transcription).

**Instruction variants and response template.**   The instruction variants for Task T3 are summarized in Table S10. All variants correspond to the same ROI-based transcription task and differ only in verbosity and explicitness. Regardless of instruction length, the assistant output must strictly conform to the unified response template specified in Table S11 and to the response construction rules defined in this section. The instruction variants shown in Table S10 include a placeholder {boxes}, which is instantiated with the ordered list of bounding boxes described in the bounding box list representation.

### E.4. Task T4: Masked DNA Completion

**Task definition.**   Task T4 extends Task T3 to a masked completion setting. While Task T3 transcribes *visible* nucleotides within known regions, Task T4 requires the model to *infer missing nucleotide content* when the DNA text inside specified regions is deliberately masked. The spatial extents of the masked regions are provided explicitly as bounding boxes, and the model must predict the original nucleotide sequences using surrounding visual and contextual information.

**Image masking procedure.**   Prior to inference, the DNA text inside each target bounding box is masked at the image level by replacing the corresponding rectangular region with a uniform white background. All other regions remain unchanged. As a result, the nucleotide content within the masked boxes is not directly observable, and successful prediction requires contextual reasoning beyond local visual cues.

**Response construction.**   Task T4 follows the same response construction protocol as Task T3 (Appendix E.3), including ordered processing of bounding boxes and one-to-one correspondence between predicted sequences and regions. The only difference is that the output sequence corresponds to the *original nucleotide content* of each masked region. Predicted sequences must use characters from the alphabet A/C/G/T/N, where N denotes uncertainty. The instruction variants for Task T4 are summarized in Table S12. Regardless of instruction length, the assistant output must strictly conform to the unified response template specified in Table S13 and to the response construction rules defined in this section.

| Length | Instruction |
|---|---|
| SHORT | `<|grounding|>Predict masked DNA for boxes (in order): {boxes}` |
| MEDIUM | `<|grounding|>Predict ORIGINAL DNA for masked boxes in the SAME order.`
`Boxes:`
`{boxes}`
`Output: <|ref|>SEQ<|/ref|><|det|>[[img_id,x1,y1,x2,y2]]<|/det|>` |
| LONG | `<|grounding|>The DNA text inside each box is masked/occluded.`
`Predict the ORIGINAL DNA sequence for each masked region.`
`Masked boxes:`
`{boxes}`
`Output in the SAME order as the boxes.`
`Use only A/C/G/T/N (use N if uncertain).`
`For each box, output EXACTLY one line:`
`<|ref|>PREDICTED_SEQUENCE<|/ref|><|det|>[[img_id,x1,y1,x2,y2]]<|/det|>`
`Rules:`
`- det MUST be list-of-boxes: [[...]] for a single box.`
`- img_id is 0-based index into the images list. If NUM_IMAGES==1, img_id MUST`
`be 0.`
`No extra text.` |

*Table S12.* Instruction variants for Task T4 (Masked DNA Completion), where {`boxes`} is instantiated with the ordered list of masked region bounding boxes.

| Response Template |
|---|
| `<|ref|>PREDICTED_SEQUENCE<|/ref|><|det|>[[img_id,x1,y1,x2,y2]]<|/det|>`
`<|ref|>PREDICTED_SEQUENCE<|/ref|><|det|>[[img_id,x1,y1,x2,y2]]<|/det|>`
$\vdots$ |

*Table S13.* Unified response template for Task T4 (Masked DNA Completion).

### E.5. Task T5: DNA Subsequence Localization

**Task definition.** Task T5 focuses on query-driven DNA subsequence localization. Given one or multiple rendered DNA pages and a query nucleotide sequence, the model is required to locate all occurrences of the query subsequence within the visualized DNA content and return their corresponding spatial extents. This task evaluates the model's ability to jointly perform nucleotide-level sequence matching and spatial localization under a visual representation of DNA.

Unlike Tasks T3 and T4, which operate on predefined spatial regions, Task T5 is driven by a symbolic nucleotide query and requires the model to search for matching subsequences across the entire set of rendered pages.

**Response construction.** Let the input consist of $P$ rendered DNA pages together with a query nucleotide sequence $q$. The assistant produces exactly one response line corresponding to the query.

If the query subsequence appears one or more times in the rendered DNA content, the assistant outputs all bounding boxes corresponding to the matched regions. If the query does not appear, the assistant outputs an empty list of bounding boxes. The ordering of bounding boxes follows the reading order of the matched regions: from top to bottom and left to right within each page, and from the first page to the last page.

Formally, the response takes the form

$$\langle q, B \rangle,$$

where $q$ denotes the query nucleotide sequence, and $B = \{b_1, b_2, \ldots, b_M\}$ is a (possibly empty) ordered list of bounding boxes. Each bounding box is represented as $b_i = [img\_id, x_1, y_1, x_2, y_2]$.

**Bounding box representation.** Each bounding box returned in Task T5 follows the same spatial representation used in Tasks T3 and T4. Specifically, bounding boxes are represented as 5-tuples of integers

$$[img\_id, x_1, y_1, x_2, y_2],$$

where *img_id* denotes the zero-based index of the image page, and $(x_1, y_1)$ and $(x_2, y_2)$ specify the top-left and bottom-right coordinates of the rectangular region in image pixel space. Bounding boxes are ordered according to the reading order of the corresponding matched subsequences.

**Instruction variants and response template.** The instruction variants for Task T5 are summarized in Table S14. All variants describe the same subsequence localization task and differ only in verbosity and explicitness. Regardless of instruction length, the assistant output must strictly conform to the unified response template specified in Table S15 and to the response construction rules defined in this section.

| Length | Instruction |
|--------|-------------|
| SHORT | `<|grounding|>Locate <|ref|>{query}<|/ref|>.` |
| MEDIUM | `<|grounding|>Locate <|ref|>{query}<|/ref|>.`
`Output exactly one line: <|ref|>QUERY<|/ref|><|det|>[...]<|/det|> (or [] if`
`not found).` |
| LONG | `<|grounding|>Locate the DNA subsequence <|ref|>{query}<|/ref|>.`
`Return ALL bounding boxes where it appears.`
`Output EXACTLY one line:`
`<|ref|>{query}<|/ref|><|det|>[[img_id,x1,y1,x2,y2],[img_id,x1,y1,x2,y2]]<|/det|>`
`If not found, output:`
`<|ref|>{query}<|/ref|><|det|>[]<|/det|>`
`Rules:`
`- Each box MUST be [img_id,x1,y1,x2,y2].`
`- img_id is 0-based index into the images list. If NUM_IMAGES==1, img_id MUST`
`be 0.`
`No extra text.` |

*Table S14.* Instruction variants for Task T5 (DNA Subsequence Localization), where {`query`} is instantiated with the nucleotide subsequence to be localized.

| Response Template |
|-------------------|
| `<|ref|>QUERY<|/ref|><|det|>[[img_id,x1,y1,x2,y2],[img_id,x1,y1,x2,y2]]<|/det|>`
*or*
`<|ref|>QUERY<|/ref|><|det|>[]<|/det|>` |

*Table S15.* Unified response template for Task T5 (DNA Subsequence Localization).

### E.6. Task T6: Chromosome Classification

**Task definition.** Task T6 focuses on chromosome-level classification of DNA sequences and serves as a non-grounding baseline task. Given one or multiple rendered DNA pages corresponding to a single DNA sequence, the model is required to predict which human chromosome the sequence originates from. Unlike Tasks T1–T5, Task T6 does not involve spatial grounding, region localization, or bounding box reasoning. Instead, it evaluates the model's ability to aggregate global sequence-level information from a visual DNA representation and perform high-level semantic classification.

**Response construction.** Let the input consist of one or more rendered DNA pages corresponding to a single DNA sequence. The assistant outputs exactly one label indicating the predicted chromosome identity.

The output must be a single token selected from the predefined label set:

$$\{\texttt{chr1}, \texttt{chr2}, \dots, \texttt{chr22}, \texttt{chrX}, \texttt{chrY}, \texttt{unknown}\}.$$

No additional text, symbols, or grounding tokens are allowed.

**Instruction variants and response template.** The instruction variants for Task T6 are summarized in Table S16. All variants correspond to the same chromosome classification task and differ only in verbosity. As shown in Table S17, regardless of instruction length, the assistant output must strictly conform to the response format defined in this section.

| Length | Instruction |
|--------|-------------|
| SHORT | Which chromosome? |
| MEDIUM | Predict chromosome label (chr1-22, chrX, chrY, or unknown). |
| LONG | Classify this DNA sequence: which human chromosome does it belong to? Answer with one label only: chr1-chr22, chrX, chrY, or unknown. |

*Table S16.* Instruction variants for Task T6 (Chromosome Classification).

| Response Template |
|-------------------|
| chr1 \| chr2 \| ...\| chr22 \| chrX \| chrY \| unknown |

*Table S17.* Unified response template for Task T6 (Chromosome Classification).

## F. Details of The Pretraining

### F.1. HG38 Pretraining

Table S18 summarizes the HG38 pretraining hyperparameters of OpticalDNA under our two-stage adaptation schedule. Both stages share the same model architecture and training objective, while differing in the emphasis of trainable modules and several dataset-level controls that improve robustness across heterogeneous prompted outputs.

**Two-stage adaptation.** Stage 1 adapts generation and document aggregation by fine-tuning the decoder $G_\psi$ with LoRA and updating the multi-page fusion module $\mathcal{F}_\theta$ with full parameters. Stage 2 shifts the emphasis to visual–text alignment by tuning the projector $\Pi_\theta$ with a higher-capacity LoRA configuration (rank $r = 128$), while keeping the fusion design unchanged. The LoRA targets in $G_\psi$ follow standard Transformer projections in both attention and MLP blocks (Table S18).

**Task sampling.** During pretraining, we balance the six prompt families by sampling tasks from a categorical distribution $\pi$ over (T1–T6), reported in Table S18. This explicitly controls the frequency of different supervision types (free-form transcription, grounding, ROI reading, completion, localization, and retrieval-style prompts), and prevents training from being dominated by the most verbose outputs.

**Tail truncation.** To improve robustness under heterogeneous ROI supervision, we apply *tail truncation* with parameters $(\rho_{\text{base}}, \rho_{\text{max}})$ (Table S18). We enable tail truncation by default for T2/T3/T5, since these tasks are ROI-centric and depend on region-level supervision (page-level bounding boxes), where training is typically more sensitive to missing or shortened region signals. Concretely, given a training sample with a flattened list of region items, we randomly delete a suffix of items, where the truncation ratio is sampled and bounded between $\rho_{\text{base}}$ and $\rho_{\text{max}}$. For the truncated items, we additionally white out their corresponding bounding-box regions on the rendered pages, and then drop empty pages and re-index remaining pages to preserve image–annotation consistency. This augmentation encourages the model to remain stable when fewer ROIs are available or when ROI supervision is partially incomplete.

**Line-span sampling.** For line-/block-based supervision, we randomize region construction using a line-span sampler. We first sample a span length $n_{\text{base}} \sim \text{Uniform}\{n_{\text{base}}^{\min}, \ldots, n_{\text{base}}^{\max}\}$, and then sample the number of spans $n_{\text{samp}} \sim \text{Uniform}\{n_{\text{samp}}^{\min}, \ldots, n_{\text{samp}}^{\max}\}$. Each span is instantiated by selecting a source line and taking a contiguous character span of length $n_{\text{base}}$ within that line. This procedure diversifies both the size and the count of regions per sample, preventing the model from overfitting to a fixed ROI layout.

**Subsequence localization sampling.** For subsequence localization (T5), we sample a query length $\ell \sim \text{Uniform}\{\ell_{\min}, \ldots, \ell_{\max}\}$. We then enumerate matches across lines to form localization targets. We allow overlapping matches by using a unit stride; otherwise we advance by $\ell$ to prevent overlap, controlling the match density (Table S18).

**Prompt-length curriculum annealing.** We anneal the sampling distribution over prompt-length variants (LONG/MEDIUM/SHORT) during training. Let $\mathcal{L} = \{\text{LONG}, \text{MEDIUM}, \text{SHORT}\}$ and $\pi_{\text{start}}^{\text{len}}, \pi_{\text{end}}^{\text{len}} \in \Delta^3$ denote the start/end distributions. In our setting, we use $\pi_{\text{start}}^{\text{len}} = (0.2, 0.2, 0.6)$ and $\pi_{\text{end}}^{\text{len}} = (0.1, 0.1, 0.8)$ for (LONG/MEDIUM/SHORT). At step $s$, we clamp and interpolate

$$t(s) = \frac{\min(\max(s, 0), S_{\text{anneal}})}{S_{\text{anneal}}}, \qquad \tilde{\pi}^{\text{len}}(s) = (1 - t(s))\, \pi_{\text{start}}^{\text{len}} + t(s)\, \pi_{\text{end}}^{\text{len}}, \tag{S7}$$

*Table S18.* HG38 pretraining hyperparameters of OpticalDNA under the two-stage adaptation schedule. Values in Stage 2 colored with orange differ from Stage 1.

| Hyperparameter | Stage 1 | Stage 2 |
|---|:---:|:---:|
| **LoRA / tunable modules** | | |
| LoRA rank $r$ | 16 | 128 |
| LoRA scaling $\alpha$ | 16 | 16 |
| LoRA dropout | 0.05 | 0.05 |
| LoRA target in $G_\psi$ | {q,k,v,o, gate, up, down}_proj | – |
| Update strategy in $\Pi_\theta$ | – | all parameter (LoRA) |
| **Multi-page fusion module $\mathcal{F}_\theta$** | | |
| $\mathcal{F}_\theta$ proj_in / proj_out | FALSE | FALSE |
| $\mathcal{F}_\theta$ #attention layers | 1 | 1 |
| $\mathcal{F}_\theta$ #attention heads | 20 | 20 |
| $\mathcal{F}_\theta$ update strategy | full | full |
| **Data sampling and randomization** | | |
| Task sampling $\boldsymbol{\pi}$ over (T1–T6) | (0.25, 0.20, 0.15, 0.15, 0.15, 0.10) | (0.17, 0.17, 0.17, 0.17, 0.17, 0.15) |
| Tail truncation ($\rho_{\text{base}}, \rho_{\text{max}}$) | (0, 0.5) | (0, 0.98) |
| Line span ($n_{\text{base}}^{\min}, n_{\text{base}}^{\max}$) | (1, 8) | (1, 8) |
| Line span ($n_{\text{samp}}^{\min}, n_{\text{samp}}^{\max}$) | (1, 3) | (1, 3) |
| Unique lines | true | true |
| Subseq locate length ($\ell_{\min}, \ell_{\max}$) | (6, 64) | (6, 32) |
| Allow overlap | true | true |
| **Training** | | |
| Global batch size ($\times 8 \times$H100) | 8 | 8 |
| Max text tokens | 4096 | 4096 |
| Warmup steps | 10,000 | 20,000 |
| Annealed sampler total steps | 112,500 | 1 |
| Learning rate | 2e-4 | 5e-4 |
| Optimizer | adamw_8bit | adamw_8bit |
| Weight decay | 1e-3 | 1e-3 |
| LR scheduler | linear | linear |
| Training steps | 227,600 | 190,000 |
| **Trainable parameters** | 90.6M / 3.43B (2.64%) | 13.5M / 3.35B (0.40%) |

and normalize $\boldsymbol{\pi}^{\text{len}}(s) = \tilde{\boldsymbol{\pi}}^{\text{len}}(s) / \sum_i \tilde{\pi}_i^{\text{len}}(s)$ before sampling a length variant. This curriculum gradually increases the probability of shorter prompts, improving robustness under reduced instruction context.

**Optimization and length cap.** We cap the maximum decoded length $T_{\max}$ (Max text tokens) as reported in Table S18, which keeps memory usage predictable under multi-page outputs. We use a linear schedule with warmup and AdamW (8-bit); all remaining optimization hyperparameters follow Table S18.

### F.2. Rice Pretraining

**Pretraining source (T2T-NIP).** We pretrain on the Nipponbare (*O. sativa japonica*) telomere-to-telomere reference genome (T2T-NIP) from RiceSuperPIRdb (Shang et al., 2022). T2T-NIP provides a gap-free assembly together with curated functional element annotations across all 12 rice chromosomes. Compared to IRGSP-1.0, it resolves the remaining $\sim$3% previously uncharacterized genomic regions, substantially improving completeness.

**Pretraining protocol and hyperparameters.** Table S19 summarizes the hyperparameter configuration used for Rice pretraining. Unless explicitly stated, the training objective, architecture, dataset construction pipeline, and sampling strategies follow the HG38 pretraining setup in Section F.1. The main difference is the adaptation strategy for the visual–text projector $\Pi_\theta$: while HG38 pretraining performs full-parameter finetuning, Rice pretraining adopts a LoRA-based update to reduce trainable parameters and improve training efficiency. All other settings (task sampling, tail truncation, line-span and subsequence localization samplers, prompt-length curriculum, and optimization schedule) are reported in Table S19.

*Table S19.* Rice pretraining hyperparameters of OpticalDNA.

| Hyperparameter | Value |
|---|---|
| **LoRA / tunable modules** | |
| LoRA rank $r$ | 128 |
| LoRA scaling $\alpha$ | 16 |
| LoRA dropout | 0.05 |
| LoRA target in $G_\psi$ | {q,k,v,o, gate, up, down}_proj |
| Update strategy in $\Pi_\theta$ | all parameters (LoRA) |
| **Multi-page fusion module $\mathcal{F}_\theta$** | |
| $\mathcal{F}_\theta$ proj_in / proj_out | FALSE |
| $\mathcal{F}_\theta$ #attention layers | 1 |
| $\mathcal{F}_\theta$ #attention heads | 20 |
| $\mathcal{F}_\theta$ update strategy | full |
| **Data sampling and randomization** | |
| Task sampling $\pi$ over (T1–T6) | (0.17, 0.17, 0.17, 0.17, 0.17, 0.15) |
| Tail truncation $(\rho_{\text{base}}, \rho_{\text{max}})$ | (0, 0.98) |
| Line span $(n_{\text{base}}^{\min}, n_{\text{base}}^{\max})$ | (1, 8) |
| Line span $(n_{\text{samp}}^{\min}, n_{\text{samp}}^{\max})$ | (1, 3) |
| Unique lines | true |
| Subseq locate length $(\ell_{\min}, \ell_{\max})$ | (6, 32) |
| Allow overlap | true |
| **Training** | |
| Global batch size ($\times 8 \times$H100) | 8 |
| Max text tokens | 4096 |
| Warmup steps | 20,000 |
| Annealed sampler total steps | 112,500 |
| Learning rate | 8e-4 |
| Optimizer | adamw_8bit |
| Weight decay | 1e-3 |
| LR scheduler | linear |
| Training steps | 150,000 |
| **Trainable parameters** | 633.6M / 3.97B (15.96%) |

# G. Details of The Downstream Tasks

## G.1. DNALONGBENCH

For readability, we use standard GTEx tissue abbreviations when reporting DNALONGBENCH eQTL results. Table S20 lists the abbreviations and their corresponding full names.

## G.2. Rice Subspecies Benchmark (RiceSubBench)

RiceSubBench evaluates long-sequence generalization under phylogeny-aware domain shift in rice genomes: we first define evaluation domains relative to the japonica pretraining source, and then construct the same downstream task within each domain.

**Data source and domain taxonomy.** We collect rice accessions from RiceSuperPIRdb, where each *accession* corresponds to a unique rice genome sample with associated reference sequence and annotations. We group accessions into five taxonomic groups: *O. sativa japonica*, *O. sativa aus*, *O. rufipogon*, *O. barthii*, and *O. glaberrima*. Table S21 reports the number of accessions per group.

**Phylogeny-aware domain partition.** Since our rice pretraining uses the Nipponbare *O. sativa japonica* reference genome, we define domains by increasing distance from japonica to evaluate cross-species generalization, consistent with rice evolutionary relationships : *O. rufipogon* is the wild progenitor of Asian cultivated rice (japonica/aus), while African rice *O. glaberrima* follows an independent domestication path from *O. barthii*. We evaluate *In-Domain (ID)* on *O. sativa japonica*

*Table S20.* GTEx tissue abbreviations used in DNALONGBENCH.

| Abbreviation | Full name |
|---|---|
| CCF | Cells Cultured fibroblasts |
| WB | Whole Blood |
| Thyroid | Thyroid |
| SNSES | Skin Not Sun Exposed Suprapubic |
| SSELL | Skin Sun Exposed Lower leg |
| MS | Muscle Skeletal |
| NT | Nerve Tibial |
| AT | Artery Tibial |
| AS | Adipose Subcutaneous |

*Table S21.* Accession statistics for RiceSubBench by taxonomic group (RiceSuperPIRdb).

| Group | #Accessions |
|---|---|
| *O. sativa japonica* | 58 |
| *O. rufipogon* | 26 |
| *O. glaberrima* | 11 |
| *O. barthii* | 8 |
| *O. sativa aus* | 4 |

accessions excluding Nipponbare, and define a three-level OOD gradient: *Near-OOD* (aus), *Mid-OOD* (rufipogon), and *Far-OOD* (barthii, glaberrima).

**Task and dataset construction (SPLICE_SITE_3CLASS).** Within each domain, we build a long-context splice site classification benchmark. Given a 16K-bp genomic window, the model predicts DONOR, ACCEPTOR, or NONE. We derive donor/acceptor junctions from transcript exon structures by ordering exons in the transcript 5'→3' direction (enforcing strand consistency per transcript), and sample positive windows centered at junctions with small jitter, using at most one window per junction to reduce redundancy. To form hard negatives, we sample NONE windows *inside gene regions* while enforcing a minimum distance ($\pm K$ bp) to any annotated junction, avoiding trivial "genic vs. intergenic" shortcuts. For each domain we subsample to $N = 6000$ windows. Samples are generated in a streaming manner over accessions and deterministically assigned to train/validation/test splits (0.8/0.1/0.1) using a seeded hashing scheme for reproducibility. Table S22 summarizes the resulting class distribution across domains.

*Table S22.* Class distribution of SPLICE_SITE_3CLASS in RiceSubBench across evaluation domains.

| Domain | Donor | Acceptor | None | Total |
|---|---|---|---|---|
| japonica | 2747 | 2767 | 486 | 6000 |
| aus | 2714 | 2634 | 652 | 6000 |
| rufipogon | 2813 | 2798 | 389 | 6000 |
| barthii | 2638 | 2607 | 755 | 6000 |
| glaberrima | 2625 | 2659 | 716 | 6000 |

### G.3. Rice Whole-Genome Phenotype Benchmark (RiceWGPB)

We construct RiceWGPB to benchmark *whole-genome* phenotype prediction in rice, where the input is a complete genome sequence and the output is an organism-level quantitative trait. The dataset is built via a three-step pipeline:

- **Phenotype acquisition.** We download rice phenotype records from the RFGB database (ZHANG et al., 2015) (https://v1.rmbreeding.cn/phenotype), which contains measurements for multiple agronomic traits.

- **ID matching to whole-genome FASTA.** Since RFGB does not provide genome FASTA sequences, we match phenotype records to RiceSuperPIRdb accessions using shared identifiers and retrieve the corresponding whole-genome FASTA as model input.

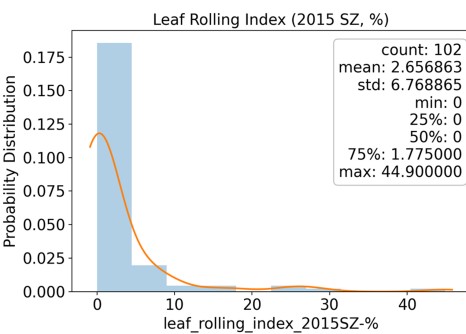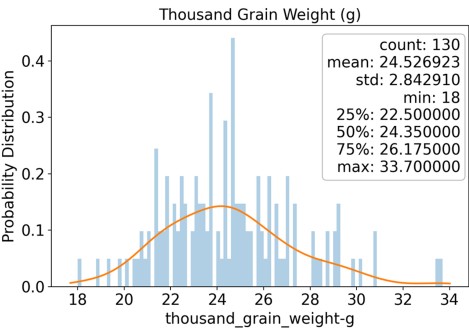

*Figure S1.* Label distributions of RiceWGPB phenotypes. Left: leaf rolling index measured in the 2015 SZ environment (LRI-15SZ). Right: thousand grain weight (TGW).

- **Trait filtering.** Due to limited coverage after matching, we retain only two phenotypes with sufficient sample size: thousand grain weight (TGW) and leaf rolling index measured in the 2015 SZ environment (LRI-15SZ).

For both tasks, we perform *species-level* splitting by `Accession` (each accession corresponds to a unique rice variety/individual) with a 70/10/20 train/valid/test ratio; dataset statistics are summarized in Table S23.

*Table S23.* Dataset statistics of RiceWGPB. We report the number of samples and accession-level train/valid/test splits (70/10/20, seed=42).

| Task | #Samples | Train | Valid | Test |
|------|----------|-------|-------|------|
| LRI-15SZ (%) | 102 | 71 | 10 | 21 |
| TGW (g) | 130 | 91 | 13 | 26 |

Moreover, Figure S1 shows the label distributions of the two phenotypes. LRI-15SZ is highly skewed with a heavy tail (median = 0, max = 44.9), where most accessions exhibit little to no leaf rolling while a small subset shows strong rolling responses. In contrast, TGW follows a more concentrated unimodal distribution around 24–25 g (mean = 24.53, std = 2.84), representing a relatively stable quantitative trait with moderate variability. Together, they form a complementary benchmark: TGW evaluates prediction under a well-behaved continuous target, whereas LRI-15SZ stresses robustness under a long-tailed and highly imbalanced label distribution.

### G.4. Motivation: a progressive evaluation from regulatory correctness to genome-scale utility

Long-sequence genomic modeling is commonly evaluated using regulatory benchmarks that test whether models capture local and long-range sequence patterns associated with molecular functions. While such benchmarks are essential, they do not fully reflect the requirements of practical genome-scale applications, which demand robustness to genetic diversity and the ability to support downstream phenotype-level decisions. We therefore organize our downstream evaluation into three progressively challenging stages, using rice as a representative and practically important crop genome. Rice provides a particularly suitable testbed due to its well-annotated reference genomes, rich subspecies diversity, and the central role of genome-to-phenotype modeling in crop improvement and breeding (Zhang et al., 2021; Long et al., 2024).

**Stage 1: Canonical regulatory function modeling (DNALONGBENCH).** DNALONGBENCH evaluates whether a model correctly captures regulatory signals such as splice sites under a standard reference genome setting. It primarily tests mechanistic correctness, including motif recognition and mid-to-long-range dependencies, and serves as a widely adopted baseline for long-sequence representation quality.

**Stage 2: Regulatory modeling under population-level genetic variation (RiceSubBench).** RiceSubBench retains the same regulatory labels (donor/acceptor/none) but constructs the task across multiple rice subspecies. This setting reflects a realistic scenario in crop genomics, where regulatory mechanisms are conserved but embedded in diverse genetic backgrounds shaped by domestication and breeding. Rather than introducing new biological semantics, this benchmark

probes the robustness of regulatory representations under population-level genotype variation and domain shift. A model that overfits to a single reference genome may perform well on DNALONGBENCH but degrade under this setting, making RiceSubBench a critical intermediate stage.

**Stage 3: Genome-wide phenotype prediction (RiceWGPB).**   RiceWGPB evaluates whole-genome phenotype prediction, where labels correspond to organism-level traits of agronomic relevance. Unlike regulatory tasks with localized supervision, phenotypic outcomes aggregate weak and distributed signals across the entire genome. This stage reflects a core challenge in crop genomics and breeding, and tests whether long-sequence representations can be effectively summarized and utilized for genome-scale decision-making.

**A progressive view.**   Taken together, the three benchmarks form a progressive evaluation pipeline: from canonical regulatory correctness, to robustness under population-level genetic variation, and finally to genome-wide phenotypic utility. By grounding the latter two stages in rice genomics, this progression provides a practical full-cycle assessment of long-sequence representations in a setting that is both biologically meaningful and application-relevant.

# H. Additional Experimental Details

This appendix summarizes (i) the hyperparameter search space used for OpticalDNA, and (ii) the implementation details for reproducing Evo-2 and LucaOne baselines on selected benchmarks.

## H.1. Hyperparameter Search Space for OpticalDNA

We tune hyperparameters via grid search and report the test performance corresponding to the best validation score.

**DNA Long-Range Benchmark.**   We search learning rates in $\{10^{-3}, 10^{-4}, 8 \times 10^{-5}, 5 \times 10^{-5}, 10^{-5}\}$, batch sizes in $\{32, 64, 128\}$, and weight decays in $\{10^{-2}, 3 \times 10^{-3}, 10^{-3}\}$. All models are trained for 100 epochs.

**Rice Subspecies Benchmark.**   We search learning rates in $\{10^{-3}, 10^{-4}, 8 \times 10^{-5}\}$, batch sizes in $\{32, 64, 128\}$, and weight decays in $\{10^{-2}, 3 \times 10^{-3}, 10^{-3}\}$. Since OpticalDNA typically converges faster on this benchmark, we train all configurations for 30 epochs.

**Whole-Genome Phenotype Benchmark.**   We search learning rates in $\{3 \times 10^{-4}, 5 \times 10^{-4}, 8 \times 10^{-4}\}$, batch sizes in $\{16, 32, 64\}$, and fix the weight decay to $10^{-5}$. We find that longer training consistently improves performance, and thus train for 400 epochs.

## H.2. Reproducing Evo-2 and LucaOne Baselines

We reproduce Evo-2 and LucaOne baselines on the Rice Subspecies Benchmark and Whole-Genome Phenotype Benchmark by using their released pretrained models as frozen feature extractors, followed by a lightweight supervised head.

**Feature extraction.**   Evo-2 and LucaOne use the pretrained Evo-2-7B and lucaone-gene models, respectively. Given an input rice DNA sequence, we chunk it into non-overlapping fragments with length 70,000 for Evo-2. We choose 70,000 to best utilize Evo-2's long-context modeling under our single-H100 setting, as the original million-token inference depends on multi-GPU implementations that are not publicly available, and 70,000 is the largest context length supported on an H100 GPU in our setup. For LucaOne, we follow the official default fragment length of 1,280. Each fragment is encoded into a feature vector, producing a sequence of feature embeddings with length approximately $\lceil L/l_{\text{frag}} \rceil$, where $L$ is the sequence length and $l_{\text{frag}}$ is the fragment length. We then apply mean pooling over fragment-level features to obtain a single fixed-dimensional representation per input sequence.

**Prediction head and evaluation.**   On top of the pooled representation, we train a single-layer linear head, following the same downstream head design as OpticalDNA for a fair comparison. We report the test RMSE corresponding to the checkpoint achieving the best validation RMSE.

**Hyperparameter tuning.**   During fine-tuning, we perform a grid search over batch size and learning rate only: batch size $\in \{16, 32, 64\}$ and learning rate $\in \{10^{-5}, 10^{-4}, 2 \times 10^{-4}, 10^{-3}\}$, yielding 12 configurations in total. All other

hyperparameters are fixed to epochs = 400 and weight decay = $10^{-5}$ to ensure sufficient convergence under limited fine-tuning capacity (i.e., a linear head on top of frozen features).

## I. Contamination Analysis and Generalization beyond Memorization

Given the large-scale pretraining and downstream genomic corpora used in this work, we further examine whether downstream test performance could be explained by train–test contamination or instance-level memorization. We conduct strict sequence-level overlap analyses for both the human eQTL benchmark and the rice benchmarks (RiceSubBench and RiceWGPB), and additionally analyze whether partial fragment overlap is sufficient to induce consistent downstream labels. Overall, the results show that direct contamination is negligible, and that partial sequence overlap is not a reliable indicator of functional equivalence.

### I.1. Strict Full-Sequence Overlap Analysis

We use a 100% full-sequence overlap criterion to assess potential train–test contamination. A test sequence is marked as matched only if the entire test sequence appears in the training data of the same benchmark task. This criterion directly tests whether a downstream test instance reappears in the corresponding training data, while avoiding the ambiguity of loose or local sequence similarity.

For the human eQTL benchmark, each test example is a 450 kb DNA sequence. As shown in Table S24, under the 100% full-sequence overlap criterion, only 12 out of 4,297 test sequences are matched, corresponding to a contamination rate of 0.279%. Thus, 99.721% of eQTL test sequences are novel under this strict criterion.

*Table S24.* Strict contamination analysis on the human eQTL benchmark. A match is counted only when the full 450 kb test sequence appears in the corresponding training set, following the 100% full-sequence overlap criterion.

| Metric | Count | Rate |
|---|---|---|
| Matched test sequences | 12 / 4,297 | 0.279% |
| Novel test sequences | 4,285 / 4,297 | 99.721% |

For the rice benchmarks, we apply the same 100% full-sequence overlap criterion. A rice test sequence is marked as matched only if the entire test sequence appears in the training data of the same benchmark task. As shown in the Table S25, under this criterion, the matched rate is also very low. For RiceSubBench, considering all rice subspecies classification tasks together, only 136 out of 27,822 test sequences are matched, corresponding to a contamination rate of 0.489%. Thus, 99.511% of RiceSubBench test sequences are novel. For RiceWGPB, the matched rates are similarly low for the two whole-genome phenotype prediction traits: 0.515% for leaf rolling index measured in the 2015 SZ environment (LRI-15SZ), and 0.471% for thousand grain weight (TGW).

*Table S25.* Strict contamination analysis on the rice benchmarks. A match is counted only when the full test sequence appears in the training data of the same benchmark task, following the 100% full-sequence overlap criterion. RiceSubBench aggregates all rice subspecies classification tasks, while RiceWGPB reports the two whole-genome phenotype prediction traits separately: leaf rolling index measured in the 2015 SZ environment (LRI-15SZ) and thousand grain weight (TGW).

| Metric | Count | Rate |
|---|---|---|
| Matched test sequences (RiceSubBench, all rice subspecies classification tasks) | 136 / 27,822 | 0.489% |
| Novel test sequences (RiceSubBench, all rice subspecies classification tasks) | 27,686 / 27,822 | 99.511% |
| Matched test sequences (RiceWGPB, LRI-15SZ) | 58 / 11,253 | 0.515% |
| Matched test sequences (RiceWGPB, TGW) | 78 / 16,569 | 0.471% |

We adopt this strict full-sequence criterion because it directly addresses the most relevant leakage question: whether a complete downstream test sequence is already present in the corresponding training data. This avoids conflating true train–test leakage with the inherent redundancy of genomic sequences, where short local motifs and partial overlaps are common even among biologically distinct regions.

## I.2. Why Full-Sequence Overlap Is Used Instead of Loose Similarity

A natural question is whether partial sequence overlap should also be treated as potential contamination. We argue that moderate fragment overlap is not a reliable criterion for leakage in genomic benchmarks. Due to the four-letter nucleotide alphabet and the recurrence of local motifs, partial overlap is common across genomic regions, but it does not imply that two sequences are functionally equivalent or share the same downstream label. Genomic function depends not only on local nucleotide composition, but also on order, spacing, orientation, broader sequence context, and long-range interactions.

To empirically examine this issue, we analyze RiceSubBench by grouping test samples according to their 20-mer containment with the training set. For each test sample, we compute the fraction of unique 20-mers that are shared with training samples, and then measure Top-1, Top-5, and Top-10 label consistency with the nearest training samples. If partial overlap were sufficient to indicate leakage or memorization, label consistency would increase monotonically with overlap and would become nearly perfect under high-overlap bins.

As shown in Table S26, this is not the case. Across containment bins from 0.0 to 0.9, Top-1 label consistency remains relatively flat, ranging from approximately 42% to 55%, without a clear monotonic trend. The largest bin, $[0.0, 0.1)$, contains 57.13% of test samples and achieves only 47.52% Top-1 consistency. Even high-overlap bins such as $[0.7, 0.8)$ and $[0.8, 0.9)$ remain around 49–55% Top-1 consistency. A sharp increase appears only near complete overlap: the $[0.9, 1.0)$ bin reaches 76.85% Top-1 consistency, while exact overlap occurs in only 2 samples, corresponding to 0.07% of the test set.

*Table S26.* Label consistency under different 20-mer containment levels on RiceSubBench. Test samples are grouped by the fraction of unique 20-mers shared with training samples. Partial overlap does not yield consistently high label agreement; a sharp increase appears only near complete overlap.

| k-mer Containment Bin | #Test Samples | Test Ratio (%) | Top-1 Consistency (%) | Top-5 / Top-10 Consistency (%) |
|---|---|---|---|---|
| $[0.0, 0.1)$ | 1,650 | 57.13 | 47.52 | 45.82 / 45.00 |
| $[0.1, 0.2)$ | 96 | 3.32 | 41.67 | 41.25 / 43.33 |
| $[0.2, 0.3)$ | 66 | 2.29 | 48.48 | 50.00 / 45.76 |
| $[0.3, 0.4)$ | 62 | 2.15 | 50.00 | 50.65 / 47.74 |
| $[0.4, 0.5)$ | 60 | 2.08 | 55.00 | 50.67 / 45.50 |
| $[0.5, 0.6)$ | 74 | 2.56 | 51.35 | 51.08 / 49.73 |
| $[0.6, 0.7)$ | 88 | 3.05 | 54.55 | 49.55 / 46.82 |
| $[0.7, 0.8)$ | 147 | 5.09 | 55.10 | 44.76 / 43.27 |
| $[0.8, 0.9)$ | 224 | 7.76 | 49.11 | 46.43 / 46.47 |
| $[0.9, 1.0)$ | 419 | 14.51 | 76.85 | 73.75 / 70.19 |
| $[1.0]$ | 2 | 0.07 | 100.00 | 60.00 / 55.00 |

These results indicate that only near-complete overlap yields substantially higher label consistency. Partial overlap, even when large, does not reliably determine the downstream label. Therefore, we use the 100% full-sequence overlap criterion as a strict and unambiguous contamination criterion, rather than treating moderate k-mer overlap as evidence of leakage.

## I.3. Generalization beyond Instance-Level Memorization

The above analyses suggest that OpticalDNA's downstream performance is unlikely to be primarily explained by instance-level memorization. First, the pretraining stage is fully unlabeled. Even if related genomic fragments appear during pretraining, the model is not exposed to downstream task labels, phenotype annotations, or eQTL supervision. Therefore, pretraining exposure alone cannot directly memorize the mapping from a downstream sequence to its task label.

Second, the k-mer containment analysis in Table S26 argues against a simple memorization explanation. If performance were dominated by recalling similar training instances, high-overlap regions would be expected to show near-perfect label consistency. Instead, label consistency remains far from perfect even under substantial overlap, and only near-complete overlap produces a clear increase. This indicates that partial sequence sharing is insufficient to explain the observed downstream performance.

Third, some degree of sequence sharing is unavoidable when genomic foundation models are pretrained and evaluated on related genomic corpora. This issue is not specific to OpticalDNA, but applies broadly to genomic foundation models trained on large-scale reference genomes or multi-species genomic collections. The key question is therefore not whether local

sequence overlap exists, but whether downstream performance is mainly driven by memorization. Our strict full-sequence overlap and k-mer containment analyses suggest that this is unlikely.

Finally, cross-species transfer provides an additional test of generalization. OpticalDNA pretrained on the human HG38 genome remains effective on rice downstream tasks, despite the large evolutionary distance and limited sequence similarity between human and rice (See Appendix O.1). Such transfer is difficult to explain by fragment-level recall alone, and instead suggests that OpticalDNA learns transferable genomic regularities and useful computational representations.

Taken together, while implicit exposure to related genomic patterns during pretraining cannot be completely ruled out, the evidence does not support a memory-driven explanation of the results. Direct train–test contamination is negligible under the 100% full-sequence overlap criterion, partial k-mer overlap does not reliably induce label consistency, and cross-species transfer further supports generalization beyond instance-level recall.

## J. Additional Ablation Results

### J.1. Downstream eQTL Ablation on DNALONGBENCH

To provide a detailed quantitative view of our ablation study, we report the per-tissue AUROC results on the DNALONG-BENCH eQTL tasks in Table S27. We directly compare DeepSeek-OCR with OpticalDNA under identical evaluation protocols. OpticalDNA consistently improves performance across all nine GTEx tissues and achieves a +5.37% relative gain on the overall average AUROC, demonstrating that our DNA-specific document formulation and pretraining yield more transferable representations for long-range functional genomics prediction.

*Table S27.* Detailed ablation results on DNALONGBENCH eQTL prediction (AUROC). The last row reports the relative improvement (↑) over DeepSeek-OCR.

| | CCF | WB | Thyroid | SNSES | SSELL | MS | NT | AT | AS | Average |
|---|---|---|---|---|---|---|---|---|---|---|
| DeepSeek-OCR | 0.815 | 0.922 | 0.750 | 0.814 | 0.810 | 0.819 | 0.873 | 0.841 | 0.763 | 0.823 |
| OpticalDNA | **0.819** | **0.927** | **0.876** | **0.931** | **0.832** | **0.877** | **0.900** | **0.852** | **0.788** | **0.867** |
| Improvement (↑) | 0.55% | 0.54% | 16.86% | 14.30% | 2.77% | 7.09% | 3.14% | 1.38% | 3.37% | 5.37% |

## K. Evaluating DNA Document Understanding Capabilities

### K.1. Length-wise OCR Transcription Analysis (T1)

We analyze the T1 OCR transcription capability of OpticalDNA-Rice across different retained sequence lengths to evaluate whether single-page image-to-text DNA transcription remains reliable as more DNA text is rendered within a fixed canvas. As shown in Table S28, OpticalDNA-Rice maintains strong transcription quality for short and moderate spans: for retained lengths below 1024 bp, EM remains between 75.0% and 85.9%, while CS stays high between 93.4% and 99.7%. Performance gradually decreases beyond 1024 bp, with a more pronounced drop near the single-page capacity, where EM/CS decreases to 65.5%/83.4% for [1536, 1792] and 42.3%/76.4% for [1792, 2048]. These results suggest that OpticalDNA-Rice reliably performs T1 OCR transcription under moderate sequence density, while very long retained spans become more challenging due to denser rendered text.

*Table S28.* Length-wise T1 OCR transcription performance of OpticalDNA-Rice. Validation samples are grouped by retained sequence length. EM and CS denote Exact Match and character similarity, respectively.

| Retained length | #Sample | EM (%) ↑ | CS (%) ↑ |
|---|---|---|---|
| < 256 | 104 | 75.0 | 99.7 |
| [256, 512) | 144 | 77.8 | 95.7 |
| [512, 768) | 135 | 85.9 | 96.6 |
| [768, 1024) | 145 | 77.9 | 93.4 |
| [1024, 1280) | 140 | 72.1 | 89.0 |
| [1280, 1536) | 122 | 71.3 | 87.3 |
| [1536, 1792) | 148 | 65.5 | 83.4 |
| [1792, 2048] | 137 | 42.3 | 76.4 |

## K.2. ROI Identification Metrics and Evaluation Details (T2)

We provide additional details for evaluating ROI identification on T2 (*Grounding + Recognition*). For each dataset (HG38 and rice), we randomly sample 1,000 validation examples from the corresponding validation split of the pretraining corpus and evaluate OpticalDNA variants pretrained on the same domain. To improve input diversity and match the transcription robustness protocol, we further apply random tail truncation by removing up to 90% of each DNA document and evaluate the remaining single-page input. Each example contains multiple line-level ROIs, where each line is paired with a nucleotide transcription and a detection box in the format $[p, x_1, y_1, x_2, y_2]$. We adopt a strict ordered evaluation setting: the $i$-th predicted line is aligned with the $i$-th ground-truth line, so line-count consistency is necessary to avoid shifted correspondences caused by insertion/deletion errors.

**Metrics.** Let a page contain $N$ ground-truth lines with transcriptions $\{Y_i\}_{i=1}^N$ and boxes $\{b_i\}_{i=1}^N$, and let the model output $\{\hat{Y}_i\}_{i=1}^{\hat{N}}$ and $\{\hat{b}_i\}_{i=1}^{\hat{N}}$. All metrics below are computed under ordered alignment.

**(1) Line Count Match (LCM).**

$$\text{LCM} = \mathbb{I}[\hat{N} = N], \tag{S8}$$

where $\mathbb{I}[\cdot]$ is the indicator function.

**(2) Text-only correctness (Text-EM).**

$$\text{Text}_i = \mathbb{I}[\hat{Y}_i = Y_i]. \tag{S9}$$

**(3) IoU-based box matching (Det-Acc).** We use an extremely strict localization criterion IoU$= 0.99$. Given two boxes $b = [p, x_1, y_1, x_2, y_2]$ and $\hat{b} = [\hat{p}, \hat{x}_1, \hat{y}_1, \hat{x}_2, \hat{y}_2]$, we compute

$$\text{IoU}(b, \hat{b}) = \frac{\text{Area}(b \cap \hat{b})}{\text{Area}(b \cup \hat{b})}, \tag{S10}$$

and define per-line detection correctness as

$$\text{Det}_i = \mathbb{I}\big[\text{IoU}(b_i, \hat{b}_i) \geq 0.99\big]. \tag{S11}$$

We also report the average IoU across aligned lines as a complementary continuous measure.

**(4) Average IoU (Det-IoU(avg)).** In addition to the thresholded detection correctness, we report the average IoU as a continuous localization quality measure:

$$\text{Det-IoU(avg)} = \frac{1}{N} \sum_{i=1}^N \text{IoU}(b_i, \hat{b}_i), \tag{S12}$$

where IoU is computed for every ordered-aligned box pair regardless of whether it exceeds the threshold. Higher values indicate better overall box overlap, with $\text{Det-IoU(avg)} \to 1$ corresponding to near-perfect localization.

**(5) Joint correctness.**

$$\text{Joint}_i = \text{Text}_i \cdot \text{Det}_i. \tag{S13}$$

**(6) Line Joint Accuracy (Joint).**

$$\text{LineJointAcc} = \frac{1}{N} \sum_{i=1}^N \text{Joint}_i. \tag{S14}$$

**(7) Page Strict Accuracy (Strict).**

$$\text{PageStrictAcc} = \mathbb{I}[\hat{N} = N] \cdot \prod_{i=1}^N \text{Joint}_i. \tag{S15}$$

**(8) Coordinate $\ell_\infty$ error ($\ell_\infty$ Err.).**

$$\|\hat{b}_i - b_i\|_\infty = \max\big(|\hat{x}_1 - x_1|, |\hat{y}_1 - y_1|, |\hat{x}_2 - x_2|, |\hat{y}_2 - y_2|\big). \tag{S16}$$

We report its average over valid line pairs.

All reported numbers are averaged across the 1,000 sampled pages for each corpus.

*Table S29.* Full ROI identification metrics on T2 under strict ordered alignment (IoU= 0.99).

| Model | LCM ↑ | Text-EM ↑ | Det-Acc ↑ | Det-IoU(avg) ↑ | Joint ↑ | Strict ↑ | $\ell_\infty$ Err. ↓ |
|---|---|---|---|---|---|---|---|
| OpticalDNA-HG38 | 0.9871 | 0.9867 | 0.9875 | 0.9963 | 0.9820 | 0.8763 | 0.3941 |
| OpticalDNA-Rice | 0.9991 | 0.9692 | 0.9891 | 0.9989 | 0.9691 | 0.8084 | 0.3243 |

**Evaluation results.** Table S29 reports the complete set of ROI identification metrics for T2, where OpticalDNA-HG38 and OpticalDNA-Rice denote the OpticalDNA variants pretrained on the HG38 and rice corpora, respectively, and evaluated on 1,000 sampled validation examples from the corresponding domain. Under strict ordered alignment and IoU= 0.99, both variants achieve high text accuracy and precise ROI localization. OpticalDNA-HG38 attains stronger line-level joint correctness (0.9820) and higher page-level strict success (0.8763), while OpticalDNA-Rice shows near-perfect line-count match (0.9991) with competitive line-level joint accuracy (0.9691). Both models achieve near pixel-level localization error ($\ell_\infty \leq 0.3941$ px), confirming highly accurate ROI boundary recovery under an extremely strict matching threshold.

Moreover, Figure S2 visualizes representative OCR predictions of OpticalDNA on rendered DNA pages, covering (a) short, (b) medium, and (c) long text regions. In each panel, the black string denotes the underlying DNA sequence on the page, while each colored bounding box indicates a localized ROI whose colored transcription is the OCR output for that region. Despite the dense layout and varying region lengths, OpticalDNA produces consistently accurate and well-aligned transcriptions across all three cases, demonstrating strong structure-aware reading and precise region-level decoding at the nucleotide level.

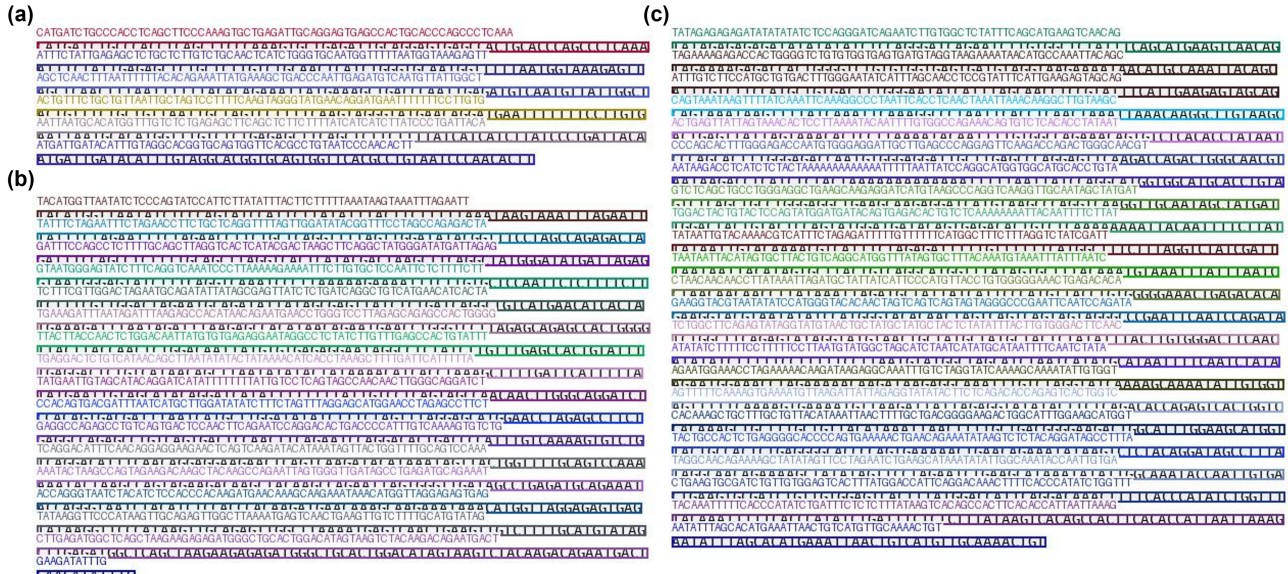

*Figure S2.* Qualitative OCR examples of OpticalDNA on DNA documents. Black text shows the original DNA sequence rendered on the page. Colored boxes denote predicted ROIs, and the colored strings are OCR transcriptions produced for the corresponding regions. We show (a) short, (b) medium, and (c) long text regions, illustrating accurate region-level transcription and alignment across diverse lengths.

### K.3. ROI-based DNA Transcription Metrics and Evaluation Details (T3)

We provide additional details for evaluating ROI-based DNA transcription on T3 (*Recognition (ROI)*). Unlike T2, where the model must both localize ROIs and transcribe them, T3 provides the ROI boxes as part of the prompt and requires the model to transcribe the DNA content within each box *in order*. Concretely, given a set of boxes $\{b_i\}_{i=1}^N$ on a DNA document page, the model outputs a list of transcriptions $\{\hat{Y}_i\}_{i=1}^{\hat{N}}$, where each $\hat{Y}_i$ is a predicted DNA base sequence (a string over $\{A,T,C,G\}$) corresponding to the content within box $b_i$.

**Evaluation protocol.** We follow the same sampling protocol as T2: for each dataset (HG38 and rice), we randomly sample 1,000 validation examples from the corresponding validation split, apply random tail truncation up to 90% to increase input

diversity, and evaluate the remaining single-page input. Since T3 shares the same ordered multi-ROI output format as T2, we reuse the T2 evaluation metrics to ensure consistent comparison across tasks. For completeness, we refer readers to Appendix K.2 for the full metric definitions (ordered alignment, text correctness, IoU-based box matching, average IoU, joint correctness, strict page success, and $\ell_\infty$ error).

**Instantiated metrics for T3.** Because boxes are given in T3, the localization-related terms are expected to be trivially satisfied if the model faithfully copies the provided boxes into the output (thus Det-Acc$= 1$ and Det-IoU(avg)$= 1$ with $\ell_\infty = 0$). Therefore, the key source of errors in T3 is transcription quality under multi-ROI decoding, which is captured by (i) line-level transcription exact match (**Text-EM**), (ii) line-level character error rate (**Joint**), and (iii) strict page success (**Strict**) that requires all ROIs on a page to be transcribed correctly.

**Evaluation results.** Table S30 reports in-domain T3 results for OpticalDNA pretrained on HG38 (*OpticalDNA-HG38*) and rice (*OpticalDNA-Rice*), each evaluated on 1,000 sampled validation pages from the corresponding corpus. As expected, providing boxes makes localization trivial (Det-Acc$= 1.0$, Det-IoU(avg)$= 1.0$, and $\ell_\infty = 0$), and performance is therefore dominated by ROI-level transcription accuracy. OpticalDNA-HG38 achieves higher line-level transcription exact match (0.8480) and strict page success (0.7507), while OpticalDNA-Rice remains competitive but shows a larger drop under the strict all-ROI criterion (0.5498), consistent with error accumulation across multiple ROIs per page.

*Table S30.* Full ROI-based transcription metrics on T3 under strict ordered alignment (IoU$= 0.99$).

| Model | LCM ↑ | Text-EM ↑ | Det-Acc ↑ | Det-IoU(avg) ↑ | Joint ↑ | Strict ↑ | $\ell_\infty$ Err. ↓ |
|---|---|---|---|---|---|---|---|
| OpticalDNA-HG38 | 1.0000 | 0.8480 | 1.0000 | 1.0000 | 0.8480 | 0.7507 | 0.0000 |
| OpticalDNA-Rice | 1.0000 | 0.7116 | 1.0000 | 1.0000 | 0.7116 | 0.5498 | 0.0000 |

Moreover, Figure S3 visualizes representative ROI-based OCR predictions of OpticalDNA on rendered DNA pages for T3. We show (a) short, (b) medium, and (c) long ROI transcripts. In each panel, the black string denotes the underlying DNA sequence rendered on the page, while each colored bounding box indicates a given ROI and the colored string is the OCR transcription produced for that ROI *in order*. Despite large variation in ROI length and dense page layouts, OpticalDNA generates accurate and well-aligned ROI transcriptions, demonstrating reliable region-level reading under ordered multi-ROI decoding.

### K.4. Subsequence Locate: Metrics and Evaluation Details (T5)

We provide additional details for evaluating T5 (*Subsequence Locate*), a retrieval-style grounding task that shares the same ordered structured output schema and evaluation pipeline as T2. In T5, the model is given a query DNA subsequence and is required to retrieve and localize its matched occurrence(s) on a rendered DNA page by predicting bounding boxes. Accordingly, T5 evaluates two coupled aspects: (i) whether the model preserves the queried subsequence ("what to find"), and (ii) whether it grounds the correct region(s) ("where to find").

**Evaluation protocol.** We follow the same sampling and evaluation implementation as T2. Since T5 uses the same ordered multi-instance output schema as T2, we reuse the full T2 metric suite for consistent comparison; full definitions are provided in Appendix K.2.

**Instantiated metrics for T5.** Unlike T3, where boxes are given and localization becomes trivial, T5 requires *joint* subsequence retrieval and localization under potentially repeated occurrences. Therefore, we report the same core metrics as T3 to enable aligned analysis across tasks: (i) line count match (**LCM**), (ii) text correctness (**Text-EM/CER**), (iii) localization quality (**Det-Acc@IoU** and **Det-IoU(avg)**), (iv) line-level joint success (**Both-Acc**), (v) strict page-level success (**Strict-All-Both**), and (vi) geometric error measured by coordinate $\ell_\infty$ distance (**Det-Linf**). In addition, we report both concatenated-text metrics and per-line metrics to disambiguate errors due to instance ordering versus instance-wise transcription.

**Results and discussion.** Table S31 reports T5 performance of OpticalDNA on HG38 and rice. Overall, T5 remains challenging: the model achieves moderate query faithfulness (Text-EM $\approx 0.57$–$0.60$), while localization accuracy drops substantially as the IoU threshold increases, indicating that subsequence-to-box grounding is the dominant bottleneck. In

**(a)**

```
CTAGTTCTATTAAGGGTAGGATTTGGTGTTTGAGTTTTGAAGCTCAGCCCCAAAATTGAGATGTATCTT
TAGTGTATAAGGCACCTTGTACCCCAACTCCAGGAGCCCTAGGTTTGCACCTTCTGAATGCCGCAGTGCAG
CAGCCCAGGTATCTCGTCACTTTCTTGCCAAGCATCTTGCATTGCTTTGCTTTGCTCTATATTCTTGCAGATAGAT
TTGCTTCATGAATAAAGCCCTTAGGCGACCTTGCCGTGCCCTCTCTAAGCAATCTGGATTAATTTAACAGCC
TCATCGAAAGGGATACTCTTTACAGGGTATTTCTGCAAAAAACAGTCAAGATAAACCAATGTTTGAACA
TGTTAACCACATCAATTGATCCCAGACACATCACATGCTTACTTCAAACACAATCCTGCTAAGGGTCCCC
ACGGCTGATCATGAGTGTGAGAACCGCTGAACACTGT
```

**(b)**

```
TTAAGAATTTACTGGCATCAGCGCTTTCTGTGCAATCCCAGTGGCTGGCAGGACACTTCCCTTCCAGGCA
GTTCTCCTGCATTCATCTGCAGGGTGATACACTGAGGGATGACTGGAATTAAGAAACTAAACACGGCCAG
GTGCAGTGGGTGGCTCACATCTCTAATCCCAGCCTGTAAGTGCAGCCTTTGAGTTCTTTGACTCTCCATC
TCCTCCAGGTGAGATTAGACTGGCCTAGTAACTCACTTATAGCCGATAGAATAAGCCATAAGTGATAGGC
TAGGTAGAAAAAGTTCTTCCTCCTTGGACACTTGCCTGAAAAGCACAACTACTCTGAGATTGCCATGTGA
GAGAGGACCCGGCCACCAGCTCCACCAGCCAGCCTTCTGAGCGAGCTTTGAGTGAGCCATCCGGGATGGC
AGCCCAGTTGAGTCTTCAAACAATGGAACCTCAGCCAACATCTGCCAGCAGCAGCATGAAGGAGACCCCA
AGTGAGAACAGCACCAGCTATGCCTTCTTTTTTTTTTTTAATGGAGTGTCACTCTGTCGCCCAGGCTGGAG
TGCAGTGGCACCATGTGGCTCACTGCACCCTTTATCTCCTGGGTTCAAGCAATTCTCCTGCCTCAGCCTCC
TGAGGAGCTGGGATTACAGGCGCCCACCACCATGCCCAGCTAATTTTTGTTTTTTTTTTTTTTAGTAGA
TACCGGGTTTCACCATGTTGGCCAGGCTGGTCTCGAACTCCTGACCTCAGATGATCCACCCACCTCAGCCT
CCCAAAGTGCTGGGATTACAGGCGTGAGCCACTGTGCCCGGCTTTTTTTTTTTTTTTTTTTTTTTTGTCAAGG
TCTGTATTAGTCAGGGTTCTCTAGAGGGACAGAACTAATTACATGTATATGTATAATGTTTGTTTTTGT
TTTTGTTTTTGTTTTTTTTGAGACAGGGTGTGTATTAGTCAGAGTTCTCTAGAGGGACAGGACTAATAG
GATAGATGTATATATGAAGGGGAAATTGATTAAGGAGTATTGACTCACACAATCACGAGGTGAGGTCCCA
CAATAGGCCGTCTGCAAGCCAAGGATCAAGGAAGCCAGTCCAAGTCCCAAAACCTCAAAAGTAGGGAAGC
CGACAGTGCAGCCTTCAGTCTGTGGTCAAAGGTCCAAGAGTCCCAAAGCTAAAGAACTTGGAGTCTGATG
CTTGAGGTCAGGAAGCATCTAGCACAGGAGAAAGATGGAGGCCAGAAGACTCAGCCAGTCTAGTCTTTCC
ACATTCCTCTACCTGCT
```

**(c)**

```
TACTCCAGATTTCAGACAAATGTAGTGTGTTCTGTTGTCTGGCTTCTTTCACTCTGCTTGTTTTTGGAA
CTCATCCATGTTGTTGCATGCATCAGTGATTTGAAGAACTATGAACTTAGGAGTAAGAAGCAGATAAGT
ATGATGTTGGTAATATAAGTATCAAGGGGCCGTTCTTATTTGGCTTTCTGAGAAACCTGACCTTTATACT
AGTCTACCTGAAGCATGTATTTACCTGAGAGGAAATCGGGACTTGGTAGCAGAAGAAAAGCTCAGGGAAAG
ACCAAGGCACACAGGTTTTCCTCACATAATGACTGTGGCATTAACTCTTCAACATAACGGCTCTGGAAAC
ACATTTTGGACAATGGCTTTTTTTTCAGTACAATATCCTTTTATCCAAAAAGCACACTAAATAAATAT
GCTAGAAGCAGTGATATTTTGACATCCAAGGCAGGCAAAACCACACTCAGAATTTAGTTGACATAACAG
TGTCTCACCAAAGAATATCATATGAAACAGACCTGAACTGTGCACCTGTGCACCTTGTTTGCACAGACTC
AGTGTGCCTTGCAAATGTGAACCCGGCGAGAGAGAGAAGGAGCCAGCACCTTCTATAGCTCTCAGTTTCT
TGTACATTTGTGCTATTCCCACACTGCCAAGCACTCTTGCGCTCAGCCTTTATTTTTCTCTAGCAATGTT
AATCTGTATATAGCTTAAAGCATCATCTTTGCAGGCAAAATAGTTGTCAGCCCAAAGGTTGAAAAATGGGT
GAAGTAAACATGAAGAAAGAATGAGCCTTCCAAAGAATGATGCCCTCCCTCAACTACTCTTGCTCCTTTG
GTGAGTTCCAAAAACCTTTATTACCTTGCACTTTATGTACAAACTAGATTCCTCTACCTTTGTGTTGTT
GTTTATTACTTGCATTACTTACTTTTATTGTTCAATGGATAGAGTTTCCTACCTGTGAAAATTATCTG
AGTATCTTGGTTTGGAGGACAATGCCTCATAATTTTTTTCCATTAAAATTAATGGGAAAAATTTCTTT
CATTAAATGACTTTGTCCTTAACAGCAAGAATTTCAGGAATGAATTGGGTGGTTAAACAGGGATAGATA
GGTGGTATCAGACTTCTGACTTTCAAAAATGGGATTTTGCACCATGGCAGTTCTCTTGAATTGTTTTAA
GTCCTTTTGTCACTCCTGTGGCATGTAAATTGGGTATTTACTTAAGTAAATGCAATGCAGATGGAAAGG
AGACCCACACAGAGGTAAGAAGGAAAGGGGGAGGAGGAAAAGTCTAAGATGCTCAGAAATGGCAATTTCC
TCTCACAGGCTTTGGCTTAAAGAAGGAGGGGTGGATCTGGAACTCCGCCTGCCCTGTGGCAACCTGGGCAT
CAGCTACACTGAAGAAGGCCAGGCAAGGCCAGAATCTGGGTAGTCCAGGCACTTAACTTTCGCTCAGATT
CCACACGGAATCGGCTCCTGGCTGGAGCTCAGGGGCCTCAGGACGGGCCCAGGCAGGGACACTTCTGGACTCTGGAGGTTCAGG
TTCTGCCCCAATTTGTGGGAGTTGAGGTGCGGGCCCCAGGCAGGGACACTTCTGGACTCTGGAGGTTCAGG
ATCCTAAGTTACACCTGCCACAGACTTCAAAGGACATTTCCTGAGGGACCCAGGGGGGAAGGACCTCGCCT
CAAGGACCCTGCCTCCTAGGCAAGTCCCCAGAGGCCCCCAGTGACTCCAAGGGCTCCAGATGCCAAAGCA
GAGGGACCATCAAAGGGAAACACTAGGACTCCCTCAGCAAAGTCCAGACTCTGGGGGCTCTGCAGGACAAG
GGACC
```

*Figure S3.* Qualitative ROI-based OCR results of OpticalDNA on DNA documents for T3. Black text indicates the rendered ground-truth DNA sequence, while colored boxes denote the *given* ROIs specified in the prompt. Colored strings show the corresponding OCR transcriptions predicted *in order*. We include (a) short, (b) medium, and (c) long ROIs, demonstrating accurate region-level transcription across diverse ROI lengths.

*Table S31.* Full T5 metrics under strict ordered alignment at multiple IoU thresholds $\tau$. We report instance count match (LCM), query faithfulness (Text-EM), localization (Det-Acc@IoU$\geq \tau$, Det-IoU(avg)), joint success (Both-Acc, Strict-All-Both), and $\ell_\infty$ coordinate error.

| Model | $\tau$ | LCM ↑ | Text-EM ↑ | Det-Acc@IoU ↑ | Det-IoU(avg) ↑ | Both-Acc ↑ | Strict ↑ | $\ell_\infty$ Err. (px) ↓ |
|---|---|---|---|---|---|---|---|---|
| OpticalDNA-HG38 | 0.50 | 0.976 | 0.602 | 0.0462 | 0.0442 | 0.0240 | 0.0240 | 184.6184 |
|  | 0.90 | 0.976 | 0.602 | 0.0129 | 0.0442 | 0.0083 | 0.0083 | 184.6184 |
|  | 0.95 | 0.976 | 0.602 | 0.0074 | 0.0442 | 0.0046 | 0.0046 | 184.6184 |
|  | 0.99 | 0.976 | 0.602 | 0.0055 | 0.0442 | 0.0037 | 0.0037 | 184.6184 |
| OpticalDNA-Rice | 0.50 | 0.8428 | 0.5721 | 0.6800 | 0.5782 | 0.3926 | 0.3926 | 87.8245 |
|  | 0.90 | 0.8428 | 0.5721 | 0.2279 | 0.5782 | 0.1870 | 0.1870 | 87.8245 |
|  | 0.95 | 0.8428 | 0.5721 | 0.0995 | 0.5782 | 0.0902 | 0.0902 | 87.8245 |
|  | 0.99 | 0.8428 | 0.5721 | 0.0623 | 0.5782 | 0.0605 | 0.0605 | 87.8245 |

particular, the text metrics remain invariant across $\tau$, suggesting that the primary failure is in *where to find* rather than *what to find*.

A key difficulty of T5 is its multi-instance retrieval nature. The queried subsequence can occur multiple times on the same page, resulting in variable instance counts and requiring accurate multi-instance localization under ordered alignment. This is reflected by the gap between LCM and detection/joint metrics: even when the predicted instance count is often correct, missing matches, producing extra boxes, or small coordinate offsets can substantially reduce Det-Acc and Both-Acc, especially at high IoU thresholds.

Despite low detection scores under strict thresholds, the non-trivial text correctness and partial recovery of matched occurrences suggest that the visual encoder captures fine-grained representations supporting subsequence-level retrieval to some extent. A practical direction to strengthen T5 is to reduce supervision variance by controlling query specificity (e.g., constraining subsequence lengths or bounding occurrence counts), which simplifies the task by reducing multi-instance ambiguity; another orthogonal direction is to scale model capacity to improve fine-grained grounding.

### K.5. Mask Completion: Metrics and Evaluation Details (T4)

We provide additional details for evaluating T4 (*Mask Completion*), a context-based DNA understanding task. In T4, the model is given a DNA document (potentially multi-page) with masked regions and is required to infer the missing nucleotide strings from surrounding context. The model predicts the masked DNA for each queried box *in order*.

**Evaluation protocol (no tail truncation).** We randomly sample 1,000 validation examples from each corpus (HG38 and rice) and evaluate the corresponding pretrained OpticalDNA variant *in-domain*. Following the T2 evaluation implementation, we adopt strict ordered alignment between predictions and ground truth. For multi-page inputs, we enable document-level aggregation and perform prediction after merging pages via the multi-page fusion module. Unlike T2/T3, we do not apply random tail truncation for T4, because mask completion is inherently context-dependent and requires sufficient surrounding DNA evidence. All metrics are averaged over the sampled examples.

**Metrics.** Since T4 shares the same structured outputs and ordered evaluation setting as T2, we reuse the T2 metric suite, including **LCM** (line count match), **Text-EM** (exact match of predicted masked strings), **Det-Acc@IoU** and **Det-IoU(avg)** (localization correctness), **Both-Acc** (joint success requiring both correct text and correct box), **Strict** (page-level strict success requiring all instances to be jointly correct), and $\ell_\infty$ **Err.** (coordinate error). We refer readers to Appendix K.2 for the formal definitions of IoU, Both-Acc, Strict, and $\ell_\infty$ error. To better capture partial correctness in mask completion beyond exact matches, we additionally report **Text-CER**, defined as the average character error rate (normalized edit distance) between the predicted and ground-truth masked strings under ordered alignment (lower is better).

For each sample, we compute CER for each ordered-aligned pair $(Y_i, \hat{Y}_i)$ and average across all ground-truth instances in the sample; we then average the resulting value across samples:

$$\text{Text-CER} = \mathbb{E}_{\text{sample}} \left[ \frac{1}{N} \sum_{i=1}^{N} \frac{\text{ED}(Y_i, \hat{Y}_i)}{|Y_i|} \right], \tag{S17}$$

where $\text{ED}(\cdot, \cdot)$ denotes Levenshtein edit distance (insertions, deletions, substitutions) and $|Y_i|$ is the length of the ground-truth masked string. Lower is better.

**Main results.** Table S32 reports T4 mask completion results on HG38 and rice. Both OpticalDNA variants achieve perfect instance count match (LCM= 1.0) with perfect localization metrics (Det-Acc= 1.0, Det-IoU(avg)= 1.0, and $\ell_\infty$ error = 0), demonstrating stable structured outputs and reliable box reproduction. On the completion side, the models obtain Text-EM around 0.13 and Text-CER around 0.58, indicating that exact recovery of masked nucleotide spans remains challenging. The strict page success is lower (Strict $\approx$ 0.05), reflecting error accumulation when multiple masked regions must be completed correctly within the same sample. Overall, masked subsequence recovery is closely tied to both the available contextual evidence and the length of the target span, making this task intrinsically challenging. Nevertheless, the non-zero completion accuracy and consistent behavior across corpora suggest that OpticalDNA can leverage surrounding context to recover masked bases to a measurable extent.

*Table S32.* Full T4 metrics under strict ordered alignment without tail truncation. All detection-related metrics use IoU= 0.99.

| Model | LCM ↑ | Text-EM ↑ | Text-CER ↓ | Det-Acc ↑ | Det-IoU(avg) ↑ | Both-Acc ↑ | Strict ↑ | $\ell_\infty$ Err. ↓ |
|---|---|---|---|---|---|---|---|---|
| OpticalDNA-HG38 | 1.0000 | 0.1347 | 0.5803 | 1.0000 | 1.0000 | 0.1347 | 0.0490 | 0.0000 |
| OpticalDNA-Rice | 1.0000 | 0.1272 | 0.5789 | 1.0000 | 1.0000 | 0.1272 | 0.0480 | 0.0000 |

**Span-based ablation: longer masked spans require richer context.** The difficulty of T4 is strongly affected by the masked span length. To quantify this effect, we stratify evaluation by fixing line_span_min_n_base = line_span_max_n_base $\in \{1, 2, \ldots, 8\}$ and re-evaluate under the same protocol. Table S33 reports the results using the same metric set as the main table.

Across both OpticalDNA variants, performance degrades consistently as the span increases: Text-EM and Both-Acc drop rapidly from span = 1 to longer spans, while Text-CER increases, indicating more frequent base-level prediction errors and stronger error accumulation for longer completions. Since localization remains perfect throughout the ablation, this trend highlights the growing reliance on richer contextual evidence when recovering longer masked subsequences.

*Table S33.* Span-based ablation on T4 (IoU= 0.99; no tail truncation).

| Setting | LCM ↑ | Text-EM ↑ | Text-CER ↓ | Det-Acc@IoU ↑ | Det-IoU(avg) ↑ | Both-Acc ↑ | Strict ↑ | $\ell_\infty$ Err. ↓ |
|---|---|---|---|---|---|---|---|---|
| *OpticalDNA-Rice-HG38* | | | | | | | | |
| span=1 | 1.0000 | 0.5483 | 0.4517 | 1.0000 | 1.0000 | 0.5483 | 0.3310 | 0.0000 |
| span=2 | 1.0000 | 0.2815 | 0.5498 | 1.0000 | 1.0000 | 0.2815 | 0.1340 | 0.0000 |
| span=3 | 1.0000 | 0.0898 | 0.5951 | 1.0000 | 1.0000 | 0.0898 | 0.0370 | 0.0000 |
| span=4 | 1.0000 | 0.0602 | 0.6177 | 1.0000 | 1.0000 | 0.0602 | 0.0220 | 0.0000 |
| span=5 | 1.0000 | 0.0195 | 0.6197 | 1.0000 | 1.0000 | 0.0195 | 0.0090 | 0.0000 |
| span=6 | 1.0000 | 0.0187 | 0.6111 | 1.0000 | 1.0000 | 0.0187 | 0.0080 | 0.0000 |
| span=7 | 1.0000 | 0.0100 | 0.6102 | 1.0000 | 1.0000 | 0.0100 | 0.0070 | 0.0000 |
| span=8 | 1.0000 | 0.0077 | 0.6053 | 0.9997 | 0.9999 | 0.0077 | 0.0060 | 0.0167 |
| *OpticalDNA-Rice* | | | | | | | | |
| span=1 | 1.0000 | 0.5597 | 0.4403 | 1.0000 | 1.0000 | 0.5597 | 0.3510 | 0.0000 |
| span=2 | 1.0000 | 0.2423 | 0.5775 | 1.0000 | 1.0000 | 0.2423 | 0.1060 | 0.0000 |
| span=3 | 1.0000 | 0.0962 | 0.6019 | 1.0000 | 1.0000 | 0.0962 | 0.0380 | 0.0000 |
| span=4 | 1.0000 | 0.0425 | 0.6066 | 1.0000 | 1.0000 | 0.0425 | 0.0170 | 0.0000 |
| span=5 | 1.0000 | 0.0195 | 0.6197 | 1.0000 | 1.0000 | 0.0195 | 0.0090 | 0.0000 |
| span=6 | 1.0000 | 0.0072 | 0.6070 | 1.0000 | 1.0000 | 0.0072 | 0.0010 | 0.0000 |
| span=7 | 1.0000 | 0.0022 | 0.5975 | 1.0000 | 1.0000 | 0.0022 | 0.0010 | 0.0000 |
| span=8 | 1.0000 | 0.0013 | 0.6057 | 1.0000 | 1.0000 | 0.0013 | 0.0010 | 0.0000 |

**Discussion and outlook.** T4 serves as a direct probe of nucleotide-level contextual reasoning in the DNA document setting. Conceptually, it parallels masked language modeling (MLM) by requiring reconstruction of masked nucleotide spans from surrounding context, while operating under our region-aware structured generation format. The span-based ablation further shows that completion quality is highly sensitive to the available context and the target span length, highlighting the importance of stronger context utilization for reducing base-level reconstruction errors. These observations provide actionable insights for future pretraining of *vision-based masked models* on DNA documents, including mask-span curricula, context-aware sampling strategies, and more scalable reconstruction objectives over rendered genomic layouts.

Overall, T4 complements transcription- and grounding-style evaluations by emphasizing masked base recovery, and offers a principled pathway to extend OpticalDNA toward stronger base-level DNA understanding and reconstruction.

### K.6. Chromosome Classification (T6)

**Task and evaluation.** T6 evaluates chromosome classification from a DNA subsequence, where the model predicts the chromosome label of the input region. We report classification accuracy (*Acc.*). Following the T4 setting (Appendix K.5), we do not apply tail truncation and perform prediction with multi-page fusion to fully utilize the 2,048-base context.

**Results and discussion.** As shown in Table S34, OpticalDNA achieves 0.062 classification accuracy when pretrained on HG38 and 0.104 when pretrained on rice. This suggests that chromosome-level identification remains challenging under 2,048-base inputs, since local patterns and repeat-derived motifs can recur across chromosomes, making short contexts insufficiently distinctive for reliable chromosome assignment. Consistent with prior analyses of $k$-mer redundancy in repeat-rich eukaryotic genomes (Li et al., 2014), these results motivate future work on longer-context modeling toward chromosome-scale prediction.

*Table S34.* Chromosome classification results on T6. We report classification accuracy (*Acc.*).

| Model | Acc. ↑ |
|---|---|
| OpticalDNA-HG38 | 0.062 |
| OpticalDNA-Rice | 0.104 |

### K.7. Grad-CAM Interpretability on Page-Fused Visual Tokens

**Grad-CAM via multi-page fusion.** We visualize OpticalDNA's evidence localization using Grad-CAM on the *multi-page fusion* representations. Given a multi-page DNA document, we encode each page into a 2D grid of visual tokens, apply the

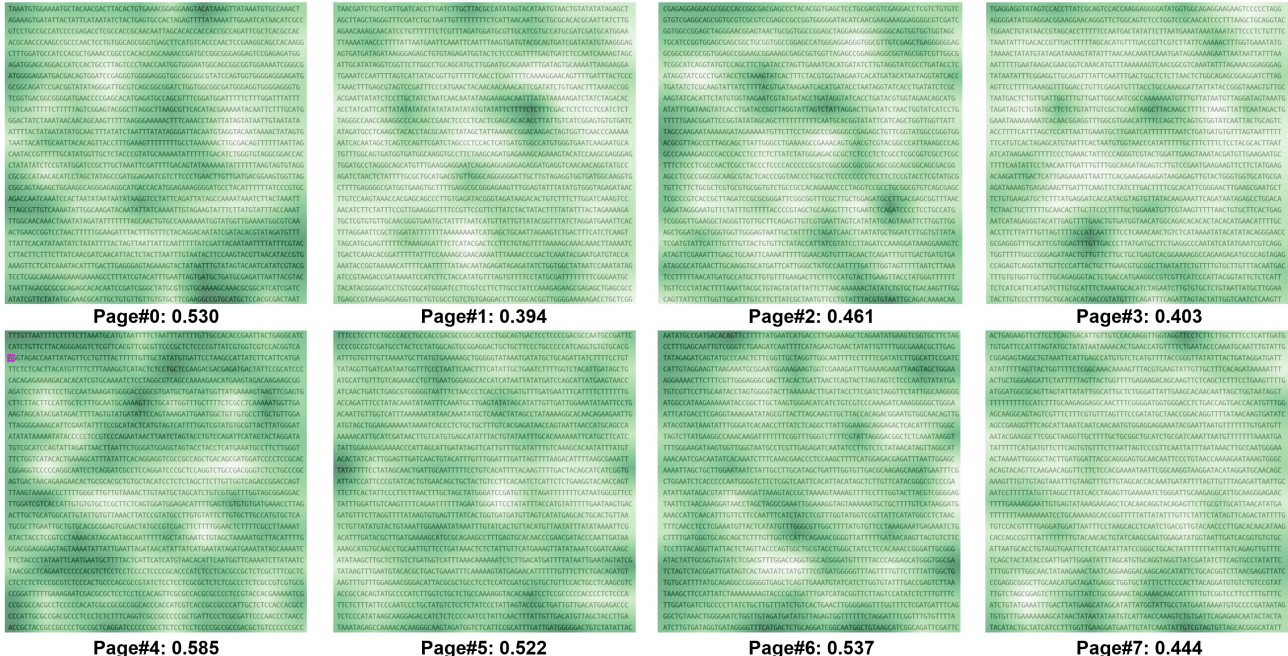

*Figure S4.* Grad-CAM visualization on the first 8 pages of a Japonica test document (Case 1) for SPLICE_SITE_3CLASS. Purple boxes indicate the splice-site locus; the number under each page is the page-level mean heatmap score.

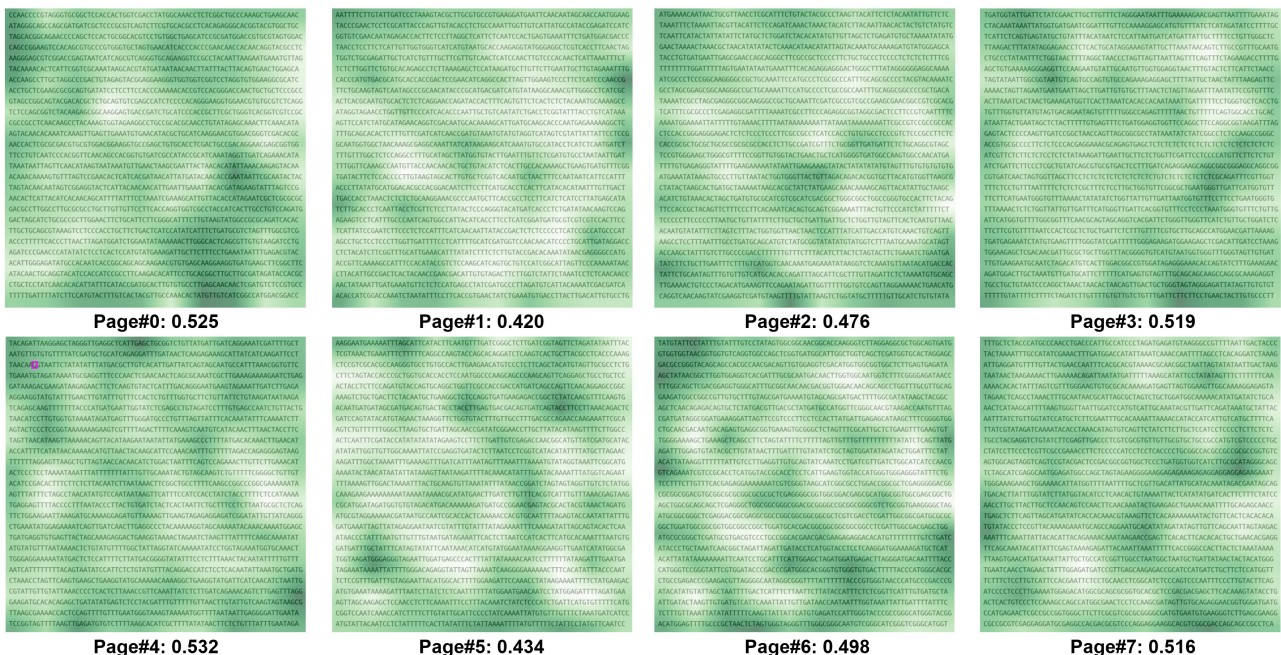

*Figure S5.* Grad-CAM visualization on the first 8 pages of a second Japonica test document (Case 2) for SPLICE_SITE_3CLASS. The model shows block-wise evidence aggregation and emphasizes pages/regions near the splice site (purple box).

multi-page fusion layer independently per page (i.e., single-page fusion), and then aggregate pages via mean pooling to obtain a document-level representation for classification. To generate a page-specific explanation for page $p$, we backpropagate the score $s$ of the target class to the multi-page fusion activation map $A_p \in \mathbb{R}^{C \times H \times W}$ and compute a gradient-weighted

saliency map:

$$\mathrm{CAM}_p(i,j) = \mathrm{ReLU}\left(\sum_{c=1}^{C} \alpha_{p,c} A_{p,c}(i,j)\right), \quad \alpha_{p,c} = \frac{1}{HW} \sum_{i=1}^{H} \sum_{j=1}^{W} \frac{\partial s}{\partial A_{p,c}(i,j)}.$$

The resulting heatmap is upsampled to the page resolution and overlaid on the rendered DNA page. We additionally report a page-level importance score by averaging the upsampled heatmap values, enabling a coarse ranking of pages by their overall attribution mass.

**Case study on SPLICE_SITE_3CLASS (Japonica test set).** We select two multi-page DNA documents with *donor* labels from the Japonica test set and visualize the first eight pages in Figure S4 and Figure S5. Across both examples, the model shows region-level evidence aggregation: instead of responding to individual bases or characters, the Grad-CAM highlights extend over short continuous regions, reflecting the spatial grouping induced by visual tokenization and the encoder's receptive fields. This behavior contrasts with the fine-grained, token-by-token attention patterns typically observed in 1D sequence models, and is closer to human reading strategies that process text at the level of phrases or lines rather than single letters. Such region-level processing naturally scales to long DNA sequences, as it avoids exhaustive base-wise scanning and allows the model to focus computation on locally informative regions under a fixed token budget. The purple box indicates the annotated donor splice site; Grad-CAM activations concentrate around this region, and pages containing the splice site tend to exhibit higher page-level mean attribution scores (reported below each page). Together, these observations suggest that OpticalDNA aligns its visual evidence with biologically meaningful donor loci and emphasizes the corresponding pages during inference, highlighting its practical interpretability and applicability for large-scale genomic analysis.

## L. Token Bottleneck Analysis

To examine whether OpticalDNA's downstream performance depends critically on the retained visual-token budget, we conduct a token bottleneck analysis on the nine eQTL tasks. Starting from the default OpticalDNA representation with 100 visual tokens, we progressively reduce the token budget by applying non-overlapping mean pooling over the flattened visual-token sequence.

Specifically, given a flattened visual-token sequence, we merge every $m$ adjacent tokens by mean pooling with stride $m$. If the token length is not divisible by $m$, the remaining tail tokens are also averaged into a final merged token. The default setting corresponds to $m = 1$, i.e., no token merging. Larger values of $m$ produce fewer retained tokens and stronger bottlenecks.

As shown in Table S35, OpticalDNA remains stable under moderate token compression. Reducing the token budget from 100 to 50 or 20 retained tokens yields only mild performance changes, with average AUROC remaining 0.842–0.846 compared with 0.852 under the original 100-token setting. Performance degrades more noticeably under severe bottlenecks of 10 tokens or fewer, and drops to 0.776 when the entire sequence is compressed into a single token. Overall, OpticalDNA is robust to moderate token compression, while clear degradation appears only under severe bottlenecks.

*Table S35.* Token bottleneck analysis on nine eQTL tasks. The merge factor denotes the non-overlapping mean-pooling window over flattened visual tokens. The original OpticalDNA setting uses 100 retained tokens. Results are reported as AUROC.

| Merge factor | Retained tokens | AS | AT | CCF | MS | NT | SNSES | SSELL | Thyroid | WB | Avg. |
|---|---|---|---|---|---|---|---|---|---|---|---|
| 1 | 100 | 0.813 | 0.852 | 0.812 | 0.880 | 0.884 | 0.904 | 0.835 | 0.791 | 0.895 | **0.852** |
| 2 | 50 | 0.774 | 0.838 | 0.785 | 0.880 | 0.867 | 0.912 | 0.844 | 0.775 | 0.903 | 0.842 |
| 5 | 20 | 0.802 | 0.863 | 0.740 | 0.855 | 0.883 | 0.925 | 0.869 | 0.776 | 0.903 | 0.846 |
| 10 | 10 | 0.783 | 0.850 | 0.711 | 0.850 | 0.877 | 0.903 | 0.809 | 0.767 | 0.854 | 0.823 |
| 20 | 5 | 0.728 | 0.835 | 0.713 | 0.851 | 0.843 | 0.895 | 0.800 | 0.758 | 0.829 | 0.806 |
| 25 | 4 | 0.746 | 0.845 | 0.708 | 0.838 | 0.833 | 0.894 | 0.778 | 0.760 | 0.808 | 0.801 |
| 50 | 2 | 0.735 | 0.829 | 0.684 | 0.826 | 0.835 | 0.859 | 0.741 | 0.731 | 0.794 | 0.782 |
| 100 | 1 | 0.737 | 0.792 | 0.706 | 0.811 | 0.839 | 0.838 | 0.733 | 0.742 | 0.784 | 0.776 |

## M. End-to-End Computational Cost Analysis

We evaluate the end-to-end computational cost of OpticalDNA and representative genomic foundation models under a unified protocol on binary classification datasets. All models use frozen backbones with only a task-specific linear head

trained for downstream evaluation, and all measurements are conducted on identical hardware. Table S36 reports fine-tuning throughput, FLOPs per base, peak GPU memory, trainable parameter counts, inference throughput, and total parameters activated at inference. OpticalDNA achieves higher fine-tuning and inference throughput than JanusDNA, NT-v2-500M, and GENERator-1.2B, while maintaining moderate GPU memory usage. Overall, OpticalDNA maintains strong downstream performance while remaining computationally practical, providing an efficient solution for long-context genomic modeling.

*Table S36.* End-to-end computational cost comparison under a unified protocol on binary classification datasets. All models use frozen backbones with a task-specific linear head for downstream evaluation, and throughput is measured in million bases per second (M bases/s). Trainable Params denotes the number of trainable parameters during fine-tuning, while Total Params denotes the parameters activated at inference.

| Model | Fine-tuning | | | | Inference | | |
|---|---|---|---|---|---|---|---|
| | Thrpt. ↑ | MFLOPs/base ↓ | Mem. (GB) ↓ | Trainable Params | Thrpt. ↑ | Mem. (GB) ↓ | Total Params |
| JanusDNA | 0.05 | 17.95 | 3.18 | 145 | 0.05 | 3.56 | 35.61M |
| Caduceus | 0.52 | 27.27 | 13.83 | 257 | 0.53 | 13.31 | 14.02M |
| NT-v2-500M | 0.10 | 204.79 | 3.22 | 1,025 | 0.10 | 3.22 | 498.35M |
| GENERator-1.2B | 0.04 | 381.77 | 5.98 | 10,241 | 0.04 | 5.98 | 1.15B |
| OpticalDNA | 0.12 | 177.76 | 5.81 | 128,001 | 0.12 | 5.82 | 408.06M |

## N. Rendering and Formatting Robustness

Since OpticalDNA represents DNA sequences as visual documents, an important question is whether the learned representations are overly sensitive to rendering choices or image-resolution changes. We therefore conduct systematic ablations over document formatting, low-resolution downsampling, and representation consistency under different visual perturbations. These experiments are designed to answer two questions: (i) whether downstream performance depends on a particular rendering configuration, and (ii) whether identical DNA segments produce consistent representations when rendered with different visual formats.

### N.1. Page-Layout Ablation

We first evaluate how downstream performance changes under different page-layout configurations. Rendering is jointly determined by several factors, including text density, page aspect ratio, layout mode, starting position, and font. We conduct this ablation on the Adipose Subcutaneous eQTL task and report AUROC in Table S37. The default configuration uses a $640 \times 640$ canvas, font size 14, line spacing 1.6, continuous rendering, and margins of 20 pixels on all sides.

Overall, OpticalDNA is not tied to a single rendering format. Several variants match or outperform the default layout. For example, increasing text density improves AUROC from 0.813 to 0.838 (dense_square), while a wide and dense layout further improves performance to 0.858 (wide_dense). Font changes also remain competitive, with sans-serif and serif variants achieving 0.839 (sans) and 0.835 (serif) AUROC, respectively. These results indicate that the model does not rely on a particular font or exact page geometry.

At the same time, layout choices matter in a structured way. Continuous layouts that preserve horizontal sequence continuity tend to perform better than rigid fixed-window layouts. For example, wide continuous rendering achieves 0.820 AUROC (wide_layout), whereas fixed-window layouts with 32, 48, or 64 bp per line obtain 0.758–0.774 AUROC (fixed_win_32, fixed_win_48, fixed_win_64). Similarly, a fixed-block layout with 64 bp windows and one gap line reaches 0.777 AUROC (fixed_block_64_gap1). This suggests that imposing artificial local segmentation can disrupt useful continuity patterns, whereas continuous rendering better preserves local sequence flow in the visual document.

Starting-position perturbations also remain reasonably stable. Shifting the content toward the top-left gives 0.783 AUROC (shift_00_20_00_20), while shifting it away from the top-left gives 0.811 AUROC (shift_40_20_40_20), close to the default 0.813. Together, these results show that OpticalDNA is robust to common rendering variations, while favoring layouts that preserve continuous local reading structure.

*Table S37.* Page-layout ablation on the Adipose Subcutaneous eQTL task. "cont" denotes continuous rendering, "win-line" denotes fixed-window rendering by line, and "win-block" denotes fixed-window rendering by block. For compactness, "shift_00_20_00_20" abbreviates "start_pos_shift_00_20_00_20", and "shift_40_20_40_20" abbreviates "start_pos_shift_40_20_40_20".

| Scheme | Canvas | Font | Spacing | Layout | Window | Gap | Margins (T,B,L,R) | Description | AUROC |
|---|---|---|---|---|---|---|---|---|---|
| original | 640×640 | 14 | 1.6 | cont | – | – | 20,20,20,20 | default | 0.813 |
| dense_square | 640×640 | 12 | 1.4 | cont | – | – | 20,20,20,20 | denser text | 0.838 |
| sparse_square | 640×640 | 16 | 1.8 | cont | – | – | 20,20,20,20 | sparse text | 0.819 |
| wide_layout | 800×512 | 14 | 1.6 | cont | – | – | 20,20,20,20 | wider page | 0.820 |
| tall_layout | 512×800 | 14 | 1.6 | cont | – | – | 20,20,20,20 | taller page | 0.809 |
| wide_dense | 800×512 | 12 | 1.4 | cont | – | – | 20,20,20,20 | wide+dense | 0.858 |
| tall_sparse | 512×800 | 16 | 1.8 | cont | – | – | 20,20,20,20 | tall+sparse | 0.808 |
| fixed_win_32 | 640×640 | 14 | 1.6 | win-line | 32 | 0 | 20,20,20,20 | 32-bp fixed windows/line | 0.758 |
| fixed_win_48 | 640×640 | 14 | 1.6 | win-line | 48 | 0 | 20,20,20,20 | 48-bp fixed windows/line | 0.771 |
| fixed_win_64 | 640×640 | 14 | 1.6 | win-line | 64 | 0 | 20,20,20,20 | 64-bp fixed windows/line | 0.774 |
| fixed_block_64_gap1 | 640×640 | 14 | 1.6 | win-block | 64 | 1 | 20,20,20,20 | 64-bp windows in blocks | 0.777 |
| shift_00_20_00_20 | 640×640 | 14 | 1.6 | cont | – | – | 0,20,0,20 | shifted toward top-left | 0.783 |
| shift_40_20_40_20 | 640×640 | 14 | 1.6 | cont | – | – | 40,20,40,20 | shifted away top-left | 0.811 |
| sans | 640×640 | 14 | 1.6 | cont | – | – | 20,20,20,20 | sans-serif font | 0.839 |
| serif | 640×640 | 14 | 1.6 | cont | – | – | 20,20,20,20 | serif font | 0.835 |

## N.2. Low-Resolution Downsampling

We further evaluate whether OpticalDNA is sensitive to low-resolution failures by downsampling the rendered $640 \times 640$ DNA images to smaller resolutions while keeping the underlying DNA content and layout fixed. We conduct the analysis on the Adipose Subcutaneous eQTL task and report AUROC in Table S38. OpticalDNA shows graceful degradation under downsampling: performance remains strong at moderate resolutions, with $256 \times 256$ yielding 0.812 AUROC, nearly matching the original $640 \times 640$ result of 0.813, and $320 \times 320$ achieving 0.828 AUROC. Clear degradation appears mainly under extreme downsampling, where AUROC drops to 0.799 at $128 \times 128$ and 0.768 at $64 \times 64$. These results indicate that OpticalDNA is robust to moderate resolution reduction, while very low resolutions can degrade performance by removing fine-grained nucleotide-level visual details.

*Table S38.* Low-resolution downsampling analysis on the Adipose Subcutaneous eQTL task. Rendered $640 \times 640$ DNA images are resized to different resolutions before feature extraction.

| Resolution | AUROC |
|---|---|
| $640 \times 640$ (original) | 0.813 |
| $320 \times 320$ | **0.828** |
| $256 \times 256$ | 0.812 |
| $128 \times 128$ | 0.799 |
| $64 \times 64$ | 0.768 |

## N.3. Representation Consistency under Formatting Perturbations

We further evaluate whether identical DNA segments yield consistent representations when rendered with different visual formats. For each DNA segment, we generate multiple rendered versions under different formatting perturbations and compute pairwise cosine similarity between their extracted representations. The results are shown in Figure S6.

On the Adipose Subcutaneous test split, the learned representations remain highly consistent under common formatting changes. Starting-position shifts yield cosine similarities between 0.904 and 0.985, with an average of 0.937. Resolution changes also preserve strong consistency, with similarities ranging from 0.776 to 0.984 and an average of 0.899. General layout perturbations achieve an average similarity of 0.879. Similar trends are observed on the training and validation splits: starting-position perturbations achieve average similarities of 0.940 and 0.940, while resolution perturbations achieve 0.903 and 0.904, respectively.

The main degradation occurs only under extreme layout changes, such as comparing dense continuous rendering with fixed-window rendering, where the similarity can drop to 0.564. This is consistent with the downstream ablation in Table S37: fixed-window layouts impose rigid local segmentation and are less aligned with continuous sequence reading. Therefore, the representation-consistency analysis supports the same conclusion as the downstream ablation: OpticalDNA is robust

to common rendering variations, but extreme layout changes that disrupt local sequence continuity can alter the learned representation.

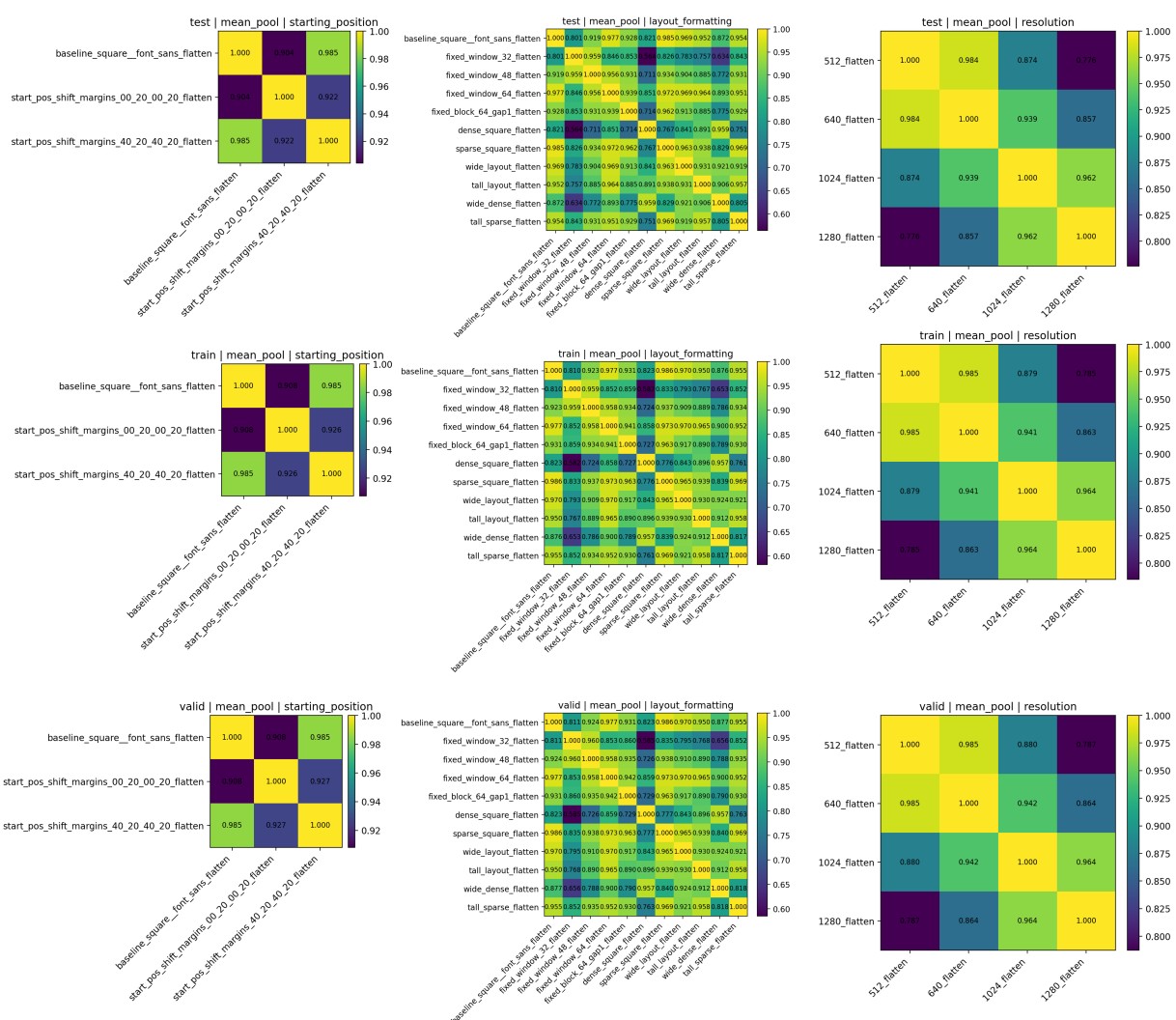

*Figure S6.* Representation consistency under rendering perturbations. We render identical DNA segments using different visual formats and compute pairwise cosine similarity between the extracted representations. Results are shown on the test, train, and validation splits using mean-pooled features. The left column evaluates starting-position shifts, the middle column evaluates broader layout-formatting changes, and the right column evaluates image-resolution changes. OpticalDNA maintains high representation similarity under common perturbations such as starting-position shifts and resolution changes, while larger drops mainly occur under extreme layout changes such as fixed-window formatting.

## N.4. Effect of Fragment Partitioning and Arrangement

How a DNA sequence is partitioned and arranged on the page is an important rendering choice. Even when the nucleotide content is unchanged, different line breaks, window boundaries, or block separators can change which bases appear as local neighbors in the visual document. Therefore, we further examine whether OpticalDNA relies on a specific fragment arrangement, or whether it remains effective when the same DNA content is organized under different partitioning schemes.

The fixed-window and fixed-block settings in Table S37 provide a stress test for this issue. Unlike continuous rendering, which writes nucleotides sequentially across each line, fixed-window layouts impose artificial boundaries every 32, 48, or 64 bp, and the fixed-block setting further separates windows with blank lines. These settings alter local neighborhood structure,

line breaks, and fragment arrangement while preserving the underlying DNA content.

The results show that rigid partitioning is less favorable than continuous rendering, but does not make the representation unusable. Fixed-window and fixed-block layouts achieve 0.758–0.777 AUROC, compared with 0.813 for the default continuous layout and up to 0.858 for the wide dense continuous layout. This indicates that OpticalDNA is not restricted to a single fragment arrangement, but also reveals a clear preference: preserving local sequence continuity provides a more effective computational structure than imposing artificial window boundaries.

Together with the representation-consistency analysis in Figure S6, these results suggest that OpticalDNA is robust to common formatting variations, while extreme perturbations that disrupt local sequence continuity can change the learned representation and moderately reduce downstream performance.

## O. Cross-Species Transfer and Pretraining Fairness Analysis

The main rice experiments compare rice-pretrained OpticalDNA with large multi-species genomic foundation models such as Evo-2 and LucaOne. Although these are important and competitive baselines, their pretraining corpora and task alignment differ from OpticalDNA. We therefore conduct additional analyses to disentangle the effect of the OCR-style visual document formulation from the effect of domain-matched rice pretraining.

### O.1. RiceSubBench under Cross-Species Transfer

We first evaluate cross-species transfer from human HG38 to rice on RiceSubBench. Specifically, Caduceus, JanusDNA, and OpticalDNA-HG38 are compared under the same HG38-pretraining setting. We additionally report OpticalDNA-Rice as a domain-matched reference. Results are shown in Table S39.

Under the matched HG38-pretraining setting, OpticalDNA-HG38 consistently outperforms the HG38-pretrained sequence baselines across all five rice datasets. Averaged over the five datasets, OpticalDNA-HG38 achieves 0.595 accuracy and 0.745 AUROC, compared with 0.449/0.568 for Caduceus and 0.477/0.673 for JanusDNA. This indicates that the advantage of OpticalDNA is not simply explained by rice-domain pretraining.

OpticalDNA-HG38 also remains close to OpticalDNA-Rice. The average accuracy differs by only 0.003, and the average AUROC remains comparable. This suggests that the visual document formulation transfers across species and is not tightly tied to a specific genome used during pretraining.

*Table S39.* Cross-species transfer on RiceSubBench. Caduceus, JanusDNA, and OpticalDNA-HG38 are compared under the same HG38-pretraining setting, while OpticalDNA-Rice is reported as a domain-matched reference. Results are reported as Accuracy / AUROC.

| Model | japonica | aus | rufipogon | barthii | glaberrima |
|---|---|---|---|---|---|
| Caduceus (HG38) | 0.490 / 0.569 | 0.444 / 0.577 | 0.438 / 0.584 | 0.429 / 0.601 | 0.442 / 0.508 |
| JanusDNA (HG38) | 0.448 / 0.638 | 0.464 / 0.689 | 0.533 / 0.670 | 0.488 / 0.701 | 0.453 / 0.665 |
| OpticalDNA (HG38) | 0.587 / **0.744** | 0.555 / **0.727** | 0.637 / **0.774** | **0.616** / **0.753** | 0.580 / 0.726 |
| OpticalDNA (Rice) | **0.590** / 0.739 | **0.556** / 0.725 | **0.639** / 0.762 | 0.608 / 0.747 | **0.599** / **0.731** |

### O.2. RiceWGPB Control

*Table S40.* RiceWGPB control results. RMSE is reported for thousand grain weight (TGW) and leaf rolling index measured in the 2015 SZ environment (LRI-15SZ). Lower is better.

| Model | TGW ↓ | LRI-15SZ ↓ |
|---|---|---|
| Caduceus | 25.14 | 9.92 |
| LucaOne | 8.82 | 9.74 |
| Evo-2 | 3.06 | 9.62 |
| OpticalDNA (Rice) | **2.95** | **9.53** |

We further include a RiceWGPB control to contextualize the whole-genome phenotype prediction results. Table S40 reports RMSE on thousand grain weight (TGW) and leaf rolling index measured in the 2015 SZ environment (LRI-15SZ). Lower values indicate better performance. Caduceus remains substantially worse than OpticalDNA-Rice on TGW and does not

surpass Evo-2 or LucaOne on either trait. OpticalDNA-Rice achieves the best RMSE on both TGW and LRI-15SZ.

