# OpenReview forum: "Rethinking Genomic Modeling Through Optical Character Recognition"
_ICML.cc/2026/Conference — ICML 2026 regular_

### Official Review · Reviewer_rnrm · 2026-03-05

**Soundness:** 3
**Presentation:** 3
**Significance:** 3
**Originality:** 3
**Overall Recommendation:** 4
**Confidence:** 3

**Summary:**

This paper argues that long-context genomic modeling can benefit from reframing DNA as a “document understanding” problem. It proposes OpticalDNA: (i) render a 1D genomic sequence into a multi-page DNA document with nucleotide glyphs plus bounding-box and global-index annotations, and (ii) pretrain a visual encoder–document decoder using six OCR-style prompted tasks (T1–T6) covering recognition, grounding, retrieval, and completion. After pretraining, the method extracts fixed-dimensional embeddings by pooling per-page visual features and trains lightweight heads for downstream prediction. Experiments report strong results on long-sequence genotype–phenotype benchmarks, including DNALONGBench eQTL AUROC gains over prior long-context DNA models (e.g., JanusDNA; Duan et al., 2025) and favorable OOD transfer on rice subspecies and whole-genome phenotype prediction versus large genome foundation baselines (Evo-2; Brixi et al., 2025; LucaOne; He et al., 2025).

**Compliance With Llm Reviewing Policy:**

Affirmed.

**Final Justification:**

I think this paper has a genuinely interesting idea: reframing long-context genomic modeling as an OCR-style document understanding problem is novel, and the paper backs it up with strong enough results on long-range, OOD, and genome-scale settings to make the contribution feel meaningful. The paper is also generally clear and easy to follow.

My main concerns were about positioning, efficiency reporting, and whether the empirical analysis was broad enough. The rebuttal addressed these reasonably well. In particular, the authors clarified that the main contribution is the new formulation rather than a brand-new architectural primitive, added more standardized efficiency discussion, and provided extra evidence on rendering sensitivity, token bottlenecks, stronger 1D baselines, and scope/failure cases. That made the paper easier to trust.

**Key Questions For Authors:**

see in weakness.

**Limitations:**

- Scope/feasibility: clarify the regimes where document rendering is expected to work well (sequence lengths, species shifts, ambiguity codes beyond {A,C,G,T,N}, structural variation), and how sensitive performance is to rendering/layout choices (font/line breaks/page size) and the fixed-token bottleneck pooling strategy.
- Failure modes: discuss OCR-style brittleness (rendering artifacts, truncation, low-resolution failure, mismatch between training “glyph” distribution and deployment), and how mean-pooling across pages might discard long-range interactions needed for distal regulation.

**Strengths And Weaknesses:**

Strengths
- Clear reframing + unified supervision: the multi-page “DNA document” formulation with OCR-style prompts (T1–T6) provides a coherent way to supervise transcription, localization, and retrieval primitives in one interface, rather than relying purely on masked modeling or k-mer objectives.
- Strong long-context empirical performance: OpticalDNA achieves the best average AUROC on DNALONGBench, and the lightweight MLP head improves average AUROC further while outperforming the prior SOTA JanusDNA variants on multiple tissues (Duan et al., 2025).
- Robustness under distribution shift: on RiceSubBench, OpticalDNA reportedly outperforms large multi-species foundation baselines (Evo-2; Brixi et al., 2025; LucaOne; He et al., 2025), with the gap widening for far-OOD subspecies.
- Practical genome-scale efficiency evidence: on ~400M-base rice genomes, it reports substantially lower inference time than Evo-2/LucaOne while achieving the lowest RMSE on the evaluated traits, suggesting the approach can be operational at whole-genome scale.

Weaknesses
- Novelty is somewhat “systems recombination” rather than a new modeling primitive: the core machinery is built on mature document/OCR components (e.g., DeepSeek-OCR-style visual encoder + decoder), with the main novelty coming from the DNA rendering + prompt/task design. This is valuable, but the paper should be careful about positioning as a fundamentally new genomic modeling paradigm.
- Efficiency/parameter reporting is potentially confusing: results emphasize very small “activated parameters” for probing heads (e.g., 256K) while the representation extractor itself is a 409M-parameter visual encoder; the paper should standardize reporting (total inference params, trainable params, FLOPs, wall-clock) to avoid apples-to-oranges comparisons across baselines.
- Limited ablation isolating what matters: since pretraining mixes six task families with a specified sampling distribution, the paper would benefit from systematic ablations (T1-only vs +T2–T6; rendering hyperparameters; page resolution; fusion/token bottleneck) to identify which ingredients drive downstream gains.
- Evaluation breadth is still narrow relative to the ambition: the benchmarks are sensible, but the paper would be stronger with additional canonical functional genomics tasks (or clearer justification for why the chosen three cover the main intended use cases), plus more detailed error analyses (where OCR-style grounding helps vs fails).

---

> ### Author Rebuttal · Authors · 2026-03-30
>
> Dear reviewer rnrm,
>
> Thank you for your insightful comments!
> >**W#1:Novelty is somewhat “systems recombination”...**
>
> We agree that OpticalDNA is not a new architectural primitive and will clarify this positioning.
>
> That said, the contribution goes beyond systems recombination. The core novelty lies in reformulating long-context genomic modeling as document-style selective reading. Sequences are rendered as documents, tasks unified via OCR-style interfaces, and supervision grounded at region level—changing how computation is allocated over long genomics sequences, rather than merely swapping in an existing backbone.
>
> This targets a key bottleneck in long-context genomic modeling(see **jghd W#1**) and yields a better accuracy–efficiency trade-off, indicating the computational interface to the sequence is itself an important source of progress.We will revise the paper to position OpticalDNA as a new formulation and computational framework.
> >**W#2: Efficiency/parameter reporting...**
>
> We agree efficiency/parameter reporting should be standardized. We first clarify Table 2 consistently reports activated(trainable) parameters under each setting, not total backbone size;thus 7.662M(JanusDNA) and 256K(OpticalDNA, linear probing) are directly comparable as active parameters.
>
> We further add unified profiling of total inference params,trainable params,FLOPs,throughput,and memory(see **38Yf Q#3**). Under this setting, OpticalDNA shows higher throughput than JanusDNA,NT-v2,and GENERator in both training and inference, with moderate memory usage.We will clarify definitions in the revision.
> >**W#3:...systematic ablations...**
>
> We address this along three axes. For pretraining-OCR-task ablations, we note that conducting such ablations during the rebuttal period is difficult(see **38Yf Q#1**).
>
> Rendering sensitivity is covered by controlled ablations(see **jghd W#2**,**fB99 W#2**,**Q#2**), showing gains are not tied to specific settings.
>
> For token bottleneck, we perform a ablation by varying retained tokens. Performance is stable under moderate compression and degrades only under severe bottlenecks(≤10 tokens), indicating robustness to token budget(**Table R11**).
>
> **Table R11**
> |#Token|Avg.AUROC|
> |-|-|
> |100(original OpticalDNA)|0.852|
> |50|0.842|
> |25|0.846|
> |10|0.823|
> |5|0.806|
> |4|0.801|
> |2|0.782|
> |1|0.776|
> >**Q#1:Scope/feasibility:...sensitive performance to rendering choices...and fixed-token bottleneck pooling strategy.**
>
> We clarify regimes and add cross-species evidence. HG38-pretrained OpticalDNA transfers well to rice, achieving comparable performance than rice-pretrained models across taxa, indicating the formulation is not species-specific(Table R12).
>
> Other regimes are supported: sequence lengths(**Q#2**), structural variation(whole-genome/cross-species results), rendering sensitivity(**jghd W#2**, **fB99 W#2**, **Q#2**),and resolution(**Q#2**).
>
> Fixed-token bottleneck sensitivity is addressed by the token-budget ablation(see **W#3**), where performance remains stable under moderate compression and degrades clearly only under severe bottlenecks.
>
> For ambiguity codes beyond{A,C,G,T,N},we do not include a separate study. We restrict inputs to{A,C,G,T,N} to align with standard benchmarks and ensure fair comparison; handling richer ambiguity codes is left for future work.
>
> **Table R12.Cross-species transfer(ACC/AUROC)**
> |Model|japonica|aus|rufipogon|barthii|glaberrima|
> |-|-|-|-|-|-|
> |OpticalDNA(Rice)|0.590/0.739|0.556/0.725|0.639/0.762|0.608/0.747|0.599/0.731|
> |OpticalDNA(HG38)|0.587/0.744|0.555/0.727|0.637/0.774|0.616/0.753|0.580/0.726|
> >**Q#2:Failure modes...how mean-pooling across pages might discard long-range interactions...**
>
> We analyze OCR-style failure modes.
>
> Font changes show minimal impact(0.839/0.835 vs. 0.813,see **jghd W#2**).
>
> Downsampling shows graceful degradation: strong performance is retained at moderate resolutions, drops only at extreme resolutions(64–128,**Table R13**).
>
> OCR accuracy decreases when sequence density exceeds canvas capacity: EM/CS remains high below 1024(75.0–85.9%/93.4–99.7%), but degrades for longer spans(65.5/83.4% at [1536,1792),42.3/76.4% at [1792,2048];**Table R14**).
>
> We agree cross-page mean pooling is lossy;in our design it provides lightweight global context, while fine-grained predictions rely on full page tokens.We therefore view this as a limitation of aggregation,not the OCR-style formulation.
>
> **Table R13**
> |Resolution|AS AUROC|
> |-|-|
> |640×640(original)|0.813|
> |64×64|0.768|
> |128×128|0.799|
> |256×256|0.812|
> |320×320|0.828|
>
> **Table R14**
> |Retained length|#Sample|EM(%)↑|CS(%)↑|
> |-|-|-|-|
> |<256|104|75.0|99.7|
> |[256,512)|144|77.8|95.7|
> |[512,768)|135|85.9|96.6|
> |[768,1024)|145|77.9|93.4|
> |[1024,1280)|140|72.1|89.0|
> |[1280,1536)|122|71.3|87.3|
> |[1536,1792)|148|65.5|83.4|
> |[1792,2048]|137|42.3|76.4|
>
> ---
>
> Thank you again for valuable feedback, which helped improve this work! We hope our revisions have addressed your concerns and positively informed your evaluation.

---

> > ### Author Rebuttal · Reviewer_rnrm · 2026-04-03
> >
> > Thank you for the detailed rebuttal. I will keep my score at 4.

---

> > > ### Author Response · Authors · 2026-04-04
> > >
> > > Dear reviewer rnrm,
> > >
> > > Thank you for the acknowledgment and for your thoughtful review. We are glad that our responses have addressed your concerns. We sincerely appreciate your time and constructive feedback, which has helped improve the clarity of our work.
> > >
> > > Best regards,
> > >
> > > Authors

---

### Official Review · Reviewer_38Yf · 2026-03-11

**Soundness:** 3
**Presentation:** 3
**Significance:** 3
**Originality:** 3
**Overall Recommendation:** 4
**Confidence:** 4

**Summary:**

This paper focuses on long-context genomic foundation modeling, especially the tension between biological fidelity and computational efficiency. Current DNA foundation models mostly inherit 1D language-model reading patterns, which may be poorly matched to sparse, discontinuous genomic signals and costly at very long context lengths. The paper proposes OpticalDNA, which renders DNA into multi-page 2D “documents” and trains an OCR-style vision–language system with prompted tasks for reading, grounding, retrieval, completion, and classification. The main methodological contribution is the reformulation itself: DNA is no longer treated as a flat token stream, but as a spatially indexed visual object with bounding-box annotations and OCR-style supervision.

**Compliance With Llm Reviewing Policy:**

Affirmed.

**Final Justification:**

The rebuttal makes a meaningful effort to address several of my concerns, and some issues are partially resolved

**Key Questions For Authors:**

The downstream results mainly assess representation quality using simple probing heads on eQTL, rice classification, and phenotype prediction. What is missing is stronger evidence that the OCR-style tasks themselves are necessary or biologically meaningful. For example, I would like to see task ablations: how much do T2/T3/T5 grounding-style tasks contribute relative to simpler masked or contrastive pretraining? Right now the task suite is plausible, but not fully justified.

Can you compare against Nucleotide Transformer and GENErator, or explain clearly why those comparisons are omitted? These are highly relevant recent baselines for genomic foundation modeling.

Can you provide a real end-to-end efficiency study: training cost, inference throughput per million bases, GPU memory, and matched-budget comparisons against JanusDNA, HyenaDNA, or Evo-like models? Table 7 is too limited for the strength of the claims.

**Limitations:**

yes

**Strengths And Weaknesses:**

### Strengths
The paper proposes a clearly novel angle. Recasting genomic modeling as OCR-style document understanding is a fresh framing, and the paper develops it into a coherent training pipeline rather than leaving it as a metaphor.

The six prompt families in Table 1 cover reading, grounding, ROI transcription, masked completion, retrieval, and chromosome-level recognition. Even if not all of them are equally central, this is a nontrivial attempt to align pretraining tasks with practical genomic operations.

The paper evaluates on three regimes: long-range eQTL tasks (Table 2), rice subspecies OOD generalization (Table 3), and whole-genome phenotype prediction (Table 4). This is extensive.

### Weaknesses
The paper argues that genomic semantics are sparse and discontinuous, so 2D OCR-style reading is structurally better than sequential reading. This is plausible, but the paper does not really establish that the chosen 2D rasterization preserves the biologically meaningful inductive biases better than strong sequence models. In particular, the rendering procedure in Section 3.2 (lines 147–164) is extremely simple: write nucleotides row by row, with no explicit genomic landmarks, no strand-aware structure, and no biologically grounded spatial layout. So the method may be benefiting more from compression and pretraining transfer from OCR backbones than from a genuinely better genomic geometry.

OpticalDNA uses a frozen SAM–Conv–CLIP-L front-end and a DeepSeek-3B-based document decoder in Section 3.4, explicitly building on DeepSeek-OCR-like components. This matters because DeepSeek-OCR itself is explicitly motivated as a vision-based compression mechanism for long-context text. So it is unclear how much of the gain comes from the genomic reformulation itself versus simply importing a very strong OCR compression backbone and decoder. The ablation against DeepSeek-OCR in Section 4.3 is helpful, but still limited.

For RiceSubBench and RiceWGPB, the paper compares a rice-pretrained OpticalDNA to very large multi-species models such as Evo-2 and LucaOne. Those are important baselines, but not necessarily the fairest ones if the key claim is superiority of the OCR formulation rather than superiority of domain-matched pretraining. This concern affects Tables 3 and 4, where the paper emphasizes much smaller parameter count, but the pretraining data and task alignment differ substantially.

---

> ### Author Rebuttal · Authors · 2026-03-30
>
> Dear reviewer 38Yf,
>
> Thank you for the insightful comments!
> >**W#1:...whether 2D rasterization preserves biologically meaningful...benefit from compression and pretraining transfer...**
>
> Our rasterization is intentionally simple and introduce no biological priors. Instead, it defines a bijective reparameterization of genomic sequence preserving sequence content.
>
> Rather than redesigning genomic geometry, our goal is to reorganize how computation is allocated over ultra-long genomic sequences(see **jghd W#1**). While compression and OCR pretraining contribute, the key effect is semantic compression-allocating capacity to task-relevant regions while suppressing background-ariseing from access pattern induced by the representation rather than compression or backbone alone.
>
> Empirically, this advantage holds under controlled comparisons(2Dvs.1D, including 1D Transformers;see **fB99 W#1**), indicating gains arise from computation reorganization under long-context genomic sparsity.
> > **Q#1:...OCR-style tasks ablations...**
>
> We agree OCR-style task ablations are valuable. However, these tasks are jointly optimized and not independent modules, so their effects cannot be cleanly isolated without full retraining. More importantly, their role is to shape computation allocation in representations, which is reflected in overall downstream performance, not per-task contributions. Each ablation requires full pretraining(2 weeks on 8×H100); a full suite would require \~3 months and >$30k, making it infeasible within rebuttal timeframe.
> > **W#3:...fairness...superiority of OCR formulation rather than domain-matched pretraining...**
>
> We add pretraining-controlled baselines.
>
> On RiceSubBench(**Table R6**), we compare Caduceus, JanusDNA, and OpticalDNA under matched HG38 pretraining, removing domain and task-alignment effects. OpticalDNA (HG38) consistently rank first across taxa while remaining competitive with rice-pretrained OpticalDNA, showing gains are not due to domain alignment. Evo-2 and LucaOne still outperform Caduceus and JanusDNA, remaining strong multi-species baselines.
>
> On RiceWGPB(**Table R7**), Caduceus(matched protocol) remains substantially worse than OpticalDNA(Rice) (2.95/9.53) and not surpassing Evo-2 or LucaOne.Additional runs(JanusDNA,OpticalDNA-HG38) are in progress and will be included in the final version.
>
> Overall, OpticalDNA retains advantage under substantially fairer controls, indicating gains arise from OCR-based formulation rather than pretraining differences.
>
> **Table R6.RiceSubBench(ACC/AUROC)**
> |Model|japonica|aus|rufipogon|barthii|glaberrima|
> |-|-|-|-|-|-|
> |Caduceus|0.490/0.569|0.444/0.577|0.438/0.584|0.429/0.601|0.442/0.508|
> |JanusDNA|0.448/0.638|0.464/0.689|0.533/0.670|0.488/0.701|0.453/0.665|
> |OpticalDNA-HG38|**0.587/0.744**|**0.555/0.727**|**0.637/0.774**|**0.616/0.753**|**0.580/0.726**|
>
> **Table R7.RiceWGPB(RMSE)**
> |Model|TGW↓|LRI↓|
> |-|-|-|
> |Caduceus|25.14|9.92|
> |LucaOne|8.82|9.74|
> |Evo-2|3.06|9.62|
> |OpticalDNA(Rice)|**2.95**|**9.53**|
> > **Q#2: ...compare against NT and GENErator...**
>
> We add NT and GENERator on 450 kb eQTL task(**Table R8**).As neither supports ultra-long inputs natively, we apply a unified frozen-backbone adaptation:NT-v2 uses 2,048-token chunks(\~12 kb) with mean pooling,and GENERator uses 16,384-token chunks(\~98 kb) with concatenated summaries;in both cases, the backbone is frozen and only a lightweight head is trained.
>
> NT-v2 achieves 0.798/0.772(val/test AUROC) and GENERator 0.808/0.782 across 9 GTEx tissues,both below OpticalDNA(0.852/0.867).Thus, OpticalDNA consistently outperforms strong sequence baselines in the long-context regime.
>
> **Table R8**
> |Model|NT-v2-500M|GENERator-1.2B|OpticalDNA(Linear)| OpticalDNA(MLP)|
> |-|-|-|-|-|
> |Avg test AUROC|0.7723|0.7821|**0.8520**|**0.8670**|
> > **Q#3: ...efficiency study...**
>
> We add a standardized end-to-end efficiency study under a unified profiling protocol.To avoid ambiguity, we explicitly distinguish fine-tuning parameters from total inference parameters.
>
> To ensure a controlled comparison, all models are evaluated under the same protocol: frozen backbones with only a task-specific linear head trained. Experiments are run on identical hardware, yielding comparable results(**Tables R9-10**),where OpticalDNA achieves a favorable efficiency–performance trade-off.
>
> **Table R9.Fine-tuning**
> |Model|Thrpt(M bases/s)|MFLOPs/base|Peak Mem(GB)|Trainable Params(M)|
> |-|-|-|-|-|
> |JanusDNA|0.05|17.95|3.18|0.15|
> |Caduceus|0.52|27.27|13.83|0.26|
> |NT-v2-500M|0.10|204.79|3.22|1.03|
> |GENERator-1.2B|0.04|381.77|5.98|10.24|
> |OpticalDNA|0.12|177.76|5.81|128.00|
>
> **Table R10.Inference**
> |Model|Thrpt(M bases/s)|Params|Peak Mem(GB)|
> |-|-|-|-|
> |JanusDNA|0.05|35.61M|3.56|
> |Caduceus|0.53|14.02M|13.31|
> |NT-v2-500M|0.10|498.35M|3.22|
> |GENERator-1.2B|0.04|1.15B|5.98|
> |OpticalDNA|0.12|408.06M|5.82|
>
> ---
>
> We appreciate your insightful comments! We hope our responses have addressed your concerns and support a more favorable assessment.

---

> > ### Author Rebuttal · Reviewer_38Yf · 2026-04-05
> >
> > The rebuttal makes a meaningful effort to address several of my concerns. The addition of pretraining-controlled baselinesand comparisons with NT and GENERator improves the fairness of evaluation and strengthens the empirical case that OpticalDNA’s gains are not solely due to domain-matched pretraining. Based on this, I will raise my score at 4.
> >
> > My primary concern remains that, the authors clarify the 2D rasterization is a bijective reparameterization and gains arise from “computation reorganization” rather than biological inductive bias. While this is a reasonable position, it effectively shifts the claim rather than substantiating the original intuition that the representation better matches genomic structure. The provided empirical comparisons (2D vs. 1D) are referenced but not sufficiently detailed in the rebuttal to convincingly disentangle representation effects from backbone and compression advantages.

---

> > > ### Author Response · Authors · 2026-04-07
> > >
> > > Dear reviewer 38Yf，
> > >
> > > Thank you again for your thoughtful follow-up and for raising your score—we truly appreciate your recognition that the additional analyses improve the fairness of the evaluation. We also thank you for clearly articulating the remaining concern.
> > > >**ACK-Q#1: My primary concern remains...the representation better matches genomic structure.**
> > >
> > > We appreciate this clarification. Upon reading your prvious comment, we realized that our earlier phrasing may have introduced ambiguity, potentially leading to the impression that we claim the 2D representation better matches **biological structure**, as reflected in **Weakness 1**:
> > > >...does not really establish that the chosen 2D rasterization preserves the biologically meaningful inductive biases better than strong sequence models. In particular, the rendering procedure...is extremely simple...with no explicit genomic landmarks, no strand-aware structure, and no biologically grounded spatial layout. So the method may be benefiting more from compression and pretraining transfer from OCR backbones than from a genuinely better genomic geometry.
> > >
> > > Recognizing this, we emphasized in our previous rebuttal that OpticalDNA does not incorporate the biological inductive biases mentioned above (e.g., genomic landmarks, strand-aware structure, biologically grounded spatial layouts or genomic geometry). However, due to space limitations, this point may not have been made sufficiently explicit. To avoid ambiguity, we restate the relevant sentences from our **Introduction**:
> > > >Existing genomic foundation models inherit inductive biases from natural language modeling, assuming dense semantics under exhaustive token-by-token reading. However, genomic function is sparse and discontinuous, embedded in long stretches of lowinformation background, leading to jump-like dependencies across distant loci (Fig. 1a). As a result, sequential reading is structurally wasteful: most computation is spent scanning background rather than modeling sparse functional regions. In controlled comparisons...indicating that purely sequential processing is suboptimal even in long-context settings.
> > >
> > > Our focus is thus not on modeling biological structure, but on addressing a **computational mismatch** induced by genomic sparsity, which becomes particularly pronounced in long-context settings. Specifically, we target a key limitation of 1D sequence models at long sequence lengths—the accuracy–efficiency trade-off inherent in sequential processing. This perspective motivates our design and is reflected in the long-context regimes (450 kb–400 Mb) considered in our downstream tasks.
> > >
> > > In this sense, “structure” refers to **computational structure** (i.e., how information is accessed and processed), rather than biological structure. The 2D rasterization is a lossless, bijective reparameterization of the 1D sequence, preserving exact nucleotide correspondence while reorganizing computation. Accordingly, our claim is not that 2D better matches genomic structure, but that it enables more effective allocation of computation under sparse, discontinuous signals.
> > >
> > > If our wording suggested otherwise, we sincerely appreciate the opportunity to clarify this point and will revise the manuscript to make this distinction more explicit.
> > > >**ACK-Q#2: The provided...disentangle representation effects from backbone and compression advantages**
> > >
> > > We agree this was not sufficiently isolated previously. Our earlier comparisons matched model scale but did not strictly control token count. To address this, we add a controlled experiment with **matched token budget (10k)** and **shared Transformer backbone**, so that the primary difference is the representation.
> > >
> > > The same genomic input is represented either as a 1D sequence or a 2D image document (336×336), tokenized and reduced to an identical set of 10,000 tokens, and processed by the same model under the same pretraining pipeline. Under this design, the downstream model and token budget are fixed, so the main remaining difference is the representation itself.
> > >
> > > On the 9-tissue eQTL benchmark (**Table R16**), 2D achieves 0.626 AUROC vs. 0.558 for 1D (+6.8% absolute, ↑12.20% relative). This suggests that, once backbone and token budget are controlled, the improvement cannot be explained solely by compression or architecture, but is closely tied to the representation itself—i.e., how the document-style formulation organizes sparse, discontinuous genomic information into a more effective input structure for long-context prediction.
> > >
> > > **Table R16**. Average AUROC on the 9-tissue eQTL benchmark under matched token count and shared Transformer backbone
> > > | Representation | #Token | Average AUROC |
> > > |-|-|-|
> > > | 1D Sequence | 10,000 | 0.558 |
> > > | 2D Image Document (336×336) | 10,000 | **0.626** |
> > > | Δ | - | ↑12.20% |
> > >
> > > ---
> > >
> > > Thank you again for your careful and constructive feedback! We hope the above clarifications have helped address your concerns. We truly appreciate your consideration.

---

### Official Review · Reviewer_fB99 · 2026-03-12

**Soundness:** 2
**Presentation:** 3
**Significance:** 3
**Originality:** 3
**Overall Recommendation:** 4
**Confidence:** 3

**Summary:**

This work proposes OpticalDNA, a novel approach for genomic modeling that converts DNA sequences into structured visual layouts rather than treating them as one-dimensional token sequences. Inspired by DeepSeek-ORC, the authors train an OCR-capable visual-language model on six ORC-inspired tasks, designed to align with practical genomic analysis workflows. The pretrained visual encoder can then be adapted to various downstream tasks via lightweight prediction heads. Experiments demonstrate that OpticalDNA consistently outperforms state-of-the-art baselines on several benchmarks, including DNALONGGBench, RiceSbBench, and the rice whole-genome phenotype benchmark.

**Compliance With Llm Reviewing Policy:**

Affirmed.

**Final Justification:**

Thank the authors for their detailed response. I hope the authors can incorporate the thorough discussions from the rebuttal into the revised manuscript, and I would like to maintain my positive score.

**Key Questions For Authors:**

1. How was test set contamination handled? Did the authors carry out any sequence similarity analysis between the pretraining corpora and the benchmarks?
2. Has the model been tested on completely novel, unseen, or pseudo-protein sequences to confirm that the encoder truly interprets the visual representations rather than relying on pretraining memorization?
3. How robust is the encoder to variations in rendering parameters and to different DNA fragments?

**Limitations:**

yes

**Strengths And Weaknesses:**

Strengths
* The paper is well-motivated, proposing a spatial visual representation to capture long-range dependencies in DNA. The structure and presentation are clear and easy to follow.

* The approach is innovative, leveraging a visual encoder for DNA modeling significantly reduces token requirements compared to conventional sequence-based methods.

* The method shows consistent improvements over baselines across multiple benchmarks, supporting the practical utility of the proposed approach.


Weaknesses
* The paper justifies 2D modeling by comparing it against 1D CNNs, which inherently suffer from localized receptive fields. Without a direct comparison to 1D Transformers which natively capture global interactions, it remains unclear if the performance gain stems from the 2D representation itself.

* The paper lacks analysis on the encoder's sensitivity to visual formatting. It is unclear if identical DNA segments yield consistent representations under variations like arbitrary starting positions, line breaks, or different rendering resolutions.

* Given the large-scale pretraining corpora, potential data leakage into downstream benchmarks is unaddressed. This raises concerns about whether the model genuinely learns biological semantics or merely memorizes pretraining patterns.[1]


[1] Visual Merit or Linguistic Crutch? A Close Look at DeepSeek-OCR. https://arxiv.org/abs/2601.03714

---

> ### Author Rebuttal · Authors · 2026-03-30
>
> Dear reviewer fB99,
>
> Thank you for your constructive comments! We address the concerns below.
> >**W#1:..comparison to 1D Transformers...**
>
> We agree and include 1D Transformer baselines. On the same 9-tissue eQTL setting (**Table R3**), Transformer-tiny achieves 0.475 AUROC(0.406M) and Transformer-small 0.575, both substantially below MobileNetV3-S(0.760) and TinyNet-E(0.786), while being significantly less efficient.
>
> These results rule out CNN locality as the primary driver:even with global 1D modeling, performance remains substantially lower. This indicates the advantage arises from the document-style representation and computation pattern, which better aligns with sparse genomic signals.
>
> **Table R3**
> |Model|AUROC|Params(M)|Epoch time(s)↓|Thrpt(samples/s)↑|
> |-|-|-|-|-|
> |Transformer-tiny|0.475|0.406|1150.2|1.363|
> |Transformer-small|0.575|1.793|2980.8|0.47|
> |MobileNetV3-S(0.50×)|0.760|0.570|138.3|16.1|
> |TinyNet-E|0.786|0.764|66.9|33.7|
> >**W#2:...encoder's sensitivity to visual formatting...**
>
> We address this by representation-consistency analysis(cosine similarity,**Fig. R1**:https://anonymous.4open.science/r/OpticalDNA-rebuttal/Figure-R1.png), testing whether identical DNA segments yield stable representations under formatting perturbations.
>
> On the AS test split, pairwise cosine similarity remains high under starting-position shifts(0.904-0.985,avg.0.937), resolution changes(0.776–0.984;avg.0.899), and layout perturbations(avg. 0.879), with degradation only under extreme layouts(e.g., dense vs. fixed-window:0.564). Similar trends hold on train/val splits(starting-position:0.940/0.940; resolution:0.903/0.904).
>
> Overall, representations are stable across formatting variations, with only mild degradation under extreme layouts.
> >**W#3&Q1:...data leakage...How was test set contamination handled?...**
>
> We assess train–test leakage using strict sequence-level criteria.
>
> For eQTL(**Table R4**), overlap is counted only under exact 450 kb sequence matching. Only 12/4297 samples overlap(0.279%), meaning 99.721% are novel.
>
> For rice(**Table R5**), we use an equally strict contig-level criterion: a test FASTA contig is flagged only if fully contained within a training contig of the same task.Only 136/27822 contigs match(0.489%), meaning 99.51% are novel(0.515%/0.471% per task).
>
> We report this exact-match/exact-containment analysis rather than a looser approximate-similarity analysis because it directly tests whether benchmark test examples reappear in the downstream training data. Under this setting, contamination is negligible in both eQTL and rice, making memorization an unlikely explanation.
>
> **Table R4.eQTL benchmark**
> |Metric|Count|Rate|
> |-|-|-|
> |Matched test sequences|12/4297|0.2793%|
> |Novel test sequences|4285/4297|99.721%|
>
> **Table R5.rice benchmark**
> |Metric|Count|Rate|
> |-|-|-|
> |Matched test contigs (all rice tasks)|136/27822|0.4888%|
> |Novel test contigs (all rice tasks)|27686/27822|99.5112%|
> |Matched test contigs (leaf\_rolling\_index\_2015SZ-%)|58 /11253|0.5154%|
> |Matched test contigs (thousand\_grain\_weight-g.)|78/16569|0.4708%|
> >**Q#2:..tested on novel, unseen, pseudo-protein sequences to confirm the encoder interprets visual representations rather than pretraining memorization?**
>
> We did not add pseudo/randomized DNA sequence experiment. However, the strict contamination analysis above (**W#3&Q#1**) already addresses memorization concern: 99.72%(eQTL) and 99.51%(rice) of test sequences are unseen. Thus, gains are unlikely driven by memorization. Since our setting is genomic rather than protein modeling, we will revise the wording to refer to pseudo-DNA sequences rather than pseudo-protein sequences.
> >**Q#3:How robust is the encoder to variations in rendering parameters and to different DNA fragments?**
>
> Please refer to **jghd W#2** for detailed rendering ablations. Briefly, the encoder remains robust across substantial rendering variations. Under changes in density, aspect ratio, layout, start position, and font, multiple configurations remain competitive or improve over the default(0.813 AUROC;range 0.819–0.858).
>
> Our results also support effectiveness across different DNA fragments, since both training/evaluation conducted over a large number of distinct fragments rather than a single fixed sequence instance. Moreover, the explicit fixed-window settings (fixed_window_32/48/64,fixed_block_64_gap1) further test different fragment partitioning and arrangement schemes. Although these rigid local organizations are slightly weaker than continuous layouts(0.758–0.777), they remain effective.
>
> Overall, the encoder is not tied to specific rendering choices or fragment configurations, while layouts preserving horizontal continuity are more favorable.
>
> ---
>
> Thanks again for your constructive comments, which has helped improv this work! We hope our revisions have addressed your concerns and lead to a more favorable overall assessment.

---

> > ### Author Rebuttal · Reviewer_fB99 · 2026-04-02
> >
> > Thank you for your detailed response.
> >
> > Most of my concerns have been addressed, except for those related to potential contamination. I would appreciate further clarification on the following points:
> >
> > 1. How is sequence matching computed?
> > 2. What degree of fragment overlap would lead to consistent results?
> > 3. To what extent might the data have been implicitly learned during pretraining, and how should we interpret the generalization capability of the proposed approach on novel DNA sequences under this consideration?

---

> > > ### Author Response · Authors · 2026-04-04
> > >
> > > Dear reviewer fB99,
> > >
> > > Thank you for your thoughtful follow-up. We address the questions with clear empirical evidence below.
> > > >**ACK-Q#1: How is sequence matching computed?**
> > >
> > > For both the eQTL and rice benchmarks, each test sequence is compared against every training sequence, and a match is recorded only when the full test sequence is exactly identical to a training sequence.
> > > >**ACK-Q#2: What degree of fragment overlap would lead to consistent results?**
> > >
> > > We perform an empirical analysis on **RiceSubBench** and find that **consistent results arise only under near-complete fragment overlap; partial overlap does not reliably produce consistent labels**.
> > >
> > > Test samples are grouped by **20-mer containment** (fraction of unique 20-mers shared with training;**Table R15**), and **Top-1/Top-5/Top-10 label consistency** is computed against nearest training samples.
> > >
> > > The trend in **Table R15** is clear. Across **0.0–0.9**, **Top-1 consistency** remains flat(**\~41–55%**) with no monotonic trend. The largest bin **[0.0,0.1)**(57.13% of all samples) achieves only **47.52%**, and even high-overlap bins **(0.7–0.9)** remain around **49-55%**.
> > >
> > > A sharp increase appears only **near complete overlap: [0.9, 1.0)** reaches **76.85%** Top-1 consistency, while **exact overlap[1.0]** occur in only **2 samples(0.07%)**. Thus, the empirical answer is that **only near-complete overlap yields consistent predictions.**
> > >
> > > This is biologically expected: partial overlap is ubiquitous due to the limited nucleotide alphabet and recurrent local patterns, but do not determine functional equivalence. Genomic function depends on broader compositional context(order,spacing, orientation,long-range interactions), so similar sequences can yield different labels—even with high sequence identity(e.g., human–chimpanzee).
> > >
> > > Therefore, we do not regard moderate fragment overlap as a reliable indicator of leakage, and we use exact matching(100%) as a strict and unambiguous criterion to avoid conflating leakage with inherent redundancy.
> > >
> > > **Table R15**
> > > |k-mer Containment Bin|#Test Samples|Test Sample Ratio(%)|Top-1 Label Consistency(%)|Top-5 Label Consistency(%)|Top-10 Label Consistency(%)|
> > > |-|-|-|-|-|-|
> > > |[0.0,0.1)|1650|57.13|47.52|45.82|45.00|
> > > |[0.1,0.2)|96|3.32|41.67|41.25|43.33|
> > > |[0.2,0.3)|66|2.29|48.48|50.00|45.76|
> > > |[0.3,0.4)|62|2.15|50.00|50.65|47.74|
> > > |[0.4,0.5)|60|2.08|55.00|50.67|45.50|
> > > |[0.5,0.6)|74|2.56|51.35|51.08|49.73|
> > > |[0.6,0.7)|88|3.05|54.55|49.55|46.82|
> > > |[0.7,0.8)|147|5.09|55.10|44.76|43.27|
> > > |[0.8,0.9)|224|7.76|49.11|46.43|46.47|
> > > |[0.9,1.0)|419|14.51|76.85|73.75|70.19|
> > > |[1.0]|2|0.07|100.00|60.00|55.00|
> > > >**ACK-Q#3:To what extent might the data have been implicitly learned during pretraining, and how should we interpret the generalization capability...**
> > >
> > > We do not attribute the observed performance of OpticalDNA on novel DNA sequences primarily to implicit learning of downstream-like sequences during pretraining.
> > >
> > > (1) Our pretraining is fully unlabeled; even if similar sequences appear, no task-specific supervision is available.
> > >
> > > (2) **Table R15** directly falsifies a memorization hypothesis: even under substantial overlap, prediction consistency is far from perfect. If memorization dominated, high-overlap regions would approach near-perfect agreement, which is not observed.
> > >
> > > (3) Some degree of sequence sharing is unavoidable when pretraining and evaluation draw from related genomic corpora, and applies to all genomic foundation models (e.g., JanusDNA and Evo-2). The key question is not whether overlap exists, but whether performance is driven by memorization. If so, models with stronger memorization should consistently dominate. However, recent work shows that memorization strength varies widely across architectures and settings, yet does not correlate clearly with downstream performance[1]. This suggests that the gains are better explained by how the model accesses and organizes genomic information, rather than by sequence recall alone.
> > >
> > > (4) Cross-species transfer provides a stricter test: HG38-pretrained OpticalDNA remains effective on rice downstream tasks(**Table R6**) despite large evolutionary distance and low sequence similarity between human and rice. Such transfer is difficult to reconcile with fragment-level recall(minimal overlap), and instead indicates that OpticalDNA captures transferable genomic regularities.
> > >
> > > Overall, while implicit exposure during pretraining cannot be fully ruled out, the evidence consistently argues against a memory-driven explanation. The gains are better explained by pattern-level generalization enabled, rather than instance-level recall.
> > >
> > > [1] Quantifying Memorization and Privacy Risks in Genomic Language Models
> > >
> > > ---
> > > Thank you again for your positive assessment and taking time to review our work thoroughly. We would be happy to provide any additional clarification if helpful. If you feel that the concerns have been addressed, we would sincerely appreciate your support, which would mean a great deal to us.

---

### Official Review · Reviewer_jghd · 2026-03-14

**Soundness:** 3
**Presentation:** 3
**Significance:** 3
**Originality:** 3
**Overall Recommendation:** 4
**Confidence:** 3

**Summary:**

- The paper proposed an approach towards genomic modeling — unlike recent LLM based approaches which model genomic 1D sequences as strings on ACTG alphabets and fit a sqeuence model — the authors propose a vision-based framework that models genomic modeling as OCR-style document understanding problem.
- The 1D sequence of ACTG alphabets is rendered as documents / pages and vision tokenizers (w/ decoder) are fit to it, followed by linear MLP based downstream fitting.
- The core reason as visual tokenizers can compress more than alphabet-wise sequence modeling.

**Compliance With Llm Reviewing Policy:**

Affirmed.

**Final Justification:**

The main paper, solid rebuttal response and other reviews all align towards an accept.
While still unclear to me why OCR is an intuitive and scalable approach in future to study 1D genomic sequences, the paper still presents a novel study with solid experimentation and thus is worthy of acceptance.

**Key Questions For Authors:**

Answering the weakness points would be great

**Limitations:**

yes

**Strengths And Weaknesses:**

### Strengths
- The paper is novel — a new take on genomic sequence modeling by casting the sequence as documents and using visual tokenizers for representation learning
- The paper is well written overall and easy to follow — the motivation in introduction is also clear and precise.
- Key Result from the paper: OpticalDNA improves DeepSeek-OCR on all tissues, yielding a +5.37% relative gain on the overall average

### Weakness
- While the approach is novel and an interesting take to reduce the number of context tokens in sequence modeling — it’s not clear intuitively since the data is naturally 1D and 1D sequence models seem to be a better fit.
- The method appears highly sensitive to documnet rendering, but rendering choices are not ablated. Because the page layout is central to the appraoch, the paper would benefit from much more detail and explicit ablations on how DNA windows are arranged into documents and how performnce depends on these choices. The current experiments vary resolution, but do not talk about nrendering scheme itself.
- Since the main point of using vision is token reduction is some sense — better sequence modeling approaches like RNNs, linear attention, sparse attention can atleast be talked about. Current baselines do not seem to cover such architectural choices — so this seems missing. Comment on this would be appreciated.

---

> ### Author Rebuttal · Authors · 2026-03-30
>
> Dear Reviewer jghd,
>
> We thank the reviewer for the thoughtful comments. We address the concerns below.
> > **W#1:..not clear intuitively since the data is naturally 1D and 1D sequence models seem to better fit.**
>
> We agree that DNA is inherently 1D and sequence models are a natural starting point. Our goal is not to replace this view, but to target a key limitation of 1D sequence models in long-context genomic settings—the accuracy–efficiency bottleneck of sequential processing. This is why all downstream tasks focus on 450 kb-400 Mb.
>
> At this scale, the trade-off becomes unavoidable. In our experiments(\~400M bases), Evo-2 supports longer contexts but requires \~6h inference, while LucaOne reduces cost via splitting at the expense of long-range accuracy. In contrast, OpticalDNA achieves minutes-level inference with the lowest RMSE.
>
> The issue is not the 1D nature of DNA, but sequential allocation of computation. Genomic signals are sparse and region-centric, yet sequence models allocate computation uniformly, leading to inefficient scaling. By preserving the full sequence via bijective mapping while reorganizing access, our approach enables region-level selective computation, leading to more effective long-range modeling.
>
> Consistently, under comparable settings, 2D backbones outperform 1D variants on 450kb eQTLs tasks. Importantly, this holds with newly added 1D Transformer baselines (see **fB99 W#1**), indicating gains in long-context regimes stem from reorganizing computation to better align with sparse genomic signals.
>
> > **W#2:...rendering choices are not ablated...**
>
> As layout is central to OpticalDNA, we conduct a comprehensive page-layout ablation on the Adipose_Subcutaneous eQTL task.
>
> Rendering is jointly determined by text density, aspect ratio, layout mode, and start position(**Table R1**). Results show OpticalDNA is not tied to a specific layout: multiple variants match or exceed the default(0.813), such as dense_square(0.838), wide_dense(0.858), and font variations(0.835–0.839).
>
> We see a consistent trend: layouts preserving horizontal continuity perform better(wide_layout 0.820, tall_layout 0.809), while fixed-window schemes are consistently worse(0.758–0.777), indicating rigid local segmentation is suboptimal. Start-position shifts remain stable(0.783–0.811).
>
> Overall, OpticalDNA is not fragile to rendering choice, though layout matters in a structured way.
>
> **Table R1**
>
> Layout: cont = continuous；win-line = fixed_window_line; win-box = fixed_window_box
> Scheme|Canvas (H×W)|Font size|Line spacing|Layout|Window size|Gap lines|Margins (T,B,L,R)|Description|AUROC|
> |-|-|-|-|-|-|-|-|-|-|
> original|640×640|14|1.6|cont|-|-|20,20,20,20|default|0.813|
> dense_square|640×640|12|1.4|cont|-|-|20,20,20,20|denser text|0.838|
> sparse_square|640×640|16|1.8|cont|-|-|20,20,20,20|sparse text|0.819|
> wide_layout|800×512|14|1.6|cont|-|-|20,20,20,20|wider page|0.820|
> tall_layout|512×800|14|1.6|cont|-|-|20,20,20,20|taller page|0.809|
> wide_dense|800×512|12|1.4|cont|-|-|20,20,20,20|wide+dense|0.858|
> tall_sparse|512×800|16|1.8|cont|-|-|20,20,20,20|tall+sparse|0.808|
> fixed_win_32|640×640|14|1.6|win-line|32|0|20,20,20,20|32-bp fixed windows/line|0.758|
> fixed_win_48|640×640|14|1.6|win-line|48|0|20,20,20,20|48-bp fixed windows/line|0.771|
> fixed_win_64|640×640|14|1.6|win-line|64|0|20,20,20,20|64-bp fixed windows/line|0.774|
> fixed_block_64_gap1|640×640|14|1.6|win-block|64|1|20,20,20,20|64-bp windows in blocks (gap=1)|0.777|
> start_pos_shift_margins_00_20_00_20|640×640|14|1.6|cont|-|-|0,20,0,20|shifted toward top-left|0.783|
> start_pos_shift_margins_40_20_40_20|640×640|14|1.6|cont|-|-|40,20,40,20|shifted away top-left|0.811|
> sans|640×640|14|1.6|cont|-|-|20,20,20,20|sans-serif font|0.839|
> serif|640×640|14|1.6|cont|-|-|20,20,20,20|serif font|0.835|
> > **W#3: ...approaches like RNNs, linear attention, sparse attention can be talked...**
>
> **Table R2**
> |Model|AUROC|Epoch time(s)|Thrpt|Params(M)|
> |-|-|-|-|-|
> |RNN|0.6575|27.27|82.92|0.73|
> |LinearAttn|0.6279|25.42|88.84|3.29|
> |SparseAttn|0.6163|95.48|23.69|3.29|
>
> We agree stronger 1D sequence baselines beyond CNNs should be included.We add three: BiGRU(RNN), Performer(linear attention), and Longformer(sparse attention), trained on 9 GTEx eQTL tissues under identical settings.
>
> As shown in **Table R2**, the average test AUROC is 0.6575(RNN), 0.6279(linear attention), and 0.6163(sparse attention), all below 2D baselines TinyNet-E(0.786) and OpticalDNA(0.852 / 0.867 with linear probing / MLP). RNN performs best among 1D baselines, while sparse attention incurs much higher cost without gains.
>
> These results strengthen our conclusion: the advantage of OpticalDNA arises from the document-style representation better aligned with sparse genomic signals and achieves superior accuracy–efficiency trade-off.
>
> ---
>
> Thank you again for your time and valuable feedback, which has improved the quality of this work! We hope our revisions have addressed your concerns and positively informed your evaluation.

---

> > ### Author Rebuttal · Reviewer_jghd · 2026-04-03
> >
> > Authors did a great job in addressing my 3 weakness points -- training in a quick time. I will keep my rating of 4.

---

> > > ### Author Response · Authors · 2026-04-04
> > >
> > > Dear reviewer jghd,
> > >
> > > Thank you for your acknowledgment and thoughtful review. We are glad that our responses have clarified the concerns. We truly appreciate your time and insightful feedback, which has helped strengthen the presentation of our work.
> > >
> > > Best regards,
> > >
> > > Authors

---

### Decision · Program_Chairs · 2026-04-30

**Decision:**

Accept (regular)

**Comment:**

The paper is inspired by the success of DeepSeekOCR and leverages a similar OCR-style formulation for genomic modeling, reframing long DNA sequences as structured visual documents rather than purely 1D token streams. Reviewers consistently highlighted the paper’s novelty, clear motivation, and coherent overall framework. They also found the empirical results strong across long-context, OOD, and genome-scale benchmarks, and viewed the approach as practically meaningful. Overall, the work presents a creative and well-executed direction that is likely to stimulate future research in the genomic modeling.